# FLATNESS-AWARE STOCHASTIC GRADIENT LANGEVIN DYNAMICS

## ABSTRACT

Generalization in deep learning is closely tied to the pursuit of flat minima in the loss landscape, yet classical Stochastic Gradient Langevin Dynamics (SGLD) offers no mechanism to bias its dynamics toward such low-curvature solutions. This work introduces Flatness-Aware Stochastic Gradient Langevin Dynamics (fS-GLD), designed to efficiently and provably seek flat minima in high-dimensional nonconvex optimization problems. At each iteration, fSGLD uses the stochastic gradient evaluated at parameters perturbed by isotropic Gaussian noise, commonly referred to as Random Weight Perturbation (RWP), thereby optimizing a randomized-smoothing objective that implicitly captures curvature information. Leveraging these properties, we prove that the invariant measure of fSGLD stays close to a stationary measure concentrated on the global minimizers of a loss function regularized by the Hessian trace whenever the inverse temperature and the scale of random weight perturbation are properly coupled. This result provides a rigorous theoretical explanation for the benefits of random weight perturbation. In particular, we establish non-asymptotic convergence guarantees in Wasserstein distance with the best known rate and derive an excess-risk bound for the Hessian-trace regularized objective. Extensive experiments on noisy-label and large-scale vision tasks, in both training-from-scratch and fine-tuning settings, demonstrate that fSGLD achieves superior or comparable generalization and robustness to baseline algorithms while maintaining the computational cost of SGD, about half that of SAM. Hessian-spectrum analysis further confirms that fSGLD converges to significantly flatter minima.

## 1 INTRODUCTION

Consider the overdamped Langevin dynamics governed by the stochastic differential equation (SDE)

$$\mathrm{d}Z_t = -\nabla u(Z_t)\mathrm{d}t + \sqrt{2\beta^{-1}}\mathrm{d}B_t, \qquad (1)$$

which admits a unique invariant (Gibbs) measure $\pi_\beta(\theta)$ proportional to $\exp(-\beta u(\theta))$, where $\beta > 0$ is the inverse temperature and $(B_t)_{t \geq 0}$ is a $d$-dimensional Brownian motion. As $\beta$ increases, this Gibbs measure concentrates on the global minimizers of $u$, establishing a direct link between Langevin dynamics and global optimization. Building on this property, Stochastic Gradient Langevin Dynamics (SGLD) (Welling & Teh, 2011; Raginsky et al., 2017) was proposed as the Euler-Maruyama discretization of the Langevin SDE in which the exact gradient $\nabla u$ is replaced by a stochastic gradient. SGLD has attracted considerable attention as a prominent optimization algorithm for nonconvex problems, and under mild regularity conditions a series of works has established non-asymptotic global convergence guarantees (Raginsky et al., 2017; Xu et al., 2018; Majka et al., 2020; Chau et al., 2021; Zhang et al., 2023). Despite these elegant theoretical results, SGLD has not become a widely used optimizer in deep learning practice, largely because it lacks an intrinsic mechanism to favor flat minima, which are closely associated to strong generalization.

Alongside advances in SGLD, a separate line of work in deep learning has explored flatter solutions to improve generalization, inspired by the flat minima hypothesis (Hochreiter & Schmidhuber, 1997). As a result, numerous flatness-aware optimization algorithms have been developed, including Random Weight Perturbation (RWP) (Bisla et al., 2022; Li et al., 2024a), Entropy-SGD (Chaudhari et al., 2017), Entropy-MCMC (Li & Zhang, 2024), Sharpness-Aware Minimization (SAM) (Foret et al., 2021) and their variants (Xie et al., 2024; Li et al., 2024b; Tahmasebi et al., 2024; Luo et al.,

2024; Chen et al., 2024; Kang et al., 2025; Wei et al., 2025; Liu et al., 2022a;b; Du et al., 2022b; Li et al., 2025). In principle, flatness-aware optimization promotes exploration of flat regions by replacing the standard stochastic gradient with a perturbed gradient. For example, SAM applies a worst-case adversarial perturbation within a local neighborhood, whereas RWP uses symmetric random noise to generate the gradient perturbation and can be viewed as computing the stochastic gradient of a randomized-smoothing objective (Duchi et al., 2012). However, SAM's min–max formulation requires double gradient evaluations, leading to roughly twice the computational cost of standard SGD. On the theoretical side, recent studies have produced important advances in the analysis of SAM and related flatness-aware optimization methods, yielding valuable insights on generalization bounds, stability, and (local) convergence properties; e.g., see Andriushchenko & Flammarion (2022); Bartlett et al. (2023); Si & Yun (2023); Yu et al. (2024); Khanh et al. (2024); Oikonomou & Loizou (2025); Zhang et al. (2024); Li et al. (2024a). However, with a few notable exceptions (Ahn et al., 2024; Gatmiry et al., 2024), the global convergence properties of flatness-aware optimization in nonconvex settings, as well as a rigorous theoretical understanding of the role of RWP, remain relatively unexplored.

To address these challenges, we introduce Flatness-Aware Stochastic Gradient Langevin Dynamics (fSGLD), a principled synthesis of randomized smoothing and Langevin dynamics that efficiently explores flat minima. While randomized-smoothing surrogates are known to encode second-order information such as the Hessian trace, they also contain higher-order remainder terms of which effects are not negligible in high-dimensional nonconvex problems, weakening the intended flatness-aware regularization effect. Our key theoretical contribution is to show that when the two key hyper-parameters, the inverse temperature parameter $\beta$ and the perturbation scale $\sigma$, are properly balanced, the invariant measure of fSGLD concentrates on the global minimizers of the true Hessian-trace regularized objective, thereby isolating the genuine flatness-aware regularization effect. This principled coupling is crucial, as it ensures that the global exploration driven by Langevin dynamics is effectively guided across a landscape smoothed by the perturbation noise, steering the process toward genuinely flat regions. In particular, we establish non-asymptotic convergence guarantees in Wasserstein distance and an explicit excess-risk bound for the Hessian-trace-regularized objective, providing the rigorous evidence of the benefits of RWP in nonconvex settings. Our framework bridges and advances the theory and practice of flatness-aware stochastic optimization, opening new avenues to incorporate geometric smoothing into Langevin sampling and paving the way for more effective and principled flatness-regularized learning. To validate these results, we evaluate fSGLD on noisy-label datasets (CIFAR-10N/100N, WebVision) and large-scale vision fine-tuning (ViT-B/16). Extensive experiments demonstrate that fSGLD consistently matches or outperforms baselines including SGD, AdamW, SGLD, and SAM in generalization and robustness while maintaining the computational cost of standard SGD. Notably, using the theoretically prescribed coupling between $\beta$ and $\sigma$ yields substantially better performance than simply fixing a large $\beta$, which is the common SGLD practice. In summary, fSGLD is the first to combine the SGLD framework with the concept of flatness and to provide a global convergence analysis for flatness-aware optimization, thereby advancing the theoretical and practical foundations of both areas.

## 2 PROBLEM SETTING AND FSGLD ALGORITHM

**Notation.** Let $(\Omega, \mathcal{F}, \mathbb{P})$ be a fixed probability space. We denote the probability law of a random variable $Y$ by $\mathcal{L}(Y)$. Fix integers $d, m \geq 1$. Let $I_d$ be the identity matrix of dimension $d$. The Euclidean scalar product is denoted by $\langle \cdot, \cdot \rangle$, with $|\cdot|$ standing for the corresponding norm. Let $f : \mathbb{R}^d \to \mathbb{R}$ be a continuously differentiable function, and we denote its gradient by $\nabla f$. For any integer $q \geq 1$, let $\mathcal{P}(\mathbb{R}^q)$ be the set of probability measures on $\mathcal{B}(\mathbb{R}^q)$. For $\mu, \nu \in \mathcal{P}(\mathbb{R}^d)$, let $\mathcal{C}(\mu, \nu)$ denote the set of probability measures $\Gamma$ on $\mathcal{B}(\mathbb{R}^{2d})$ such that its respective marginals are $\mu$ and $\nu$. For any $\mu$ and $\nu \in \mathcal{P}(\mathbb{R}^d)$, the Wasserstein distance of order $p \geq 1$ is defined as

$$W_p(\mu, \nu) = \left( \inf_{\Gamma \in \mathcal{C}(\mu, \nu)} \int_{\mathbb{R}^d} \int_{\mathbb{R}^d} |x - y|^p \, \mathrm{d}\Gamma(x, y) \right)^{\frac{1}{p}}. \tag{2}$$

### 2.1 INTRACTABLE HESSIAN-BASED REGULARIZATION

We consider the following nonconvex stochastic optimization problem:

$$\min_{\theta \in \mathbb{R}^d} u(\theta) := \min_{\theta \in \mathbb{R}^d} \mathbb{E}\big[U(\theta, X)\big], \tag{3}$$

where $u : \mathbb{R}^d \to \mathbb{R}$ is a four-times continuously differentiable function with gradient $h := \nabla u$, $U : \mathbb{R}^d \times \mathbb{R}^m \to \mathbb{R}$ is a measurable function satisfying $\mathbb{E}[\|U(\theta, X)\|] < \infty$ for all $\theta \in \mathbb{R}^d$, and $X$ is a random variable with probability law $\mathcal{L}(X)$. In practice, the gradient $h$ of $u$ is usually unknown and one only has access to its unbiased estimate, i.e. $h(\theta) = \mathbb{E}[\nabla_\theta U(\theta, X)]$.

To improve generalization, we incorporate an inductive bias for flatness through a flatness-aware objective. More specifically, instead of optimizing the original objective $u$, we aim to solve the following *Hessian-trace regularized objective*:

$$v(\theta) := u(\theta) + \frac{\sigma^2}{2} \mathrm{tr}\left(H(\theta)\right), \tag{4}$$

where $\mathrm{tr}(H(\theta))$ is the trace of the Hessian of $u$ evaluated at $\theta$ and $\sigma > 0$ controls the strength of the sharpness regularization. The global minimizers of this regularized objective $v$ represent a trade-off between low loss from the original objective $u$ and low curvature. For brevity, we will refer to these points as the *global flat minima* (i.e., $\arg\min_{\theta \in \mathbb{R}^d} v(\theta)$). However, computing $\mathrm{tr}\left(H(\theta)\right)$ is expensive in high dimension.

## 2.2 Randomized Smoothing as a Tractable Surrogate

To obtain a tractable alternative to the Hessian-trace regularized objective in 4, we introduce a Gaussian perturbation $\epsilon \sim \mathcal{N}(0, \sigma^2 I_d)$ with $\sigma \in (0, 1)$, independent of $X$, and define the *randomized-smoothing surrogate objective*:

$$g_\epsilon(\theta) := \mathbb{E}\left[u(\theta + \epsilon)\right] = \mathbb{E}\left[\mathbb{E}_X\left[U(\theta + \epsilon, X)\right]\right]. \tag{5}$$

where the outer expectation is taken with respect to the noise $\epsilon$ and $\mathbb{E}_X[\cdot]$ denotes the conditional expectation given $\epsilon$. This simple surrogate allows us to access curvature information. By Taylor's theorem, we have

$$u(\theta + \epsilon) = u(\theta) + \nabla u(\theta)^\top \epsilon + \tfrac{1}{2} \epsilon^\top H(\theta) \epsilon + \mathcal{R}(\theta, \epsilon),$$

where $\mathcal{R}(\theta, \epsilon)$ is the remainder term. Taking the expectation over $\epsilon \sim \mathcal{N}(0, \sigma^2 I_d)$ yields the key connection:

$$\begin{aligned} g_\epsilon(\theta) &= u(\theta) + \tfrac{\sigma^2}{2} \mathrm{tr}\left(H(\theta)\right) + \mathbb{E}[\mathcal{R}(\theta, \epsilon)] \\ &= v(\theta) + \mathbb{E}[\mathcal{R}(\theta, \epsilon)]. \end{aligned} \tag{6}$$

Thus, optimizing the tractable surrogate $g_\epsilon$ introduces the desired inductive bias toward flat minima by implicitly minimizing the Hessian–trace regularized objective $v$, provided that the remainder term $\mathbb{E}[\mathcal{R}(\theta, \epsilon)]$ is negligible.

## 2.3 FSGLD Algorithm

To optimize the surrogate objective $g_\epsilon$ in 5, we propose the Flatness-Aware Stochastic Gradient Langevin Dynamics (fSGLD) algorithm. Formally, let $\theta_0$ be an $\mathbb{R}^d$-valued random variable representing the initial value, $(X_k)_{k \in \mathbb{N}}$ be an i.i.d sequence of data, $(\epsilon_k)_{k \in \mathbb{N}}$ be i.i.d copies of the Gaussian perturbation $\epsilon \sim \mathcal{N}(0, \sigma^2 I_d)$, and $(\xi_k)_{k \in \mathbb{N}}$ be an independent sequence of standard $d$-dimensional Gaussian random variables. We assume that $\theta_0$, $(\epsilon_k)_{k \in \mathbb{N}}$, and $(\xi_k)_{k \in \mathbb{N}}$ are all mutually independent. Then, the fSGLD algorithm is given by

$$\begin{cases} \theta_0^{\mathrm{fSGLD}} &:= \theta_0, \\ \theta_{k+1}^{\mathrm{fSGLD}} &= \theta_k^{\mathrm{fSGLD}} - \lambda \nabla_\theta U(\theta_k^{\mathrm{fSGLD}} + \epsilon_{k+1}, X_{k+1}) + \sqrt{2\lambda\beta^{-1}}\xi_{k+1}, \qquad k \in \mathbb{N} \end{cases} \tag{7}$$

where $\lambda > 0$ is the stepsize, $\beta > 0$ is the inverse temperature. We make three important remarks about this update rule. First, the gradient term in 7 is a unbiased stochastic gradient of $g_\epsilon$, as its expectation over both the data $X$ and the perturbation $\epsilon$ recovers the true gradient $\nabla g_\epsilon$:

$$\nabla g_\epsilon(\theta) = \mathbb{E}[\mathbb{E}_X[\nabla_\theta U(\theta + \epsilon, X)]]. \tag{8}$$

Second, the fSGLD can be interpreted as the standard SGLD for the original objective $u$ combined with RWP. Third, under appropriate conditions, which will be introduced in the next section, the fSGLD algorithm generates a Markov chain that converges to a unique invariant (Gibbs) measure. This measure, denoted by $\pi_\beta^{\mathrm{fSGLD}}$, is associated with the randomized-smoothing surrogate objective $g_\epsilon$, i.e., $\pi_\beta^{\mathrm{fSGLD}}(\theta) \propto \exp(-\beta g_\epsilon(\theta))$. The formal convergence guarantees are provided in Appendix C.2.

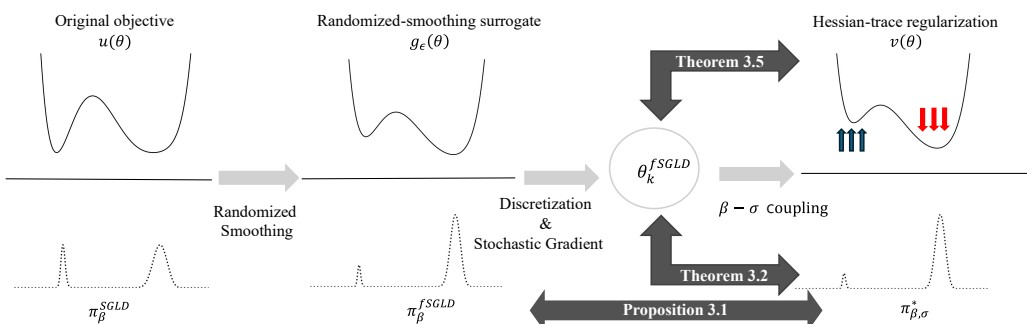

Figure 1: A schematic overview of the theoretical framework of fSGLD. The process begins with the **original objective** $u(\theta)$ and its associated Gibbs measure $\pi_\beta^{\text{SGLD}}$ (left). **Randomized smoothing** transforms this into a tractable **surrogate objective**, $g_\epsilon(\theta)$, which is the basis for the fSGLD algorithm and its invariant measure, $\pi_\beta^{\text{fSGLD}}$ (center). This highlights a key distinction: while the Gibbs measure of standard SGLD, $\pi_\beta^{\text{SGLD}}$, is indifferent to the flatness of the minima, the fSGLD framework is designed such that its invariant measure, $\pi_\beta^{\text{fSGLD}}$, targets the distribution over the flattest minima. Our ultimate goal is to target the **Hessian-trace regularized objective** $v(\theta)$ and its corresponding measure $\pi_{\beta,\sigma}^\star$, which concentrates on the desired global flat minima (right).

## 3 THEORETICAL RESULTS

In this section, we present the main theoretical results that rigorously validate the fSGLD algorithm. We begin by stating the formal assumptions for our analysis. We then prove that the invariant measure of fSGLD converges to an ideal target distribution over flat minima when its key hyper-parameters $\beta$ and $\sigma$ are properly coupled. Building on this, we derive non-asymptotic convergence guarantees for the fSGLD iterates in both Wasserstein distance and for the excess risk. The logical flow of our theoretical framework is summarized in the schematic illustration in Figure 1.

### 3.1 ASSUMPTIONS

We first state the formal assumptions for our main theoretical results. Specifically, our assumptions impose standard conditions on: (i) moments of the initial parameters, stochastic gradient, and the noise processes; (ii) a Lipschitz condition on the stochastic gradient; and (iii) a dissipativity condition to ensure the stability of the Langevin dynamics.

**Assumption 1** (Moments of the initial parameter, stochastic gradient, and independence of the data and noise perturbation)**.** *We assume the initial parameter $\theta_0$ has a finite fourth moment, $\mathbb{E}[|\theta_0|^4] < \infty$, and that we have access to an unbiased stochastic gradient for the original objective $u$, $\mathbb{E}[\nabla_\theta U(\theta, X)] = h(\theta)$, where the data sequence $(X_k)_{k \in \mathbb{N}}$ is i.i.d. Furthermore, the perturbation noise $(\epsilon_k)_{k \in \mathbb{N}} \sim \mathcal{N}(0, \sigma^2 I_d)$ with $\sigma \in (0, 1)$, $(X_k)_{k \in \mathbb{N}}$, $(\xi_k)_{k \in \mathbb{N}} \sim \mathcal{N}(0, I_d)$, and $\theta_0$ are mutually independent.*

We discuss how the sampling model in Assumption 1 relates to the sampling scheme used in our numerical experiments (Section 4) in Remark B.4.

**Assumption 2** (Lipschitzness)**.** *There exists $\varphi : \mathbb{R}^m \to [1, \infty)$ with $\mathbb{E}[|(1 + |X_0|)\varphi(X_0)|^4] < \infty$, and constants $L_1, L_2 > 0$ such that, for all $x, x' \in \mathbb{R}^m$ and $\theta, \theta' \in \mathbb{R}^d$,*

$$|\nabla_\theta U(\theta, x) - \nabla_{\theta'} U(\theta', x)| \le L_1 \varphi(x) |\theta - \theta'|,$$
$$|\nabla_\theta U(\theta, x) - \nabla_\theta U(\theta, x')| \le L_2 (\varphi(x) + \varphi(x'))(1 + |\theta|)|x - x'|,$$

**Assumption 3** (Dissipativity)**.** *There exist a measurable function (symmetric matrix-valued) function $A : \mathbb{R}^m \to \mathbb{R}^{d \times d}$ and a measurable function $\hat{b} : \mathbb{R}^m \to \mathbb{R}$ such that for any $x \in \mathbb{R}^m$, $y \in \mathbb{R}^d$, $\langle y, A(x)y \rangle \ge 0$ and for all $\theta \in \mathbb{R}^d$ and $x \in \mathbb{R}^m$,*

$$\langle \nabla_\theta U(\theta, x), \theta \rangle \ge \langle \theta, A(x)\theta \rangle - \hat{b}(x).$$

*The smallest eigenvalue of $\mathbb{E}[A(X_0)]$ is a positive real number $\bar{a} > 0$ and $\mathbb{E}[\hat{b}(X_0)] = \bar{b} > 0$.*

Note that this dissipativity condition is a standard requirement for analysis of SGLD in the literature; e.g., see Raginsky et al. (2017); Xu et al. (2018); Deng et al. (2020a;b; 2022); Futami & Fujisawa (2023). In particular, our version in Assumption 3 follows the more general formulation of Zhang et al. (2023), which allows for dependency on the data $X$. Moreover, several direct consequences of these assumptions, which are useful for our subsequent analysis, are detailed in Appendix B.

### 3.2 TARGET GIBBS MEASURE FOR GLOBAL FLAT MINIMA

Our analysis begins by defining the ideal target distribution which concentrates on the global flat minima. The natural choice is the Gibbs measure associated with $v$, which we define as $\pi_{\beta,\sigma}^\star$:

$$\pi_{\beta,\sigma}^\star(\mathrm{d}\theta) \propto \exp(-\beta v(\theta))\mathrm{d}\theta. \tag{9}$$

By construction, as the inverse temperature $\beta \to \infty$, this measure concentrates on the global flat minima.

The central question is whether the invariant measure of fSGLD, $\pi_\beta^{\text{fSGLD}}$, converges to this ideal Gibbs measure $\pi_{\beta,\sigma}^\star$. For these two Gibbs measures to align, the remainder term $\mathbb{E}[\mathcal{R}(\theta,\epsilon)]$ in 6 must be negligible. In high-dimensional nonconvex problems, this is a non-trivial condition, as higher-order terms can be substantial and unpredictable, potentially corrupting the intended regularization effect. For this reason, in the low-temperature limit ($\beta \to \infty$), a careful interplay between the inverse temperature $\beta$ and the noise scale $\sigma$ becomes essential. Please refer to Appendix A for the formal relationship between two Gibbs measures $\pi_\beta^{\text{fSGLD}}$ and $\pi_{\beta,\sigma}^\star$.

The following proposition shows that when the perturbation scale $\sigma$ and inverse temperature $\beta$ are properly coupled, the invariant measure of fSGLD converges to the ideal target measure in the Wasserstein distance of order two.

**Proposition 3.1.** *Let Assumptions 1, 2, and 3 hold, and let $\sigma = \beta^{-\frac{1+\eta}{4}}$ for $\eta \in (0,1)$. Then*

$$W_2(\pi_\beta^{\text{fSGLD}}, \pi_{\beta,\sigma}^\star) \leq \underline{D},$$

*where $\underline{D} = O(\beta^{-\frac{\eta}{2}} d \log d)$, whose explicit expression is given in 57. Moreover,*

$$\lim_{\beta \to \infty} W_2(\pi_\beta^{\text{fSGLD}}, \pi_{\beta,\sigma}^\star) = 0. \tag{10}$$

The proof of Proposition 3.1 is postponed to Appendix C.2. This proposition rigorously shows how RWP induces the desired Hessian-trace regularization effect through a theoretically-prescribed coupling of the two key hyperparameters, $\sigma$ and $\beta$. As demonstrated in our experiments, this coupling yields meaningful improvements in generalization.

### 3.3 CONVERGENCE GUARANTEES FOR FSGLD

Having established that fSGLD correctly targets the ideal distribution for flat minima, our first main result provides non-asymptotic error bounds on the Wasserstein-1 and -2 distances between the law of the $k$-th fSGLD iterate $\mathcal{L}(\theta_k^{\text{fSGLD}})$ and the target Gibbs measure $\pi_{\beta,\sigma}^\star$. All proofs for the results in this section are provided in Appendix C.2.

**Theorem 3.2.** *Let Assumptions 1, 2, and 3 hold, and let $\sigma = \beta^{-\frac{1+\eta}{4}}$ for $\eta \in (0,1)$. Then, there exist constants $\dot{c}, D_1, D_2, D_3, \underline{D} > 0$ such that, for every $\beta > 0$, for $0 < \lambda \leq \lambda_{max}$ with $\lambda_{max}$ given in 24, and $k \in \mathbb{N}$,*

$$W_1(\mathcal{L}(\theta_k^{\text{fSGLD}}), \pi_{\beta,\sigma}^\star) \leq D_1 e^{-\dot{c}\lambda k/2}(1 + \mathbb{E}[|\theta_0|^4]) + (D_2 + D_3)\sqrt{\lambda} + \underline{D}, \tag{11}$$

*where $\dot{c}$ is given in Lemma C.6, and*

$$D_1 = O\left(e^{D_\star(1+d/\beta)(1+\beta)}\left(1 + \frac{1}{1 - e^{-\dot{c}}}\right)\right), \quad \text{with } D_\star > 0 \text{ independent of } d, \beta, k,$$

$$D_2 = O\left(1 + \sqrt{\frac{d}{\beta}}\right), \quad D_3 = O\left(e^{D_\star(1+d/\beta)(1+\beta)}\left(1 + \frac{1}{1 - e^{-\dot{c}}}\right)\right), \quad \underline{D} = O(\beta^{-\frac{\eta}{2}} d \log d).$$

*The explicit expressions of $D_1$, $D_2$, $D_3$ are given in 72, and $\underline{D}$ is given in 57. Furthermore, let $\beta_{\bar{\delta}}$, $\lambda_{\bar{\delta}}$, $k_{\bar{\delta}}$ be as in 76, 77, and 78 respectively. For any $\bar{\delta} > 0$, if we choose $\beta \geq \beta_{\bar{\delta}}$, $\lambda \leq \lambda_{\bar{\delta}}$, and $k \geq k_{\bar{\delta}}$, then*

$$W_1(\mathcal{L}(\theta_k^{fSGLD}), \pi_{\beta,\sigma}^\star) \leq \bar{\delta}.$$

**Corollary 3.3.** *Let Assumption 1, 2 and 3 hold, and let $\sigma = \beta^{-\frac{1+\eta}{4}}$ for $\eta \in (0, 1)$. Then, there exists constants $\dot{c}, D_4, D_5, D_6, \underline{D} > 0$ such that, for every $\beta > 0$, $0 < \lambda \leq \lambda_{max}$ with $\lambda_{max}$ given in 24, and $k \in \mathbb{N}$,*

$$W_2(\mathcal{L}(\theta_k^{fSGLD}), \pi_{\beta,\sigma}^\star) \leq D_4 e^{-\dot{c}\lambda k/4}(\mathbb{E}[|\theta_0|^4] + 1) + (D_5 + D_6)\lambda^{1/4} + \underline{D}, \qquad (12)$$

*where $\dot{c}$ is given in Lemma C.6, and,*

$$D_4 = O\left(e^{D_\star(1+d/\beta)(1+\beta)}\left(1 + \frac{1}{1 - e^{-\dot{c}/2}}\right)\right), \quad \text{with } D_\star > 0 \text{ independent of } d, \beta, k,$$

$$D_5 = O\left(1 + \sqrt{\frac{d}{\beta}}\right), \quad D_6 = O\left(e^{D_\star(1+d/\beta)(1+\beta)}\left(1 + \frac{1}{1 - e^{-\dot{c}/2}}\right)\right), \quad \underline{D} = O(\beta^{-\frac{\eta}{2}}d\log d).$$

*The explicit expressions of $D_4$, $D_5$, $D_6$ are given in 74, and $\underline{D}$ is given in 57. In addition, let $\beta_{\widetilde{\delta}}$, $\lambda_{\widetilde{\delta}}$, $k_{\widetilde{\delta}}$ be as in 80, 81, and 82 respectively. For any $\widetilde{\delta} > 0$, if we choose $\beta \geq \beta_{\widetilde{\delta}}$, $\lambda \leq \lambda_{\widetilde{\delta}}$, and $k \geq k_{\widetilde{\delta}}$, then*

$$W_2(\mathcal{L}(\theta_k^{fSGLD}), \pi_{\beta,\sigma}^\star) \leq \widetilde{\delta}.$$

**Remark 3.4.** *We emphasize that Theorem 3.2 and Corollary 3.3 recover the best known convergence results for SGLD under comparable assumptions, see e.g. Zhang et al. (2023). Unfortunately, the constants $D_1$, $D_3$, $D_4$, $D_6$ have exponential dependence on d and $\beta$ due to the coupling arguments of Eberle et al. (2019), commonly used in the SGLD literature (Chau et al., 2021; Zhang et al., 2023). In this setting, any improvement in the dimension dependence would necessitate substantially strengthening the contraction-rate estimates in Eberle et al. (2019, Theorem 2.2).*

**Remark 3.5.** *The proofs of Theorem 3.2 and Corollary 3.3 rely on the following decomposition:*

$$W_p(\mathcal{L}(\theta_k^{fSGLD}), \pi_{\beta,\sigma}^\star) \leq W_p(\mathcal{L}(\theta_k^{fSGLD}), \mathcal{L}(Z_t^{\lambda,fSGLD})) + W_p(\mathcal{L}(Z_t^{\lambda,fSGLD}), \pi_\beta^{fSGLD})$$
$$+ W_p(\pi_\beta^{fSGLD}, \pi_{\beta,\sigma}^\star), \quad p = \{1, 2\}, \quad t \in (kT, (k+1)T]. \qquad (13)$$

*The first term on the right-hand side of 13 corresponds to the discretization error between the fSGLD recursion 7 and the time-rescaled version of flatness Langevin SDE 26 associated with the randomized-smoothing surrogate objective $g_\epsilon$ defined in 5. The second term captures the convergence error between this SDE and its invariant measure $\pi_\beta^{fSGLD}$. The third term is the distance between the two measures provided in Proposition 3.1. The proofs of the first two error terms follow the general structure of Chau et al. (2021); Zhang et al. (2023), but require substantial adaptation to handle the fSGLD update (instead of SGLD) as well as the surrogate objective function $g_\epsilon$ (instead of the original objective $u$). The proof of the third error term is entirely new.*

While the previous results guarantee convergence from a sampling perspective, our final result analyzes fSGLD as an optimizer. The following theorem provides a non-asymptotic bound on the expected excess risk with respect to the Hessian-trace regularized objective $v$.

**Theorem 3.6.** *Let Assumption 1, 2 and 3 hold, and let $\sigma = \beta^{-\frac{1+\eta}{4}}$ for $\eta \in (0, 1)$. Then, there exist constants $\dot{c}$, $D_1^\#$, $D_2^\#$, $D_3^\# > 0$ such that, for every $\beta > 0$, $0 < \lambda \leq \lambda_{max}$ with $\lambda_{max}$ given in 24, $k \in \mathbb{N}$,*

$$\mathbb{E}[g_\epsilon(\theta_k^{fSGLD})] - \inf_{\theta \in \mathbb{R}^d} v(\theta) \leq D_1^\# e^{-\dot{c}\lambda k/4} + D_2^\# \lambda^{1/4} + D_3^\#, \qquad (14)$$

*where $\dot{c}$ is given in Lemma C.6, and*

$$D_1^\# = O\left(e^{D_\star(1+d/\beta)(1+\beta)}\left(1 + \frac{1}{1 - e^{-\dot{c}/2}}\right)\right),$$

$$D_2^\# = O\left(e^{D_\star(1+d/\beta)(1+\beta)}\left(1 + \frac{1}{1 - e^{-\dot{c}/2}}\right)\right),$$

$$D_3^\# = O\left((d/\beta)\log(D_\star(\beta^{(1-\eta)/2} + 1))\right).$$

*The explicit expressions of $D_1^\#$ and $D_2^\#$ are given in 85, $D_3^\#$ is defined in 90. Moreover, let $\beta_\delta$, $\lambda_\delta$, $k_\delta$ be as in 91, 92, and 93 respectively. For any $\underline{\delta} > 0$, if we choose $\beta \geq \beta_\delta$, $\lambda \leq \lambda_\delta$, and $k \geq k_\delta$, then*

$$\mathbb{E}[g_\epsilon(\theta_k^{fSGLD})] - \inf_{\theta \in \mathbb{R}^d} v(\theta) \leq \underline{\delta}.$$

**Remark 3.7.** *In this special case when the perturbation scale $\sigma \to 0$, the fSGLD update reduces exactly to SGLD algorithm and the constants in the non-asymptotic bounds in Theorem 3.2, Corollary 3.3, and Theorem 3.6 coincide with the ones in (Zhang et al., 2023).*

This result provides a rigorous guarantee that fSGLD finds global flat minima by effectively solving the Hessian-trace regularized objective.

## 4 NUMERICAL EXPERIMENTS

### 4.1 EXPERIMENTAL SETUP

**Datasets.** We evaluate our method on three challenging noisy label datasets including CIFAR-10N and CIFAR-100N (Wei et al., 2022), and WebVision (Li et al., 2017). CIFAR-10N and CIFAR-100N include real-world annotation errors introduced by human annotators, offering realistic yet standardized benchmarks for noisy label learning. For CIFAR-10N, we use the aggregate noise setting. WebVision is a large-scale, in-the-wild benchmark, consisting of more than 2.4 million images with labels automatically collected from Google and Flickr based on the 1,000 ImageNet ILSVRC2012 categories. Following standard protocol Li et al. (2020); Ortego et al. (2021); Li et al. (2022), we use the first 50 classes from its Google image subset and report Top-1 (WV-1) and Top-5 (WV-5) accuracy on the official validation set.

**Models.** We use ResNet-34 and ResNet-50 for training from scratch. For fine-tuning experiments, we use the pre-trained ViT-B/16 (Dosovitskiy et al., 2021) architecture, which has been trained on the ImageNet-1K (Deng et al., 2009) dataset as the backbone on CIFAR-10N and CIFAR-100N.

**Baselines and Implementation Details.** We compare fSGLD against four baselines: SGD with momentum, AdamW (Loshchilov & Hutter, 2019), SGLD (Welling & Teh, 2011), and SAM (Foret et al., 2021). To ensure a fair comparison, all optimizer hyperparameters are tuned using Optuna (Akiba et al., 2019) with 20 trials of Bayesian optimization. For each optimizer, the search spaces were carefully chosen to include previously reported optimal hyperparameters from the literature, ensuring that all baselines are strongly tuned. For fSGLD, we search for the optimal noise scale $\sigma$, while the inverse temperature $\beta$ is determined by our theoretically-prescribed coupling, $\beta = \sigma^{-4/(1+\eta)}$ with $\eta = 0.01$. For experiments with training from scratch, all experiments are trained for 150 epochs with a batch size of 128. The learning rate decays by a factor of 0.1 in the 50th and 100th epochs. For fine tuning, models are trained for 75 epochs with a batch size of 128, decaying the rate by a factor of 0.1 at the 50th epoch. The detailed hyperparameter search spaces for each optimizer and experimental settings are provided in Appendix D.1.

### 4.2 EMPIRICAL PERFORMANCE ON REAL-WORLD NOISY LABEL DATASETS

**Training from scratch.** We first evaluate the performance of all optimizers when training ResNet models from scratch. Table 1 presents the results across all dataset-architecture combinations. Our proposed method, fSGLD ($\beta$-$\sigma$ coupled), consistently achieves the best or second-best performance on every benchmark. Notably, on the CIFAR-100N dataset which presents significant challenges due to its higher noise ratio and larger number of classes, fSGLD significantly outperforms all baselines.

In terms of computational cost, the wall-clock time per iteration (s/iter) shows that fSGLD has a training speed comparable to standard optimizers like SGD, AdamW, and SGLD. In contrast, SAM incurs nearly double the computational overhead due to its min-max formulation requiring two gradient evaluations per step. This highlights a key advantage of our method: fSGLD matches or surpasses SAM's strong performance with a computational budget similar to standard SGD.

Table 1: Performance comparison on ResNet-34 and ResNet-50. Results are reported as mean±std over five different random seeds. Within each model block, the best result is **bold** and the second-best is underlined. WV-1/WV-5 denote Top-1/Top-5 accuracy on WebVision. The wall-clock time per iteration (s/iter) measured on CIFAR-10N for each model architecture.

| Model | Optimizer | CIFAR-10N | CIFAR-100N | WV-1 | WV-5 | (s/iter) |
|---|---|---|---|---|---|---|
| ResNet-34 | SGD | $89.31_{\pm0.84}$ | $58.47_{\pm0.20}$ | $71.87_{\pm0.44}$ | $89.33_{\pm0.30}$ | 22.0 |
| | AdamW | $89.25_{\pm0.66}$ | $56.77_{\pm0.47}$ | $68.69_{\pm0.32}$ | $87.01_{\pm0.24}$ | 22.5 |
| | SAM | $91.53_{\pm0.22}$ | $59.18_{\pm0.33}$ | $\underline{73.49}_{\pm0.36}$ | $\underline{90.32}_{\pm0.31}$ | 41.3 |
| | SGLD | $88.77_{\pm0.51}$ | $57.33_{\pm0.36}$ | $70.87_{\pm0.67}$ | $88.06_{\pm0.30}$ | 22.2 |
| | fSGLD ($\beta$-$\sigma$ coupled) | $\mathbf{91.72}_{\pm0.20}$ | $\mathbf{62.02}_{\pm0.29}$ | $\mathbf{73.55}_{\pm0.27}$ | $89.86_{\pm0.12}$ | 23.7 |
| | fSGLD ($\beta$ fixed) | $\underline{91.56}_{\pm0.19}$ | $\underline{61.55}_{\pm0.45}$ | $73.23_{\pm0.34}$ | $\mathbf{90.63}_{\pm0.38}$ | 23.7 |
| ResNet-50 | SGD | $89.41_{\pm0.26}$ | $57.52_{\pm0.17}$ | $71.11_{\pm0.59}$ | $88.31_{\pm0.40}$ | 31.9 |
| | AdamW | $89.26_{\pm0.31}$ | $57.28_{\pm0.90}$ | $69.92_{\pm0.67}$ | $87.97_{\pm0.34}$ | 32.3 |
| | SAM | $\underline{90.88}_{\pm0.49}$ | $59.01_{\pm0.60}$ | $72.52_{\pm0.46}$ | $89.53_{\pm0.44}$ | 60.7 |
| | SGLD | $88.89_{\pm0.40}$ | $56.90_{\pm0.65}$ | $69.43_{\pm0.40}$ | $87.17_{\pm0.22}$ | 32.1 |
| | fSGLD ($\beta$-$\sigma$ coupled) | $\mathbf{91.26}_{\pm0.08}$ | $\mathbf{62.08}_{\pm0.45}$ | $\mathbf{73.31}_{\pm0.50}$ | $\mathbf{90.07}_{\pm0.20}$ | 34.1 |
| | fSGLD ($\beta$ fixed) | $90.72_{\pm0.29}$ | $\underline{61.56}_{\pm1.08}$ | $\underline{72.87}_{\pm0.64}$ | $\underline{89.59}_{\pm0.41}$ | 34.1 |

**Fine-tuning.** We also evaluate performance in the fine-tuning setting, using a pre-trained ViT-B/16 model on CIFAR-10N and CIFAR-100N. The results are presented in Table 2. Our method, fSGLD ($\beta$-$\sigma$ coupled), consistently outperforms standard optimizers like SGD and SGLD, and achieves performance competitive with or superior to SAM at roughly half the computational overhead.

### 4.3 ABLATION STUDY: THE EFFECT OF THE $\beta$-$\sigma$ COUPLING

To empirically validate our theoretical claim, we examine the effect of the theoretically-prescribed $\beta$-$\sigma$ coupling. We compare fSGLD ($\beta$-$\sigma$ coupled) against fSGLD ($\beta$ fixed) which reflects a common heuristic of setting a large, fixed $\beta$ for optimization. The results, summarized in Table 1 and Table 2, show that the coupled version consistently outperforms the fixed version in all settings, with the single exception of the WV-5 metric on ResNet-34. This provides strong empirical evidence that our theoretically-prescribed coupling is crucial for improving performance.

Table 2: Fine-tuning performance comparison on ViT-B/16.

| Model | ViT-B/16 | | |
|---|---|---|---|
| **Dataset** | CIFAR-10N | CIFAR-100N | (s/epoch) |
| SGD | 94.64 | 71.80 | 343.2 |
| AdamW | 95.57 | 72.30 | 344.5 |
| SAM | **96.75** | 74.66 | 656.7 |
| SGLD | 94.13 | 71.36 | 344.8 |
| fSGLD ($\beta$ fixed) | 96.70 | $\underline{75.16}$ | 345.8 |
| fSGLD ($\beta$-$\sigma$ coupled) | $\underline{96.72}$ | **75.18** | 345.8 |

### 4.4 SENSITIVITY ANALYSIS

Table 3: Performance with respect to the number of random perturbations $n$ used in fSGLD.

| | CIFAR-10N | (s/epoch) |
|---|---|---|
| $n = 1$ | $91.72_{\pm0.18}$ | 23.7 |
| $n = 2$ | $91.57_{\pm0.18}$ | 41.8 |
| $n = 3$ | $91.79_{\pm0.17}$ | 60.4 |
| $n = 4$ | $92.04_{\pm0.13}$ | 78.5 |
| $n = 5$ | $91.83_{\pm0.19}$ | 97.0 |

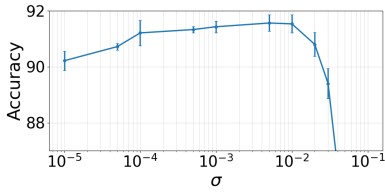

Figure 2: Sensitive analysis of noise standard deviation $\sigma$ on CIFAR-10N with ResNet-34.

While our fSGLD algorithm uses a single perturbation per iteration ($n = 1$), we examine how performance is affected by using multiple perturbations, which can provide a more accurate estimation of the Hessian trace. As shown in Table 3, increasing $n$ can improve accuracy, but this comes at a nearly linear increase in computational cost. Remarkably, fSGLD already achieves strong performance with just a single perturbation, making $n = 1$ a practical and efficient choice.

Next, we analyze the effect of the perturbation scale $\sigma$, as illustrated in Figure 2. The performance on CIFAR-10N remains stable and robust across a wide range of small to moderate values of $\sigma$. However, performance degrades sharply when $\sigma$ becomes excessively large, as the strong perturbations begin to destabilize the training process.

## 4.5 HESSIAN SPECTRUM

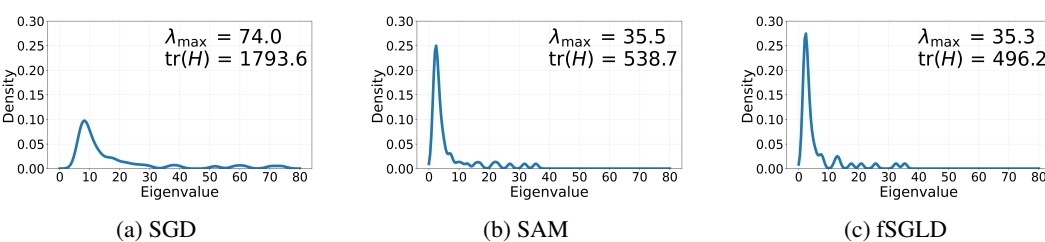

(a) SGD          (b) SAM          (c) fSGLD

Figure 3: The distribution of the leading eigenvalues and Hessian trace of ResNet-34 trained on CIFAR-10N with SGD, SAM, and fSGLD.

To empirically verify our theoretical insight that fSGLD finds flat minima by implicitly regularizing the Hessian trace, we analyze the curvature of the loss landscape at the solutions found by SGD, SAM, fSGLD. Note that we use the best hyperparameter configuration for each optimizer.

We compute two standard measures of sharpness for a ResNet-34 trained on CIFAR-10N: the maximum eigenvalue ($\lambda_{\max}$) of the Hessian and its trace ($\mathrm{tr}(H(\theta))$). Since exact computation is intractable, we estimate the top 50 eigenvalues using the Lanczos algorithm (Lin et al., 2016; Ghorbani et al., 2019) and approximate the trace with Hutchinson's method (Avron & Toledo, 2011; Ubaru et al., 2017). Detailed settings are described in Appendix D.2.

The results, presented in Figure 3, confirm our hypothesis. fSGLD converges to solutions with a significantly smaller maximum eigenvalue and Hessian trace compared to standard SGD. Remarkably, the degree of flatness achieved by fSGLD is comparable to SAM in terms of $\lambda_{\max}$ and even lower in terms of $\mathrm{tr}(H(\theta))$. This result is achieved at roughly half the computational cost of SAM. These results empirically validate our theoretical analysis, confirming that the proposed algorithm effectively promotes convergence to flatter minima.

## 5 RELATED WORK AND DISCUSSIONS

We review the most relevant literature on SAM, RWP, Hessian-based optimization, and SGLD.

**Flat Minima and Generalization.** Empirical studies (Keskar et al., 2017; Jastrzkebski et al., 2017; Jiang et al., 2020) and theoretical analyses (Dziugaite & Roy, 2017; Neyshabur et al., 2017) consistently show that flatter minima are strongly correlated with better generalization in deep neural networks. However, elucidating precise notions of sharpness and their relationship to generalization remains an open and active area of research (Andriushchenko & Flammarion, 2022; Ding et al., 2024; Wen et al., 2023; Tahmasebi et al., 2024).

**SAM and RWP.** The success of SAM (Foret et al., 2021) has produced a wide range of follow-up work to improve its efficiency, effectiveness, and applicability. Extensions include algorithmic improvements to approximate the inner maximization more efficiently (Liu et al., 2022a; Du et al., 2022a; Kwon et al., 2021; Xie et al., 2024; Li et al., 2024b; Chen et al., 2024; Kang et al., 2025). Beyond these, several Hessian-based regularization approaches have explored flatness from a different angle. For example, Zhang et al. (2024) propose Noise-Stability Optimization, and Li et al.

(2024a) studies random weight perturbation with explicit Hessian penalties. Both works focus on PAC-Bayes generalization bounds and local convergence to stationary points, providing algorithm-agnostic guarantees about the perturbed loss rather than the training dynamics of a specific optimizer. By contrast, we show that the invariant measure of fSGLD yields global, non-asymptotic convergence guarantees and an explicit link between random weight perturbation and Hessian-trace regularization. Lastly, the concept of using noise for regularization was formalized through the framework of randomized smoothing (Duchi et al., 2012), and our work makes this connection explicit for Langevin dynamics, differing fundamentally from explicit Hessian-penalty methods that rely on costly approximations (Sankar et al., 2021).

**SGLD and its Convergence Rate.** Following the seminal works of Welling & Teh (2011); Raginsky et al. (2017), numerous variants of SGLD have been developed to improve its practical performance, such as variance reduction techniques (Kinoshita & Suzuki, 2022; Dubey et al., 2016; Huang & Becker, 2021), preconditioned SGLD (Li et al., 2016), replica exchange SGLD (Dong & Tong, 2021; Deng et al., 2020a). A parallel line of research has focused on its theoretical properties, particularly its non-asymptotic convergence rate. Early results (Raginsky et al., 2017; Xu et al., 2018) showed convergence in the Wasserstein-2 distance at a rate dependent on the number of iterations. More recently, the state-of-the art analyses have established convergence rates of $O(\lambda^{1/2})$ in Wasserstein-1 and $O(\lambda^{1/4})$ in Wasserstein-2 distance (Zhang et al., 2023). Our convergence rates are consistent with these best-known results. However, a crucial distinction is that prior work proves convergence to the minimizers of the original objective $u$, whereas our guarantees are for convergence to global flat minima.

## 6 CONCLUSION AND LIMITATIONS

In this work, we introduced Flatness-Aware Stochastic Gradient Langevin Dynamics (fSGLD), a novel algorithm that synthesizes randomized smoothing with Langevin dynamics to efficiently target flat minima. By evaluating the gradient at parameters perturbed by Gaussian noise, a technique known as Random Weight Perturbation (RWP), fSGLD optimizes a surrogate objective that provably incorporates Hessian trace information without explicit computation.

Our main theoretical contribution is a rigorous non-asymptotic analysis of this process. We establish convergence guarantees in Wasserstein distance and provide the explicit excess risk bound for this class of flatness-aware optimizers. Crucially, our theory shows that the desired regularization effect emerges from a precise coupling of the noise scale $\sigma$ and the inverse temperature $\beta$.

Empirically, fSGLD demonstrates superior or competitive performance against strong baselines, including SAM, on challenging noisy-label and fine-tuning benchmarks. These gains are achieved at a computational cost comparable to standard SGD, roughly half that of SAM. Our analysis of the Hessian spectrum further confirms that fSGLD converges to significantly flatter minima, providing a direct validation of its mechanism. Ultimately, our work provides one of the provable links between an efficient algorithmic design (RWP within SGLD) and quantifiable generalization benefits, bridging the gap between heuristic flatness-seeking methods and rigorous convergence theory.

**Limitations and Future Directions.** Applying fSGLD to diffusion-based generative models is a particularly promising direction; investigating whether its bias towards flatter regions of the loss landscape can lead to more diverse or higher-quality samples is a compelling open question. On the theoretical side, we leave for future work the extension of our analysis to the case where $u$ is semiconvex (i.e., its gradient is one-sided Lipschitz), rather than satisfying Assumption 2.

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

# A    RELATIONSHIP BETWEEN $\pi_{\beta,\sigma}^{\star}$ AND $\pi_{\beta}^{\text{fSGLD}}$

We derive the relationship between the target measure $\pi_{\beta,\sigma}^{\star}$ and the invariant measure $\pi_{\beta}^{\text{fSGLD}}$ of the fSGLD algorithm, which will be used to prove Proposition 3.1, Theorem 3.2, and Corollary 3.3. By Taylor's theorem, we obtain

$$u(\theta + \epsilon) = u(\theta) + \nabla u(\theta)^T \epsilon + \frac{1}{2}\epsilon^T H(\theta)\epsilon + \mathcal{R}(\theta, \epsilon), \tag{15}$$

where $\mathcal{R}(\theta, \epsilon)$ denotes the remainder term. Taking the expectation over $\epsilon \sim \mathcal{N}(0, \sigma^2 I_d)$ in 15, we have

$$\begin{aligned}
g_\epsilon(\theta) &= u(\theta) + \frac{1}{2}\mathbb{E}[\epsilon^T H(\theta)\epsilon] + \mathbb{E}[\mathcal{R}(\theta, \epsilon)] \\
&= u(\theta) + \frac{1}{2}\text{tr}\left(H(\theta) \cdot \mathbb{E}[\epsilon^T \epsilon]\right) + \mathbb{E}[\mathcal{R}(\theta, \epsilon)] \\
&= v(\theta) + \mathbb{E}[\mathcal{R}(\theta, \epsilon)],
\end{aligned} \tag{16}$$

where

$$v(\theta) = u(\theta) + \frac{\sigma^2}{2}\text{tr}\left(H(\theta)\right),$$

and

$$\begin{aligned}
\mathbb{E}[\mathcal{R}(\theta, \epsilon)] &= \frac{1}{6}\sum_{i,j,k=1}^{d} \frac{\partial^3 u}{\partial\theta_i \partial\theta_j \partial\theta_k}(\theta)\, \mathbb{E}[\epsilon_i \epsilon_j \epsilon_k] + \frac{1}{24}\sum_{i,j,k,l=1}^{d} \frac{\partial^4 u}{\partial\theta_i \partial\theta_j \partial\theta_k \partial\theta_l}(\theta)\, \mathbb{E}[\epsilon_i \epsilon_j \epsilon_k \epsilon_l] \\
&\quad + \sum_{j=5}^{\infty} \frac{1}{j!}\sum_{i_1,i_2,\ldots,i_j=1}^{d} \frac{\partial^j u}{\partial\theta_{i_1}\partial\theta_{i_2}\ldots\partial\theta_{i_j}}(\theta)\, \mathbb{E}[\epsilon_{i_1}\epsilon_{i_2}\ldots\epsilon_{i_j}] \\
&= \frac{1}{24}\sum_{i,j,k,l=1}^{d} \frac{\partial^4 u}{\partial\theta_i \partial\theta_j \partial\theta_k \partial\theta_l}(\theta)\, \sigma^4(\delta_{ij}\delta_{kl} + \delta_{ik}\delta_{jl} + \delta_{il}\delta_{jk}) \\
&\quad + \sum_{j=6}^{\infty} \frac{1}{j!}\sum_{i_1,i_2,\ldots,i_j=1}^{d} \frac{\partial^j u}{\partial\theta_{i_1}\partial\theta_{i_2}\ldots\partial\theta_{i_j}}(\theta)\, \mathbb{E}[\epsilon_{i_1}\epsilon_{i_2}\ldots\epsilon_{i_j}],
\end{aligned} \tag{17}$$

where $\delta_{ij}$ denotes the Kronecker delta. Since $\sigma \in (0,1)$ by Assumption 1, we have $\mathbb{E}[\mathcal{R}(\theta, \epsilon)] = O(\sigma^4)$. Let the normalization constant of $\pi_{\beta}^{\text{fSGLD}}$ be given by

$$Z_\beta := \int_{\mathbb{R}^d} e^{-\beta g_\epsilon(\theta)}\, \mathrm{d}\theta, \tag{18}$$

and let the normalization constant of $\pi_{\beta,\sigma}^{\star}$ be given by

$$Z_{\beta,\sigma} := \int_{\mathbb{R}^d} e^{-\beta v(\theta)}\, \mathrm{d}\theta. \tag{19}$$

Using 16, 18, and 19, we obtain

$$\begin{aligned}
\pi_{\beta}^{\text{fSGLD}}(\mathrm{d}\theta) &= Z_\beta^{-1}\exp(-\beta g_\epsilon(\theta))\, \mathrm{d}\theta \\
&= Z_\beta^{-1} Z_{\beta,\sigma}\exp(-\beta\, \mathbb{E}[\mathcal{R}(\theta, \epsilon)])\, \pi_{\beta,\sigma}^{\star}(\mathrm{d}\theta).
\end{aligned} \tag{20}$$

# B    ADDITIONAL RESULTS FOR SECTION 3.1

This section collects several technical remarks and direct consequences of the assumptions presented in Section 3.1.

**Remark B.1.** *By Assumption 1 and 2, the gradient $h(\theta) = \mathbb{E}[\nabla_\theta U(\theta, X)]$ for all $\theta \in \mathbb{R}^d$, is well-defined. In addition, one obtains for all $\theta, \theta' \in \mathbb{R}^d$,*

$$|h(\theta) - h(\theta')| \leq L_1 \mathbb{E}[\varphi(X_0)]|\theta - \theta'|.$$

*As a consequence of Assumption 2, one obtains, for fixed $\widetilde{\epsilon} \in \mathbb{R}^d$,*

$$|\nabla_\theta U(\theta + \widetilde{\epsilon}, x) - \nabla_{\theta'} U(\theta' + \widetilde{\epsilon}, x)| \leq L_1 \varphi(x)|\theta - \theta'|,$$

$$|\nabla_\theta U(\theta + \widetilde{\epsilon}, x) - \nabla_\theta U(\theta + \widetilde{\epsilon}, x')| \leq L_2(\varphi(x) + \varphi(x'))(1 + |\theta + \widetilde{\epsilon}|)|x - x'|.$$

*Also, Assumption 2 implies*

$$|\nabla_\theta U(\theta + \widetilde{\epsilon}, x)| \leq L_1 \varphi(x)|\theta| + L_2 \bar{\varphi}(x)(1 + |\widetilde{\epsilon}|) + \widetilde{G}(\widetilde{\epsilon}),$$

*where $\bar{\varphi}(x) := (\varphi(x) + \varphi(0))|x|$, and $\widetilde{G}(\widetilde{\epsilon}) := |\nabla_{\theta'} U(\widetilde{\epsilon}, 0)|$. Moreover,*

$$|\nabla g_\epsilon(\theta)| \leq L_1 \mathbb{E}[\varphi(X_0)]|\theta| + L_2 \mathbb{E}[\bar{\varphi}(X_0)](1 + \mathbb{E}[|\epsilon|]) + \mathbb{E}[\widetilde{G}(\epsilon)].$$

**Remark B.2.** *By Assumption 1 and 3, one obtains a dissipativity condition of $h$, i.e., for any $\theta \in \mathbb{R}^d$, $\langle \nabla h(\theta), \theta \rangle \geq \bar{a}|\theta|^2 - \bar{b}$. Let $\zeta \in (0, \bar{a} L_1^{-2}(\mathbb{E}[\varphi^2(X_0)])^{-1})$. As a consequence of Assumptions 2 and 3, one obtains, for any $\theta \in \mathbb{R}^d$*

$$\langle \nabla g_\epsilon(\theta), \theta \rangle \geq a|\theta|^2 - b, \tag{21}$$

*where*

$$\begin{aligned}
a &:= \bar{a} - \zeta L_1^2 \mathbb{E}[\varphi^2(X_0)] > 0, \\
b &:= (2\zeta)^{-1}\sigma^2 d + 4\zeta L_2^2 \mathbb{E}[\bar{\varphi}^2(X_0)](1 + \sigma^2 d) + 2\zeta \mathbb{E}[\widetilde{G}^2(\epsilon)] + \bar{b} > 0,
\end{aligned} \tag{22}$$

*and $\widetilde{G}$ and $\bar{\varphi}$ are given in Remark B.1.*

*Proof of Remark B.2.* Using Assumption 3 and Remark B.1, and Young's inequality, one obtains, for fixed $\widetilde{\epsilon} \in \mathbb{R}^d$

$$\begin{aligned}
\langle \nabla_\theta U(\theta + \widetilde{\epsilon}, x), \theta \rangle &= \langle \nabla_\theta U(\theta + \widetilde{\epsilon}, x), \theta + \widetilde{\epsilon} \rangle - \langle \nabla_\theta U(\theta + \widetilde{\epsilon}, x), \widetilde{\epsilon} \rangle \\
&\geq \langle \theta + \widetilde{\epsilon}, A(x)\theta + \widetilde{\epsilon} \rangle - \hat{b}(x) - \zeta 2^{-1}|\nabla_\theta U(\theta + \widetilde{\epsilon}, x)|^2 - (2\zeta)^{-1}|\widetilde{\epsilon}|^2 \\
&\geq \langle \theta, (A(x) - \zeta L_1^2 \varphi^2(x))\theta \rangle + \langle \theta, A(x)\widetilde{\epsilon} \rangle + \langle \widetilde{\epsilon}, A(x)\theta \rangle + \langle \widetilde{\epsilon}, A(x)\widetilde{\epsilon} \rangle \\
&\quad - 4\zeta L_2^2 \bar{\varphi}^2(x)(1 + |\widetilde{\epsilon}|^2) - 2\zeta \widetilde{G}^2(\widetilde{\epsilon}) - \hat{b}(x) - (2\zeta)^{-1}|\widetilde{\epsilon}|^2.
\end{aligned} \tag{23}$$

Therefore,

$$\begin{aligned}
\nabla g_\epsilon(\theta) &= \mathbb{E}[\mathbb{E}_X[\nabla_\theta U(\theta + \epsilon, X)]] \\
&\geq (\bar{a} - \zeta L_1^2 \mathbb{E}[\varphi^2(X_0)])|\theta|^2 + (\bar{a} - (2\zeta)^{-1})\sigma^2 d - 4\zeta L_2^2 \mathbb{E}[\bar{\varphi}^2(X_0)](1 + \sigma^2 d) \\
&\quad - 2\zeta \mathbb{E}[\widetilde{G}^2(\epsilon)] - \bar{b} \\
&\geq a|\theta|^2 - b,
\end{aligned}$$

where $a$ and $b$ are defined in 22.

$\square$

**Lemma B.3.** *Let Assumption 2 and 3 hold. Then $\pi_{\beta,\sigma}^\star$ has finite second moments.*

*Proof of Lemma B.3.* As a consequence of Assumption 2, $\nabla v(\theta)$ is Lipschitz continuous. Let $\bar{\zeta} \in (0, 4\bar{a}\sigma^{-2})$. Using Assumption 2, Assumption 3, and Young's inequality, one obtains

$$\begin{aligned}
\langle \nabla v(\theta), \theta \rangle &= \langle \nabla u(\theta), \theta \rangle + \frac{\sigma^2}{2} \langle \nabla (\mathrm{tr}(H(\theta))), \theta \rangle \\
&\geq \left( \bar{a} - \frac{\bar{\zeta}\sigma^2}{4} \right)|\theta|^2 - \bar{b} - \frac{\sigma^2}{4\bar{\zeta}}|\nabla(\mathrm{tr}(H(\theta)))|^2,
\end{aligned}$$

which implies that $\nabla v(\theta)$ is dissipative. Therefore, $\pi_{\beta,\sigma}^\star$ has finite second moment. $\square$

**Remark B.4.** *Our theoretical results are established under the standard i.i.d. sampling model (Assumption 1) where a sample $X_k$ is drawn at each iteration. In contrast, our experiments in Section 4 follow the common practice in deep learning, training over multiple epochs using uniform sampling without replacement. These two sampling regimes are not identical because sampling*

*without replacement is not strictly i.i.d. However, it is well established in stochastic optimization that both regimes induce gradient noise with very similar statistical properties, particularly in large-dataset settings. Sampling with replacement would align exactly with our theoretical setup. In application domains where sample generators are available, such as financial modeling Chu & Tangpi (2024), the online sampling arises naturally. Since our empirical evaluation focuses on settings that are standard in the deep learning community, we adopt the widely used multi-pass training protocol for benchmarking.*

**Remark B.5.** *Controlling the remainder term $\mathbb{E}[\mathcal{R}(\theta, \epsilon)]$ in 17 could in principle require very strong smoothness assumptions such as globally bounded fourth-order derivatives to ensure uniform control of higher-order terms. These are not standard in SGLD analyses, and our approach does not impose any such extra conditions, Instead, by leveraging only the dissipativity conditon (Assumption 3) together with local Lipschitz continuity (Assumption 2), we establish all convergence results without any global $C^4$ boundedness or similar strong regularity. This distinction highlights a key theoretical contribution of our work: rigorous non-asymptotic analysis for nonconvex high-dimensional objectives under significantly weaker and more realistic assumptions.*

# C  OVERVIEW OF THE NON-ASYMPTOTIC WASSERSTEIN ANALYSIS AND ERROR BOUND FOR THE EXPECTED EXCESS RISK

In this section, we derive the results introduced in Sections 3.2 and 3.3. We begin by presenting the framework behind these two sections.

The 'data' process $(X_k)_{k \in \mathbb{N}}$ in 7 is adapted to a given filtration $(\mathcal{X}_k)_{k \in \mathbb{N}}$ representing the flow of past information, and we denote the sigma-algebra of $\cup_{k \in \mathbb{N}} \mathcal{X}_k$ by $\mathcal{X}_\infty$. In addition, we assume that $\theta_0$, $\mathcal{X}_\infty$, $(\epsilon_k)_{k \in \mathbb{N}}$, and $(\xi_k)_{k \in \mathbb{N}}$ are all independent among themselves.

We define

$$\lambda_{\max} := \min \left\{ \frac{\min\{a, a^{\frac{1}{3}}\}}{16(1 + L_1)^2 (\mathbb{E}[(1 + \varphi(X_0))^4])^{1/2}}, \frac{1}{a} \right\}, \tag{24}$$

where $L_1$, $\varphi$ and $a$ are defined in Assumptions 2 and Remark B.2, respectively.

## C.1  AUXILIARY PROCESSES

We start by defining the process $(Z_t^{\text{fSGLD}})_{t \in \mathbb{R}_+}$ as the solution of the *flatness* Langevin SDE

$$\begin{aligned} Z_0^{\text{fSGLD}} &:= \theta_0 \in \mathbb{R}^d, \\ \mathrm{d}Z_t^{\text{fSGLD}} &:= -\nabla g_\epsilon(Z_t^{\text{fSGLD}})\mathrm{d}t + \sqrt{2\beta^{-1}} \, \mathrm{d}B_t, \end{aligned} \tag{25}$$

where $B_t$ is a standard $d$-dimensional Brownian motion. Denote by $(\mathcal{F}_t)_{t \geq 0}$ the natural filtration of $(B_t)_{t \geq 0}$ and by $\Sigma_{\theta_0}$ the sigma-algebra generated by $\theta_0$, and we assume that $(\mathcal{F}_t)_{t \geq 0}$ is independent of $\mathcal{X}_\infty \vee \Sigma_{\theta_0}$. Furthermore, denote by $\mathcal{F}_\infty$ the sigma-algebra of $\bigcup_{t \geq 0} \mathcal{F}_t$.

**Remark C.1.** *By Remark B.1, SDE 25 has a unique solution adapted to $(\mathcal{F}_t)_{t \in \mathbb{R}_+}$.*

To facilitate the convergence analysis, we introduce the time-rescaled version of the process 25. For each $\lambda > 0$, $Z_t^{\lambda, \text{fSGLD}} := Z_{\lambda t}^{\text{fSGLD}}$, $t \in \mathbb{R}_+$, and let $\widetilde{B}_t^\lambda := B_{\lambda t}/\sqrt{\lambda}$, $t \geq 0$. We observe that $(\widetilde{B}_t)_{t \geq 0}$ is a Brownian motion and

$$\begin{aligned} Z_0^{\lambda, \text{fSGLD}} &:= \theta_0 \\ \mathrm{d}Z_t^{\lambda, \text{fSGLD}} &:= -\lambda \nabla g_\epsilon(Z_t^{\lambda, \text{fSGLD}}) \, \mathrm{d}t + \sqrt{2\lambda\beta^{-1}} \, \mathrm{d}\widetilde{B}_t^\lambda. \end{aligned} \tag{26}$$

The natural filtration of $(\widetilde{B}_t)_{t \geq 0}$ is denoted by $(\mathcal{F}_t^\lambda)_{t \geq 0}$ with $\mathcal{F}_t^\lambda := \mathcal{F}_{\lambda t}$, $t \in \mathbb{R}_+$. For a positive real number $a$, we denote its integer part by $\lfloor a \rfloor$. Then, we define $(\bar{\theta}_t^{\text{fSGLD}})_{t \in \mathbb{R}_+}$, the continuous-time interpolation of fSGLD 7, as

$$\begin{aligned} \bar{\theta}_0^{\text{fSGLD}} &:= \theta_0, \\ \mathrm{d}\bar{\theta}_t^{\text{fSGLD}} &:= -\lambda \nabla_\theta U(\bar{\theta}_{\lfloor t \rfloor}^{\text{fSGLD}} + \epsilon_{\lceil t \rceil}, X_{\lceil t \rceil}) \, \mathrm{d}t + \sqrt{2\lambda\beta^{-1}} \mathrm{d}\widetilde{B}_t. \end{aligned} \tag{27}$$

At grid-points, we note that the law of the interpolated process is the same as the law of the fSGLD algorithm 7, i.e. $\mathcal{L}(\bar{\theta}_k^{\text{fSGLD}}) = \mathcal{L}(\theta_k^{\text{fSGLD}})$, for each $k \in \mathbb{N}$. Moreover, we introduce the following continuous-time process $(\Phi_t^{s,u,\lambda,\text{fSGLD}})_{t \geq s}$, which is beneficial for our analysis, and define it as the solution of the following SDE

$$\Phi_s^{s,u,\lambda,\text{fSGLD}} := v \in \mathbb{R}^d$$

$$d\Phi_t^{s,u,\lambda,\text{fSGLD}} := -\lambda \nabla g_\epsilon(\Phi_t^{s,u,\lambda,\text{fSGLD}}) \, dt + \sqrt{2\lambda\beta^{-1}} \, d\widetilde{B}_t^\lambda.$$

**Definition C.2.** *Fix $k \in \mathbb{N}$. For any $t \geq kT$, define $\bar{\Phi}_t^{\lambda,k,\text{fSGLD}} := \Phi_t^{kT,\bar{\theta}_{kT}^{\text{fSGLD}},\lambda,\text{fSGLD}}$, where $T := \lfloor 1/\lambda \rfloor$.*

In other words, $\bar{\Phi}_t^{\lambda,k,\text{fSGLD}}$ in Definition C.2 is a process started from the value of the continuous-time interpolation fSGLD process 27 at time $kT$ and run until time $t \geq kT$ with the continuous-time flatness Langevin dynamics.

## C.2 PROOFS OF THE RESULTS IN SECTIONS 3.2 AND 3.3

To prove Proposition 3.1, Theorem 3.2, and Corollary 3.3, we will use the following results in Corollary C.3 and Lemma C.4 below.

Recall that for any $\mu$ and $\nu \in \mathcal{P}(\mathbb{R}^d)$, then Kullbak-Leibler divergence (or relative entropy) between $\mu$ and $\nu$ is defined as

$$\text{KL}(\mu||\nu) = \begin{cases} \int_{\mathbb{R}^d} \log\left(\frac{d\mu}{d\nu}\right) d\mu, & \text{if } \mu \ll \nu, \\ \infty, & \text{otherwise.} \end{cases} \tag{28}$$

**Corollary C.3.** *(Bolley & Villani, 2005, Corollary 2.3) For any two Borel probability measures $\mu$ and $\nu$ with finite second moments, one obtains*

$$W_2(\mu,\nu) \leq C_\nu \left[ \sqrt{KL(\mu||\nu)} + \left(\frac{KL(\mu||\nu)}{2}\right)^{1/4} \right],$$

*where*

$$C_\nu := 2 \inf_{\widetilde{\kappa}>0} \left( \frac{1}{\widetilde{\kappa}} \left( \frac{3}{2} + \log \int_{\mathbb{R}^d} e^{\widetilde{\kappa}|\theta|^2} \, \nu(d\theta) \right) \right)^{1/2}. \tag{29}$$

**Lemma C.4.** *Let Assumption 3 hold. Then, the following set*

$$A := \left\{ \theta \in \mathbb{R}^d : |\theta| \leq \sqrt{\frac{b}{a}} \right\}, \tag{30}$$

*contains all the minimizers of $u(\theta)$, $v(\theta)$, and $g_\epsilon(\theta)$, where $a$ and $b$ are given in 22.*

*Proof of Lemma C.4.* Let $\theta_{g_\epsilon}^\star$, $\theta_u^\star$, $\theta_v^\star$ be a minimizer of $g_\epsilon(\theta)$, $u(\theta)$, and $v(\theta)$, respectively. By Assumption 3, we have

$$0 = \langle \nabla v(\theta_v^*), \theta_v^\star \rangle = \langle \nabla v(\theta_u^*), \theta_u^\star \rangle = \langle \nabla u(\theta_u^\star), \theta_u^\star \rangle \geq \overline{a}|\theta_u^\star|^2 - \overline{b}, \tag{31}$$

which implies

$$|\theta_u^\star| = |\theta_v^\star| \leq \sqrt{\frac{\overline{b}}{\overline{a}}} \leq \sqrt{\frac{b}{a}}.$$

Due to Remark B.2, we have

$$0 = \langle \nabla g_\epsilon(\theta_{g_\epsilon}^\star), \theta_{g_\epsilon}^\star \rangle \geq a|\theta_{g_\epsilon}^\star|^2 - b, \tag{32}$$

which implies

$$|\theta_{g_\epsilon}^\star| \leq \sqrt{\frac{b}{a}}.$$

$\square$

*Proof of Proposition 3.1.* Using 20 with 17, we have

$$
\begin{aligned}
\mathrm{KL}(\pi_\beta^{\mathrm{fSGLD}} || \pi_{\beta,\sigma}^\star) &= \int_{\mathbb{R}^d} \log \left( \frac{\pi_\beta^{\mathrm{fSGLD}}(\mathrm{d}\theta)}{\pi_{\beta,\sigma}^\star(\mathrm{d}\theta)} \right) \pi_\beta^{\mathrm{fSGLD}}(\mathrm{d}\theta) \\
&= \int_{\mathbb{R}^d} \log \left( Z_\beta^{-1} Z_{\beta,\sigma} \exp(-\beta \, \mathbb{E}[\mathcal{R}(\theta,\epsilon)]) \right) \pi_\beta^{\mathrm{fSGLD}}(\mathrm{d}\theta) \qquad (33) \\
&= \log \left( \frac{Z_{\beta,\sigma}}{Z_\beta} \right) - \beta \int_{\mathbb{R}^d} \mathbb{E}[\mathcal{R}(\theta,\epsilon)] \, \pi_\beta^{\mathrm{fSGLD}}(\mathrm{d}\theta).
\end{aligned}
$$

We focus on the first term on the right-hand side of 33. We denote the complementary set of $A$ in Lemma C.4 by $A^c$. Using 18 and 19, one obtains

$$
\begin{aligned}
\log \left( \frac{Z_{\beta,\sigma}}{Z_\beta} \right) &= \log \left( \frac{\int_A e^{-\beta v(\theta)} d\theta + \int_{A^c} e^{-\beta v(\theta)} d\theta}{\int_A e^{-\beta g_\epsilon(\theta)} d\theta + \int_{A^c} e^{-\beta g_\epsilon(\theta)} d\theta} \right) \\
&= \log \left( \frac{\frac{\int_A e^{-\beta v(\theta)} d\theta}{\int_A e^{-\beta g_\epsilon(\theta)} d\theta} + \frac{\int_{A^c} e^{-\beta v(\theta)} d\theta}{\int_A e^{-\beta g_\epsilon(\theta)} d\theta}}{1 + \frac{\int_{A^c} e^{-\beta g_\epsilon(\theta)} d\theta}{\int_A e^{-\beta g_\epsilon(\theta)} d\theta}} \right). \qquad (34)
\end{aligned}
$$

We provide a bound on the first term of the numerator in 34, i.e., $\frac{\int_A e^{-\beta v(\theta)} d\theta}{\int_A e^{-\beta g_\epsilon(\theta)) d\theta}}$. By 17 and the extreme value theorem, there exists a constant $C_A > 0$:

$$
|g_\epsilon(\theta) - v(\theta)| \le C_A \sigma^4, \quad \forall\, \theta \in A, \qquad (35)
$$

where

$$
C_A = \max_{\theta \in A} \left( \sum_{i,j,k,l=1}^d \frac{\partial^4 u}{\partial \theta_i \partial \theta_j \partial \theta_k \partial \theta_l}(\theta) + \sum_{j=6}^\infty \sum_{i_1,i_2,\ldots,i_j=1}^d \frac{\partial^j u}{\partial \theta_{i_1} \partial \theta_{i_2} \ldots \partial \theta_{i_j}}(\theta) \right). \qquad (36)
$$

This leads to

$$
e^{-C_A \beta \sigma^4} \int_A e^{-\beta g_\epsilon(\theta)} \mathrm{d}\theta \le \int_A e^{-\beta v(\theta)} \mathrm{d}\theta \le e^{C_A \beta \sigma^4} \int_A e^{-\beta g_\epsilon(\theta)} \mathrm{d}\theta,
$$

which implies

$$
e^{-C_A \beta \sigma^4} \le \frac{\int_A e^{-\beta v(\theta)} \mathrm{d}\theta}{\int_A e^{-\beta g_\epsilon(\theta)} \mathrm{d}\theta} \le e^{C_A \beta \sigma^4}. \qquad (37)
$$

We provide a bound on the second term of the numerator on the right-hand side of 34, i.e., $\frac{\int_{A^c} e^{-\beta v(\theta)} d\theta}{\int_A e^{-\beta g_\epsilon(\theta)} d\theta}$. We note that, for any $\theta \in A^c$, there exists $\delta_v > 0$ such that

$$
v(\theta) > v(\theta_v^*) + \delta_v, \quad \text{for } \theta \in A^c. \qquad (38)
$$

For $0 < \beta_0 < \beta$, we have

$$
-(\beta - \beta_0) v(\theta) < -(\beta - \beta_0)(v(\theta_v^*) + \delta_v),
$$

which implies

$$
\int_{A^c} e^{-\beta v(\theta)} \mathrm{d}\theta \le e^{-(\beta-\beta_0)(v(\theta_v^*) + \delta_v)} \int_{A^c} e^{-\beta_0 v(\theta)} \mathrm{d}\theta. \qquad (39)
$$

By 17 and the extreme value theorem, we obtain

$$
\exp(-\beta \, \mathbb{E}[\mathcal{R}(\theta,\epsilon)]) \ge \exp(-\beta C_A \sigma^4), \qquad \theta \in A. \qquad (40)
$$

Using 40, we have

$$
\begin{aligned}
\int_A e^{-\beta g_\epsilon(\theta)} \mathrm{d}\theta &\ge \exp(-\beta C_A \sigma^4) \int_A e^{-\beta v(\theta)} \, \mathrm{d}\theta \\
&= \exp(-\beta C_A \sigma^4) \left( \frac{\pi^{\frac{d}{2}}}{\Gamma(\frac{d}{2}+1)} \left( \frac{b}{a} \right)^{\frac{d}{2}} + \int_A \sum_{i=1}^\infty \frac{(-\beta v(\theta))^i}{i!} \mathrm{d}\theta \right) \qquad (41) \\
&\ge \frac{(1 + \bar{c}_A(\beta))}{\exp(\beta C_A \sigma^4) \Gamma(\frac{d}{2}+1)},
\end{aligned}
$$

where $\bar{c}_A(\beta) := \min_{\theta \in A} \sum_{i=1}^{\infty} \frac{(-\beta v(\theta))^i}{i!}$. Combining 39 and 41, we obtain

$$\frac{\int_{A^c} e^{-\beta v(\theta)} \mathrm{d}\theta}{\int_A e^{-\beta g_\epsilon(\theta)} \mathrm{d}\theta} \leq \frac{\exp(\beta C_A \sigma^4) \, \Gamma(\frac{d}{2}+1) \int_{A^c} e^{-\beta_0 v(\theta)} \mathrm{d}\theta}{e^{(\beta-\beta_0)(v(\theta_v^\star)+\delta_v)} \, (1+\bar{c}_A(\beta))}. \tag{42}$$

We provide a bound on the ratio in the denominator on the right-hand side of 34. By Taylor's theorem,

$$g_\epsilon(\theta) = g_\epsilon(\theta_{g_\epsilon}^\star) + \frac{1}{2}(\theta - \theta_{g_\epsilon}^\star)^T \nabla^2 g_\epsilon(\theta_{g_\epsilon}^\star)(\theta - \theta_{g_\epsilon}^\star) + R_2(\theta), \tag{43}$$

where $R_2(\theta)$ is the remainder term accounting for the residual error above second order. By the extreme value theorem, there exists $\overline{m}, \overline{M} > 0$ such that

$$\overline{m} \leq e^{-\beta R_2(\theta)} \leq \overline{M}, \quad \forall\, \theta \in A. \tag{44}$$

Using 43 and 44, one obtains

$$\int_A e^{-\beta g_\epsilon(\theta)} d\theta = e^{-\beta g_\epsilon(\theta_{g_\epsilon}^\star)} \int_A e^{-\frac{\beta}{2}(\theta - \theta_{g_\epsilon}^\star)^T \nabla^2 g_\epsilon(\theta_{g_\epsilon}^\star)(\theta - \theta_{g_\epsilon}^\star) - \beta R_2(\theta)} d\theta$$

$$\leq \overline{M} e^{-\beta g_\epsilon(\theta_{g_\epsilon}^\star)} \left(\frac{2\pi}{\beta}\right)^{d/2} \frac{1}{\sqrt{\det \nabla^2 g_\epsilon(\theta_{g_\epsilon}^\star)}}. \tag{45}$$

Using 45, it follows

$$1 + \frac{\int_{A^c} e^{-\beta g_\epsilon(\theta)} \mathrm{d}\theta}{\int_A e^{-\beta g_\epsilon(\theta)} \mathrm{d}\theta} \geq 1 + \frac{\int_{A^c} e^{-\beta g_\epsilon(\theta)} \mathrm{d}\theta}{\overline{M} e^{-\beta g_\epsilon(\theta_{g_\epsilon}^\star)} \frac{1}{\sqrt{\det \nabla^2 g_\epsilon(\theta_{g_\epsilon}^\star)}}} \geq 1.$$

Thus,

$$\frac{1}{1 + \frac{\int_{A^c} e^{-\beta g_\epsilon(\theta)} \mathrm{d}\theta}{\int_A e^{-\beta g_\epsilon(\theta)} \mathrm{d}\theta}} \leq \frac{1}{1 + \frac{\int_{A^c} e^{-\beta g_\epsilon(\theta)} \mathrm{d}\theta}{\overline{M} e^{-\beta g_\epsilon(\theta_{g_\epsilon}^\star)} \frac{1}{\sqrt{\det \nabla^2 g_\epsilon(\theta_{g_\epsilon}^\star)}}}} \leq 1. \tag{46}$$

Using 46, 42, and 37 in 34 yields

$$\log\left(\frac{Z_{\beta,\sigma}}{Z_\beta}\right) \leq \log\left(e^{C_A \beta \sigma^4} + \frac{\exp(\beta C_A \sigma^4) \, \Gamma(\frac{d}{2}+1) \int_{A^c} e^{-\beta_0 v(\theta)} d\theta}{e^{(\beta-\beta_0)(v(\theta_v^\star)+\delta_v)} \, (1+\bar{c}_A(\beta))}\right). \tag{47}$$

Since $\sigma^4 = \beta^{-(1+\eta)}$, then $\mathbb{E}[\mathcal{R}(\theta, \epsilon)] = O(\beta^{-(1+\eta)})$. Therefore, we can bound 33 using 47, so that

$$\mathrm{KL}(\pi_\beta^{\mathrm{fSGLD}} || \pi_{\beta,\sigma}^\star) \leq \log\left(e^{C_A \beta^{-\eta}} + \frac{\exp(C_A \beta^{-\eta}) \, \Gamma(\frac{d}{2}+1) \int_{A^c} e^{-\beta_0 v(\theta)} d\theta}{e^{(\beta-\beta_0)(v(\theta_v^\star)+\delta_v)} \, (1+\bar{c}_A(\beta))}\right)$$
$$- \beta \int_{\mathbb{R}^d} \mathbb{E}[\mathcal{R}(\theta, \epsilon)] \, \pi_\beta^{\mathrm{fSGLD}}(\theta) \, \mathrm{d}\theta =: C_1. \tag{48}$$

By the Stirling's formula, we have $\log(\Gamma(\frac{d}{2}+1))$ is $O(d \log d)$, and the right-hand side of 48 is $O(\beta^{-\eta} d \log d)$. Then, one obtains

$$\lim_{\beta \to \infty} \mathrm{KL}(\pi_\beta^{\mathrm{fSGLD}} || \pi_{\beta,\sigma}^\star) = 0. \tag{49}$$

We apply Corollary C.3 with $\widetilde{\kappa} = 1$, to prove the asymptotic convergence in Wasserstein distance of order two between $\pi_\beta^{\mathrm{fSGLD}}$ and $\pi_{\beta,\sigma}^\star$. First, we provide a bound on the constant $C_{\pi_{\beta,\sigma}^\star}$ in Corollary C.3 using $\log(x+y) \leq \log 2 + \max(\log(x), \log(y))$ for all $x, y > 0$

$$C_{\pi_{\beta,\sigma}^\star}^2$$

$$\leq 6 + 4 \log\left(\int_{\mathbb{R}^d} e^{|\theta|^2 - \beta u(\theta) - \frac{\beta \sigma^2}{2} \mathrm{tr}(H(\theta))} \, \mathrm{d}\theta\right)$$

$$\leq 6 + 4 \log 2 + 4 \max\left(\log\left(\int_A e^{|\theta|^2 - \beta u(\theta) - \frac{\beta \sigma^2}{2} \mathrm{tr}(H(\theta))} \, \mathrm{d}\theta\right), \log\left(\int_{A^c} e^{|\theta|^2 - \beta u(\theta) - \frac{\beta \sigma^2}{2} \mathrm{tr}(H(\theta))} \, \mathrm{d}\theta\right)\right). \tag{50}$$

From 22, recall that $b$ is $O(d\beta^{-\frac{1+\eta}{2}})$. We can control the first integral on the right-hand side of 50 using Remark C.4, i.e.

$$\log\left(\int_A e^{|\theta|^2 - \beta u(\theta) - \frac{\beta\sigma^2}{2}\operatorname{tr}(H(\theta))}\,\mathrm{d}\theta\right) \le \frac{b}{a} + \log(\widetilde{c}_{A,\beta}) + \frac{d}{2}\log\left(\frac{\pi b}{a}\right). \tag{51}$$

where $\widetilde{c}_{A,\beta} := \max_{\theta \in A} e^{-\beta u(\theta) - \frac{\beta\sigma^2}{2}\operatorname{tr}(H(\theta))}$. For $\theta \in A^c$ and $\widetilde{c} \in (0,1)$, we have, by Assumption 3,

$$
\begin{aligned}
u(\theta) &= u(\widetilde{c}\theta) + \int_{\widetilde{c}}^1 \langle \theta, \nabla u(t\theta) \rangle \,\mathrm{d}t \\
&\ge u(\theta_u^\star) + \int_{\widetilde{c}}^1 t^{-1}\langle t\theta, \nabla u(t\theta) \rangle \,\mathrm{d}t \\
&\ge u(\theta_u^\star) + \int_{\widetilde{c}}^1 t^{-1}(\bar{a}|t\theta|^2 - \bar{b}) \,\mathrm{d}t \\
&\ge \frac{\bar{a}(1 - \widetilde{c}^2)}{2}|\theta|^2 + \bar{b}\log\widetilde{c} + u(\theta_u^\star) \\
&= \bar{c}|\theta|^2 + \bar{p},
\end{aligned}
\tag{52}
$$

where $\bar{c} := \frac{\bar{a}(1-\widetilde{c}^2)}{2} > 0$, and $\bar{p} := \bar{b}\log\widetilde{c} + u(\theta_u^\star)$. For any $\theta \in A^c$, there exists $\delta_u > 0$ such that
$$u(\theta) > u(\theta_u^*) + \delta_u, \quad \text{for} \quad \theta \in A^c. \tag{53}$$
For any $\beta_0 \in (\frac{1}{\bar{c}}, \beta) = (\frac{2}{\bar{a}(1-\widetilde{c}^2)}, \beta)$, we have
$$-(\beta - \beta_0)u(\theta) < -(\beta - \beta_0)(u(\theta_u^*) + \delta_u), \quad \text{for} \quad \theta \in A^c. \tag{54}$$
Using 54, Assumption 3, 52, and $\beta_0 > \frac{1}{\bar{c}}$, one obtains

$$
\begin{aligned}
\log\left(\int_{A^c} e^{|\theta|^2 - \beta u(\theta) - \frac{\beta\sigma^2}{2}\operatorname{tr}(H(\theta))}\mathrm{d}\theta\right) &\le \log\left(e^{-(\beta-\beta_0)(u(\theta_u^*)+\delta_u) - \beta\sigma^2 d\bar{a}/2}\int_{A^c} e^{|\theta|^2 - \beta_0 u(\theta)}\mathrm{d}\theta\right) \\
&\le \log\left(e^{-(\beta-\beta_0)(u(\theta_u^*)+\delta_u) - \beta_0\bar{p} - \beta\sigma^2 d\bar{a}/2}\int_{A^c} e^{(1-\beta_0\bar{c})|\theta|^2}\mathrm{d}\theta\right) \\
&\le \log\left(e^{-(\beta-\beta_0)(u(\theta_u^*)+\delta_u) - \beta_0\bar{p} - \beta\sigma^2 d\bar{a}/2} + 2\right) + \frac{d}{2}\log\left(\frac{\pi}{\beta_0\bar{c}-1}\right).
\end{aligned}
\tag{55}
$$

Plugging 51 and 55 in 50 with $\sigma^4 = \beta^{-(1+\eta)}$, yields

$$
\begin{aligned}
C_{\pi_{\beta,\sigma}^\star}^2 \le\ & 6 + 4\log 2 \\
& + 4\max\left(\frac{b}{a} + \log(\widetilde{c}_{A,\beta}) + \frac{d}{2}\log\left(\frac{\pi b}{a}\right),\right. \\
& \left.\log\left(e^{-(\beta-\beta_0)(u(\theta_u^*)+\delta_u) - \beta_0\bar{p} - \beta^{\frac{1-\eta}{2}}d\bar{a}/2} + 2\right) + \frac{d}{2}\log\left(\frac{\pi}{\beta_0\bar{c}-1}\right)\right),
\end{aligned}
\tag{56}
$$

which is $O(d\log(d\beta^{-\frac{1+\eta}{2}}))$. Therefore, applying Corollary C.3 with 56 and 48, we obtain

$$
\begin{aligned}
& W_2(\pi_\beta^{\text{fSGLD}}, \pi_{\beta,\sigma}^\star) \\
& \le \left[6 + 4\log 2\right. \\
& \quad + 4\max\left(\frac{b}{a} + \log(\widetilde{c}_{A,\beta}) + \frac{d}{2}\log\left(\frac{\pi b}{a}\right),\right. \\
& \quad \left.\left.\log\left(e^{-(\beta-\beta_0)(u(\theta_u^*)+\delta_u) - \beta_0\bar{p} - \beta^{\frac{1-\eta}{2}}d\bar{a}/2} + 2\right) + \frac{d}{2}\log\left(\frac{\pi}{\beta_0\bar{c}-1}\right)\right)\right]^{\frac{1}{2}} \\
& \quad \times \left[\left(\sqrt{C_1} + 2^{-\frac{1}{4}}(C_1)^{1/4}\right)\right] =: \underline{D},
\end{aligned}
\tag{57}
$$

Therefore, 57 is $O(\beta^{-\frac{\eta}{2}} d \log d)$. Taking the limit for $\beta \to \infty$ in 57 and using 49, we arrive at

$$\lim_{\beta \to \infty} W_2(\pi_\beta^{\mathrm{fSGLD}}, \pi_{\beta,\sigma}^\star) = 0. \tag{58}$$

$\square$

We use the following triangle inequality to establish a non-asymptotic bound for $W_1(\mathcal{L}(\theta_k^{\mathrm{fSGLD}}), \pi_{\beta,\sigma}^\star)$:

$$
\begin{aligned}
W_1(\mathcal{L}(\theta_k^{\mathrm{fSGLD}}), \pi_{\beta,\sigma}^\star) &\leq W_1(\mathcal{L}(\bar{\theta}_t^{\mathrm{fSGLD}}), \mathcal{L}(\bar{\Phi}_t^{\lambda,k,\mathrm{fSGLD}})) + W_1(\mathcal{L}(\bar{\Phi}_t^{\lambda,k,\mathrm{fSGLD}}), \mathcal{L}(Z_t^{\lambda,\mathrm{fSGLD}})) \\
&\quad + W_1(\mathcal{L}(Z_t^{\lambda,\mathrm{fSGLD}}), \pi_\beta^{\mathrm{fSGLD}}) + W_1(\pi_\beta^{\mathrm{fSGLD}}, \pi_{\beta,\sigma}^\star).
\end{aligned} \tag{59}
$$

We control the four terms on the right-hand side of 59 separately. The bounds for the first three terms follow from Zhang et al. (2023), with Zhang et al. (2023, Assumptions 1) replaced by Assumptions 1, and using the Lipschitzness and dissipativity of $g_\epsilon$ established in Remark B.1 and Remark B.2. For completeness, we reproduce these proofs here to make the convergence analysis of fSGLD self-contained.

We define, for each $p \geq 1$, the Lyapunov function $\widetilde{V}_p$ by $\widetilde{V}_p(\theta) := (1 + |\theta|^2)^{p/2}$, $\theta \in \mathbb{R}^d$, and similarly $\widetilde{v}_p(\omega) := (1 + \omega^2)^{p/2}$, for any real $\omega \geq 0$. These functions are twice continuously differentiable and

$$\sup_\theta (|\nabla \widetilde{V}_p(\theta)|/\widetilde{V}_p(\theta)) < \infty, \qquad \lim_{|\theta| \to \infty} (|\nabla \widetilde{V}_p(\theta)|/\widetilde{V}_p(\theta)) = 0. \tag{60}$$

Let $\mathcal{P}_{\widetilde{V}_p}$ denote the set of $\mu \in \mathcal{P}(\mathbb{R}^d)$ satisfying $\int_{\mathbb{R}^d} \widetilde{V}_p(\theta) \, \mu(\mathrm{d}\theta) < \infty$. Then, we define a functional that plays a central role in establishing the convergence rate in the Wasserstein-1 distance. For $\mu, \nu \in \mathcal{P}_{\widetilde{V}_2}$, let

$$w_{1,2}(\mu, \nu) := \inf_{\Gamma \in \mathcal{C}(\mu,\nu)} \int_{\mathbb{R}^d} \int_{\mathbb{R}^d} [1 \wedge |\theta - \theta'|](1 + \widetilde{V}_2(\theta) + \widetilde{V}_2(\theta')) \, \Gamma(\mathrm{d}\theta, \mathrm{d}\theta'). \tag{61}$$

Moreover, it holds that $W_1(\mu, \nu) \leq w_{1,2}(\mu, \nu)$.

**Proposition C.5.** *Let Assumptions 1, 2, and 3 hold. Let $(\tilde{Z}_t^{fSGLD})_{t \in \mathbb{R}_+}$ be the solution of 25 with initial condition $\tilde{Z}_0^{fSGLD} = \tilde{\theta}_0$ which is independent of $\mathcal{F}_\infty$ and satisfies $\mathbb{E}[|\tilde{\theta}_0|^2] < \infty$. Then,*

$$w_{1,2}(\mathcal{L}(Z_t^{fSGLD}), \mathcal{L}(\tilde{Z}_t^{fSGLD})) \leq \hat{c} e^{-\dot{c}t} w_{1,2}(\mathcal{L}(\theta_0), \mathcal{L}(\tilde{\theta}_0)),$$

*where the constants $\dot{c}$ and $\hat{c}$ are given in Lemma C.6.*

*Proof.* From Remark B.1, one can deduce

$$
\begin{aligned}
|\nabla_\theta g_\epsilon(\theta) - \nabla_{\theta'} g_\epsilon(\theta')| &\leq \mathbb{E}[|\nabla_\theta u(\theta + \epsilon) - \nabla_{\theta'} u(\theta' + \epsilon)] \\
&\leq L_1 \mathbb{E}[\varphi(X_0)]|\theta - \theta'|.
\end{aligned} \tag{62}
$$

The rest of the proof follows using Assumption 1, 2, and 3, 62, Lemma C.15, and 60 in Zhang et al. (2023, Proof of Proposition 4.6). $\square$

The constants $\dot{c}$ and $\hat{c}$ from Proposition C.5 are given in an explicit form.

**Lemma C.6.** *The contraction constant $\dot{c} > 0$ in Proposition C.5 is given by*

$$\dot{c} := \min \left\{ \bar{\phi}, \bar{c}(2), 4\tilde{c}(2)\varepsilon\bar{c}(2)/2 \right\}/2 \tag{63}$$

*where $\bar{c}(2) = a/2$, $\tilde{c}(2) = (3/2)av_2(\bar{M}_2)$ with $\bar{M}_2$ given in Lemma C.15, $\bar{\phi}$ is given by*

$$\bar{\phi} := \left( \bar{r}\sqrt{8\pi/(\beta L_1 \mathbb{E}[\varphi(X_0)])} \exp\left( \left( \bar{r}\sqrt{\beta L_1 \mathbb{E}[\varphi(X_0)]/8} + \sqrt{8/(\beta L_1 \mathbb{E}[\varphi(X_0)])} \right)^2 \right) \right)^{-1}, \tag{64}$$

*and moreover, $\varepsilon > 0$ can be chosen such that the following inequality is satisfied*

$$\varepsilon \leq 1 \wedge \left( 4\tilde{c}(2)\sqrt{2\beta\pi/(L_1 \mathbb{E}[\varphi(X_0)])} \int_0^{\tilde{r}} \exp\left( s\sqrt{\beta L_1 \mathbb{E}[\varphi(X_0)]/8} + \sqrt{8/(\beta L_1 \mathbb{E}[\varphi(X_0)])} \right)^2 \mathrm{d}s \right)^{-1}, \tag{65}$$

*where $\tilde{r} := 2\sqrt{2\tilde{c}(2)/\bar{c}(2) - 1}$ and $\bar{r} := 2\sqrt{4\tilde{c}(2)(1 + \bar{c}(2))/\bar{c}(2) - 1}$. The constant $\hat{c} > 0$ is given by $\hat{c} := 2(1 + \bar{r}) \exp(\beta L_1 \mathbb{E}[\varphi(X_0)]\bar{r}^2/8 + 2\bar{r})/\varepsilon$.*

*Proof.* This follows by adapting the arguments of Zhang et al. (2023, Proof of Lemma 4.11) to the *flatness* Langevin SDE 25, using 62 together with Lemma C.15. □

From the definition of $\lambda_{\max}$ given in 24, it follows that $0 < \lambda \leq \lambda_{\max} \leq 1$, and hence $1/2 < \lambda T \leq 1$. We now proceed to bound the first term in 59.

**Lemma C.7.** *Let Assumptions 1, 2, and 3 hold. For any $0 < \lambda < \lambda_{max}$ given in 24, $t \in (kT, (k+1)T]$,*

$$W_2(\mathcal{L}(\bar{\theta}_t^{fSGLD}), \mathcal{L}(\bar{\Phi}_t^{\lambda,k,fSGLD})) \leq \sqrt{\lambda} \left( e^{-ak/4} \bar{D}_{2,1} \mathbb{E}[\widetilde{V}_2(\theta_0)] + \bar{D}_{2,2} \right)^{1/2},$$

*where*

$$
\begin{aligned}
\bar{D}_{2,1} &:= 4e^{4L_1^2 \mathbb{E}[\varphi^2(X_0)]}(L_1^2 \mathbb{E}[\varphi^2(X_0)]\bar{\psi}_Y + \bar{\psi}_Z), \\
\bar{D}_{2,2} &:= 4e^{4L_1^2 \mathbb{E}[\varphi^2(X_0)]}(L_1^2 \mathbb{E}[\varphi^2(X_0)]\widetilde{\psi}_Y + \widetilde{\psi}_Z),
\end{aligned}
\tag{66}
$$

*with $\bar{\psi}_Y$, $\widetilde{\psi}_Y$ given in 96, and $\bar{\psi}_Z$, $\widetilde{\psi}_Z$ given in 97.*

*Proof.* This follows by applying Lemma C.17 together with the argument used in Zhang et al. (2023, Proof of Lemma 4.7). We summarize the main steps in the following. Using synchronous coupling together with 27, Definition C.2, Remark B.1, and it follows that for any $t \in (kT, (k+1)T]$,

$$
\begin{aligned}
\left| \bar{\Phi}_t^{\lambda,k,fSGLD} - \bar{\theta}_t^{fSGLD} \right| &\leq \lambda \left| \int_{kT}^t \left[ \nabla_\theta U(\bar{\theta}_{\lfloor s \rfloor}^{fSGLD} + \epsilon_{\lceil s \rceil}, X_{\lceil s \rceil}) - \nabla g_\epsilon(\bar{\Phi}_s^{\lambda,k,fSGLD}) \right] \mathrm{d}s \right| \\
&\leq \lambda \left| \int_{kT}^t \left[ \nabla_\theta U(\bar{\theta}_{\lfloor s \rfloor}^{fSGLD} + \epsilon_{\lceil s \rceil}, X_{\lceil s \rceil}) - \nabla_\theta U(\bar{\Phi}_s^{\lambda,k,fSGLD} + \epsilon_{\lceil s \rceil}, X_{\lceil s \rceil}) \right] \mathrm{d}s \right| \\
&\quad + \lambda \left| \int_{kT}^t \left[ \nabla g_\epsilon(\bar{\Phi}_s^{\lambda,k,fSGLD}) - \nabla_\theta U(\bar{\Phi}_s^{\lambda,k,fSGLD} + \epsilon_{\lceil s \rceil}, X_{\lceil s \rceil}) \right] \mathrm{d}s \right| \\
&\leq \lambda L_1 \int_{kT}^t \varphi(X_{\lceil s \rceil}) \left| \bar{\theta}_{\lfloor s \rfloor}^{fSGLD} - \bar{\Phi}_s^{\lambda,k,fSGLD} \right| \mathrm{d}s \\
&\quad + \lambda \left| \int_{kT}^t \left[ \nabla g_\epsilon(\bar{\Phi}_s^{\lambda,k,fSGLD}) - \nabla_\theta U(\bar{\Phi}_s^{\lambda,k,fSGLD} + \epsilon_{\lceil s \rceil}, X_{\lceil s \rceil}) \right] \mathrm{d}s \right|.
\end{aligned}
\tag{67}
$$

Squaring both sides of 67 and taking expectations, we obtain using Assumption 1

$$
\begin{aligned}
\mathbb{E}\left[ \left| \bar{\Phi}_t^{\lambda,k,fSGLD} - \bar{\theta}_t^{fSGLD} \right|^2 \right] &\leq 2\lambda L_1^2 \int_{kT}^t \mathbb{E}\left[ \varphi^2(X_0) \right] \mathbb{E}\left[ \left| \bar{\theta}_{\lfloor s \rfloor}^{fSGLD} - \bar{\Phi}_s^{\lambda,k,fSGLD} \right|^2 \right] \mathrm{d}s \\
&\quad + 2\lambda^2 \mathbb{E}\left[ \left| \int_{kT}^t \left[ \nabla g_\epsilon(\bar{\Phi}_s^{\lambda,k,fSGLD}) - \nabla_\theta U(\bar{\Phi}_s^{\lambda,k,fSGLD} + \epsilon_{\lceil s \rceil}, X_{\lceil s \rceil}) \right] \mathrm{d}s \right|^2 \right].
\end{aligned}
$$

From $\lambda T \leq 1$ and Lemma C.17, we get

$$
\begin{aligned}
\mathbb{E}&\left[ \left| \bar{\Phi}_t^{\lambda,k,fSGLD} - \bar{\theta}_t^{fSGLD} \right|^2 \right] \\
&\leq 4\lambda L_1^2 \mathbb{E}\left[ \varphi^2(X_0) \right] \int_{kT}^t \mathbb{E}\left[ \left| \bar{\theta}_{\lfloor s \rfloor}^{fSGLD} - \bar{\theta}_s^{fSGLD} \right|^2 \right] \mathrm{d}s \\
&\quad + 4\lambda L_1^2 \mathbb{E}\left[ \varphi^2(X_0) \right] \int_{kT}^t \mathbb{E}\left[ \left| \bar{\theta}_s^{fSGLD} - \bar{\Phi}_s^{\lambda,k,fSGLD} \right|^2 \right] \mathrm{d}s \\
&\quad + 2\lambda^2 \mathbb{E}\left[ \left| \int_{kT}^t \left[ \nabla g_\epsilon(\bar{\Phi}_s^{\lambda,k,fSGLD}) - \nabla_\theta U(\bar{\Phi}_s^{\lambda,k,fSGLD} + \epsilon_{\lceil s \rceil}, X_{\lceil s \rceil}) \right] \mathrm{d}s \right|^2 \right] \\
&\leq 4\lambda L_1^2 \mathbb{E}\left[ \varphi^2(X_0) \right] (e^{-\lambda akT} \bar{\psi}_Y \mathbb{E}[\widetilde{V}_2(\theta_0)] + \widetilde{\psi}_Y) \\
&\quad + 4\lambda L_1^2 \mathbb{E}\left[ \varphi^2(X_0) \right] \int_{kT}^t \mathbb{E}\left[ \left| \bar{\theta}_s^{fSGLD} - \bar{\Phi}_s^{\lambda,k,fSGLD} \right|^2 \right] \mathrm{d}s \\
&\quad + 2\lambda^2 \mathbb{E}\left[ \left| \int_{kT}^t \left[ \nabla g_\epsilon(\bar{\Phi}_s^{\lambda,k,fSGLD}) - \nabla_\theta U(\bar{\Phi}_s^{\lambda,k,fSGLD} + \epsilon_{\lceil s \rceil}, X_{\lceil s \rceil}) \right] \mathrm{d}s \right|^2 \right],
\end{aligned}
\tag{68}
$$

We now bound the last term in 68 by splitting the final integral. Let $kT + N < t \le kT + N + 1$ with $N + 1 \le T, N \in \mathbb{N}$. It follows that

$$\left| \int_{kT}^{t} \left[ \nabla g_\epsilon(\bar{\Phi}_s^{\lambda,k,\text{fSGLD}}) - \nabla_\theta U(\bar{\Phi}_s^{\lambda,k,\text{fSGLD}} + \epsilon_{\lceil s \rceil}, X_{\lceil s \rceil}) \right] \mathrm{d}s \right| = \left| \sum_{n=1}^{N} I_n + R_N \right|,$$

where $I_n := \int_{kT+(n-1)}^{kT+n} [\nabla g_\epsilon(\bar{\Phi}_s^{\lambda,k,\text{fSGLD}}) - \nabla_\theta U(\bar{\Phi}_s^{\lambda,k,\text{fSGLD}} + \epsilon_{kT+n}, X_{kT+n})] \mathrm{d}s$, and $R_N := \int_{kT+N}^{t} [\nabla g_\epsilon(\bar{\Phi}_s^{\lambda,k,\text{fSGLD}}) - \nabla_\theta U(\bar{\Phi}_s^{\lambda,k,\text{fSGLD}} + \epsilon_{kT+N+1}, X_{kT+N+1})] \mathrm{d}s$. Squaring both sides, we obtain

$$\left| \sum_{n=1}^{N} I_n + R_N \right|^2 = \sum_{n=1}^{N} |I_n|^2 + 2 \sum_{n=2}^{N} \sum_{j=1}^{n-1} \langle I_n, I_j \rangle + 2 \sum_{n=1}^{N} \langle I_n, R_N \rangle + |R_N|^2.$$

Let $\mathcal{H}_\epsilon$ denote the sigma-algebra generated by $\epsilon$. We define the filtration $\mathcal{J}_t = \mathcal{F}_\infty^\lambda \vee \mathcal{X}_{\lfloor t \rfloor} \vee \mathcal{H}_{\lfloor \epsilon \rfloor}$ and we take expectations of both sides. Observe that for any $n = 2, \ldots, N$, $j = 1, \ldots, n-1$,

$$\mathbb{E}\left[ \langle I_n, I_j \rangle \right]$$
$$= \mathbb{E}\left[ \mathbb{E}[\langle I_n, I_j \rangle | \mathcal{J}_{kT+j}] \right],$$
$$= \mathbb{E}\left[ \mathbb{E}\left[ \left\langle \int_{kT+(n-1)}^{kT+n} [\nabla g_\epsilon(\bar{\Phi}_s^{\lambda,k,\text{fSGLD}}) - \nabla_\theta U(\bar{\Phi}_s^{\lambda,k,\text{fSGLD}} + \epsilon_{kT+n}, X_{kT+n})] \mathrm{d}s, \right. \right. \right.$$
$$\left. \left. \left. \int_{kT+(j-1)}^{kT+j} [\nabla g_\epsilon(\bar{\Phi}_s^{\lambda,k,\text{fSGLD}}) - \nabla_\theta U(\bar{\Phi}_s^{\lambda,k,\text{fSGLD}} + \epsilon_{kT+j}, X_{kT+j})] \mathrm{d}s \right\rangle \right| \mathcal{J}_{kT+j} \right] \right]$$
$$= \mathbb{E}\left[ \left\langle \int_{kT+(n-1)}^{kT+n} \mathbb{E}\left[ \nabla g_\epsilon(\bar{\Phi}_s^{\lambda,k,\text{fSGLD}}) - \nabla_\theta U(\bar{\Phi}_s^{\lambda,k,\text{fSGLD}} + \epsilon_{kT+n}, X_{kT+n}) \right| \mathcal{J}_{kT+j} \right] \mathrm{d}s, \right.$$
$$\left. \int_{kT+(j-1)}^{kT+j} [\nabla g_\epsilon(\bar{\Phi}_s^{\lambda,k,\text{fSGLD}}) - \nabla_\theta U(\bar{\Phi}_s^{\lambda,k,\text{fSGLD}} + \epsilon_{kT+j}, X_{kT+j})] \mathrm{d}s \right\rangle \right] = 0.$$

By the same reasoning, $\mathbb{E}\langle I_n, R_N \rangle = 0$ for all $1 \le n \le N$. Combining these results, we can bound the last term on the right-hand side of 68 using Lemma C.18

$$2\lambda^2 \mathbb{E}\left[ \left| \int_{kT}^{t} \left[ \nabla g_\epsilon(\bar{\Phi}_s^{\lambda,k,\text{fSGLD}}) - \nabla_\theta U(\bar{\Phi}_s^{\lambda,k,\text{fSGLD}} + \epsilon_{\lceil s \rceil}, X_{\lceil s \rceil}) \right] \mathrm{d}s \right|^2 \right]$$
$$= 2\lambda^2 \sum_{n=1}^{N} \mathbb{E}\left[ |I_n|^2 \right] + 2\lambda^2 \mathbb{E}\left[ |R_N|^2 \right]$$
$$\le 4e^{-a\lambda kT/2} \lambda (\bar{\psi}_Z \mathbb{E}[\widetilde{V}_2(\theta_0)] + \widetilde{\psi}_Z).$$

Consequently, 68 is bounded as follows

$$\mathbb{E}\left[ \left| \bar{\Phi}_t^{\lambda,k,\text{fSGLD}} - \bar{\theta}_t^{\text{fSGLD}} \right|^2 \right] \le 4\lambda L_1^2 \mathbb{E}\left[ \varphi^2(X_0) \right] \int_{kT}^{t} \mathbb{E}\left[ \left| \bar{\theta}_s^{\text{fSGLD}} - \bar{\Phi}_s^{\lambda,k,\text{fSGLD}} \right|^2 \right] \mathrm{d}s$$
$$+ 4e^{-a\lambda kT/2} \lambda (L_1^2 \mathbb{E}\left[ \varphi^2(X_0) \right] \bar{\psi}_Y + \bar{\psi}_Z) \mathbb{E}[\widetilde{V}_2(\theta_0)]$$
$$+ 4\lambda (L_1^2 \mathbb{E}\left[ \varphi^2(X_0) \right] \widetilde{\psi}_Y + \widetilde{\psi}_Z).$$

Applying Grönwall's inequality yields

$$\mathbb{E}\left[ \left| \bar{\Phi}_t^{\lambda,k,\text{fSGLD}} - \bar{\theta}_t^{\text{fSGLD}} \right|^2 \right] \le \lambda e^{4L_1^2 \mathbb{E}[\varphi^2(X_0)]} \left[ 4e^{-a\lambda kT/2} (L_1^2 \mathbb{E}\left[ \varphi^2(X_0) \right] \bar{\psi}_Y + \bar{\psi}_Z) \mathbb{E}[\widetilde{V}_2(\theta_0)] \right.$$
$$\left. + 4(L_1^2 \mathbb{E}\left[ \varphi^2(X_0) \right] \widetilde{\psi}_Y + \widetilde{\psi}_Z) \right].$$

Finally, we obtain using $\lambda T \ge 1/2$,

$$W_2^2(\mathcal{L}(\bar{\theta}_t^{\text{fSGLD}}), \mathcal{L}(\bar{\Phi}_t^{\lambda,k,\text{fSGLD}})) \le \mathbb{E}\left| \bar{\Phi}_t^{\lambda,k,\text{fSGLD}} - \bar{\theta}_t^{\text{fSGLD}} \right|^2$$
$$\le \lambda (e^{-an/4} \bar{C}_{2,1} \mathbb{E}[\widetilde{V}_2(\theta_0)] + \bar{C}_{2,2}),$$
(69)

where

$$\bar{D}_{2,1} := 4e^{4L_1^2 \mathbb{E}[\varphi^2(X_0)]} (L_1^2 \mathbb{E}\left[\varphi^2(X_0)\right] \bar{\psi}_Y + \bar{\psi}_Z),$$

$$\bar{D}_{2,2} := 4e^{4L_1^2 \mathbb{E}[\varphi^2(X_0)]} (L_1^2 \mathbb{E}\left[\varphi^2(X_0)\right] \widetilde{\psi}_Y + \widetilde{\psi}_Z).$$

$\square$

The bound for the second term on the right-hand side of 59 is established in the following lemma.

**Lemma C.8.** *Let Assumptions 1, 2, and 3 hold. For any $0 < \lambda < \lambda_{max}$ given in 24, $t \in (kT, (k+1)T]$,*

$$W_1(\mathcal{L}(\bar{\Phi}_t^{\lambda,k,fSGLD}), \mathcal{L}(Z_t^{\lambda,fSGLD})) \leq \sqrt{\lambda}(e^{-\dot{c}k/2} \bar{D}_{2,3} \mathbb{E}[\widetilde{V}_4(\theta_0)] + \bar{D}_{2,4}),$$

*where*

$$\bar{D}_{2,3} = \hat{c}\left(1 + \frac{2}{\dot{c}}\right)(e^{a/2}\bar{D}_{2,1} + 12),$$

$$\bar{D}_{2,4} = \frac{\hat{c}}{1 - \exp(-\dot{c})}(\bar{D}_{2,2} + 12c_3(\lambda_{max} + a^{-1}) + 9\widetilde{v}_4(\bar{M}_4) + 15),$$

(70)

*with $\bar{D}_{2,1}, \bar{D}_{2,2}$ given in 66, $\hat{c}, \dot{c}$ given in Lemma C.6, $c_3$ given in 95, and $\bar{M}_4$ given in Lemma C.15.*

*Proof.* This follows by applying Proposition C.5, Lemma C.7, Corollary C.14, and Lemma C.16 together with the arguments in Zhang et al. (2023, Proof of Lemma 4.8). $\square$

Adapting the reasoning of Lemma C.8, we establish a non-asymptotic $W_2$ bound between $\mathcal{L}(\bar{\Phi}_t^{\lambda,k,fSGLD})$ and $\mathcal{L}(Z_t^{\lambda,fSGLD})$, presented in the next corollary.

**Corollary C.9.** *Let Assumptions 1, 2, and 3 hold. For any $0 < \lambda < \lambda_{max}$ given in 24, $t \in (kT, (k+1)T]$,*

$$W_2(\mathcal{L}(\bar{\Phi}_t^{\lambda,k,fSGLD}), \mathcal{L}(Z_t^{\lambda,fSGLD})) \leq \lambda^{1/4}(e^{-\dot{c}/4} \bar{D}_{2,3}^\star (\mathbb{E}[\widetilde{V}_4(\theta_0)])^{1/2} + \bar{D}_{2,4}^\star),$$

*where*

$$\bar{D}_{2,3}^\star := \sqrt{2}\hat{c}(1 + 4/\dot{c})(e^{a/8}\bar{D}_{2,1}^{1/2} + 2\sqrt{2}),$$

$$\bar{D}_{2,4}^\star := \frac{\sqrt{2}\hat{c}}{1 - \exp(-\dot{c}/2)}(\bar{D}_{2,2}^{1/2} + 2\sqrt{2c_3}(\lambda_{max} + a^{-1})^{1/2} + \sqrt{3}\widetilde{v}_4^{1/2}(\bar{M}_4) + \sqrt{15}),$$

(71)

*with $\bar{D}_{2,1}, \bar{D}_{2,2}$ given in 66, $\hat{c}, \dot{c}$ given in Lemma C.6, $c_3$ given in 95, and $\bar{M}_4$ given in Lemma C.15.*

*Proof.* This follows using Proposition C.5, Lemma C.7, Corollary C.14, and Lemma C.16 in Zhang et al. (2023, Proof of Corollary 4.9). $\square$

We can now derive a non-asymptotic bound for the first three terms on the right-hand side of 59 in $W_1$ distance.

**Theorem C.10.** *Let Assumptions 1, 2, and 3 hold. Then, there exist constants $\dot{c}, D_1, D_2, D_3 > 0$ such that, for every $\beta > 0$, for $0 < \lambda < \lambda_{max}$, any $t \in (kT, (k+1)T]$, and $k \in \mathbb{N}$,*

$$W_1(\mathcal{L}(\bar{\theta}_t^{fSGLD}), \mathcal{L}(\bar{\Phi}_t^{\lambda,k,fSGLD})) + W_1(\mathcal{L}(\bar{\Phi}_t^{\lambda,k,fSGLD}), \mathcal{L}(Z_t^{\lambda,fSGLD})) + W_1(\mathcal{L}(Z_t^{\lambda,fSGLD}), \pi_\beta^{fSGLD})$$

$$\leq D_1 e^{-\dot{c}\lambda k/2}(1 + \mathbb{E}[|\theta_0|^4]) + (D_2 + D_3)\sqrt{\lambda},$$

*where*

$$D_1 := 2e^{\dot{c}/2}\left[(\lambda_{max}^{1/2}(\bar{D}_{2,1}^{1/2} + \bar{D}_{2,2}^{1/2} + \bar{D}_{2,3} + \bar{D}_{2,4}) + \hat{c}) + \hat{c}\left(1 + \int_{\mathbb{R}^d} \widetilde{V}_2(\theta)\pi_{\beta,\sigma}(\mathrm{d}\theta)\right)\right]$$

$$= O\left(e^{D_\star(1+d/\beta)(1+\beta)}\left(1 + \frac{1}{1 - e^{-\dot{c}}}\right)\right),$$

(72)

$$D_2 := \bar{D}_{2,1}^{1/2} + \bar{D}_{2,2}^{1/2} = O\left(1 + \sqrt{\frac{d}{\beta}}\right),$$

$$D_3 := \bar{D}_{2,3} + \bar{D}_{2,4} = O\left(e^{D_\star(1+d/\beta)(1+\beta)}\left(1 + \frac{1}{1 - e^{-\dot{c}}}\right)\right),$$

with $\hat{c}$, $\dot{c}$ given in Lemma C.6, $\bar{D}_{2,1}$, $\bar{D}_{2,2}$ given in 66 (Lemma C.7), $\bar{D}_{2,3}$, $\bar{D}_{2,4}$ given in 70 (Lemma C.8), $D_\star > 0$ is independent of $d$, $\beta$, $k$.

*Proof.* Using Lemma C.7, and Lemma C.8 in Zhang et al. (2023, Proof of Lemma 4.10), we obtain for $t \in (kT, (k+1)T]$,

$$
\begin{aligned}
& W_1(\mathcal{L}(\bar{\theta}_t^{\text{fSGLD}}), \mathcal{L}(\bar{\Phi}_t^{\lambda,k,\text{fSGLD}})) + W_1(\mathcal{L}(\bar{\Phi}_t^{\lambda,k,\text{fSGLD}}), \mathcal{L}(Z_t^{\lambda,\text{fSGLD}})) \\
& \leq (\bar{D}_{2,1}^{1/2} + +\bar{D}_{2,2}^{1/2} + \bar{D}_{2,3} + \bar{D}_{2,4})\sqrt{\lambda}[(e^{-\dot{c}k/2}\mathbb{E}[\widetilde{V}_4(\theta_0)] + 1)],
\end{aligned}
\tag{73}
$$

where $\bar{D}_{2,1}$, $\bar{D}_{2,2}$ are given in 66 (Lemma C.7), and $\bar{D}_{2,3}$, $\bar{D}_{2,4}$ are given in 70 (Lemma C.8). The remainder of the proof follows by applying 73 and Proposition C.5 in Zhang et al. (2023, Proof of Theorem 2.4). $\qquad\square$

An analogous result to Theorem C.10 holds in Wasserstein-2 distance, as stated in the next corollary.

**Corollary C.11.** *Let Assumption 1, 2 and 3 hold. Then, there exists constants $\dot{c}, D_4, D_5, D_6 > 0$ such that, for every $\beta > 0$, $0 < \lambda \leq \lambda_{max}$, any $t \in (kT, (k+1)T]$, and $k \in \mathbb{N}$,*

$$
\begin{aligned}
& W_2(\mathcal{L}(\bar{\theta}_t^{fSGLD}), \mathcal{L}(\bar{\Phi}_t^{\lambda,k,fSGLD})) + W_2(\mathcal{L}(\bar{\Phi}_t^{\lambda,k,fSGLD}), \mathcal{L}(Z_t^{\lambda,fSGLD})) + W_2(\mathcal{L}(Z_t^{\lambda,fSGLD}), \pi_\beta^{fSGLD}) \\
& \leq D_4 e^{-\dot{c}\lambda k/4}(\mathbb{E}[|\theta_0|^4] + 1) + (D_5 + D_6)\lambda^{1/4},
\end{aligned}
$$

*where*

$$
\begin{aligned}
D_4 &:= 2(\lambda_{max}^{1/2}(\bar{D}_{2,1}^{1/2} + \bar{D}_{2,2}^{1/2}) + \lambda_{max}^{1/4}(\bar{D}_{2,3}^\star + \bar{D}_{2,4}^\star) + \sqrt{2}\hat{c}^{1/2}) \\
& \quad + \sqrt{2}\hat{c}^{1/2}\left(1 + \int_{\mathbb{R}^d} \widetilde{V}_2(\theta)\pi_\beta^{fSGLD}(\mathrm{d}\theta)\right) \\
& = O\left(e^{D_\star(1+d/\beta)(1+\beta)}\left(1 + \frac{1}{1 - e^{-\dot{c}/2}}\right)\right) \\
D_5 &:= \lambda_{max}^{1/4}\bar{D}_{2,1}^{1/2} + \lambda_{max}^{1/4}\bar{D}_{2,2}^{1/2} = O\left(1 + \sqrt{\frac{d}{\beta}}\right) \\
D_6 &:= \bar{D}_{2,3}^\star + \bar{D}_{2,4}^\star = O\left(e^{D_\star(1+d/\beta)(1+\beta)}\left(1 + \frac{1}{1 - e^{-\dot{c}/2}}\right)\right),
\end{aligned}
\tag{74}
$$

*where $\hat{c}$, $\dot{c}$ given in Lemma C.6, $\bar{D}_{2,1}$, $\bar{D}_{2,2}$ given in 66 (Lemma C.7), $\bar{D}_{2,3}^\star$, $\bar{D}_{2,4}^\star$ given in 71 (Corollary C.9), $D_\star > 0$ is independent of $d$, $\beta$, $k$.*

*Proof.* This follows by applying Lemma C.7, Corollary C.9, and Proposition C.5 in Zhang et al. (2023, Proof of Corollary 2.5). $\qquad\square$

*Proof of Theorem 3.2.* Using Theorem C.10 and Proposition 3.1 in 59, we get

$$
\begin{aligned}
W_1(\mathcal{L}(\theta_k^{\text{fSGLD}}), \pi_{\beta,\sigma}^\star) &\leq W_1(\mathcal{L}(\bar{\theta}_t^{\text{fSGLD}}), \mathcal{L}(\bar{\Phi}_t^{\lambda,k,\text{fSGLD}})) + W_1(\mathcal{L}(\bar{\Phi}_t^{\lambda,k,\text{fSGLD}}), \mathcal{L}(Z_t^{\lambda,\text{fSGLD}})) \\
& \quad + W_1(\mathcal{L}(Z_t^{\lambda,\text{fSGLD}}), \pi_\beta^{\text{fSGLD}}) + W_1(\pi_\beta^{\text{fSGLD}}, \pi_{\beta,\sigma}^\star) \\
& \leq D_1 e^{-\dot{c}\lambda k/2}(1 + \mathbb{E}[|\theta_0|^4]) + (D_2 + D_3)\sqrt{\lambda} + W_2(\pi_\beta^{\text{fSGLD}}, \pi_{\beta,\sigma}^\star) \\
& \leq D_1 e^{-\dot{c}\lambda k/2}(1 + \mathbb{E}[|\theta_0|^4]) + (D_2 + D_3)\sqrt{\lambda} + \underline{D}.
\end{aligned}
\tag{75}
$$

In addition, for any $\bar{\delta} > 0$, if we choose $\lambda$, $k$ and $\beta$ in 75 such that $\lambda \leq \lambda_{max}$, and

$$
D_1 e^{-\dot{c}\lambda k/2}(1 + \mathbb{E}[|\theta_0|^4]) \leq \frac{\bar{\delta}}{3}, \qquad (D_2 + D_3)\sqrt{\lambda} \leq \frac{\bar{\delta}}{3}, \qquad \underline{D} \leq \frac{\bar{\delta}}{3},
$$

then $W_1(\mathcal{L}(\theta_k^{\text{fSGLD}}), \pi_{\beta,\sigma}^\star) \leq \bar{\delta}$. This yields

$$
\beta \geq \beta_{\bar{\delta}} := \left(3\underline{D}^0/\bar{\delta}\right)^{\frac{1}{\eta}},
\tag{76}
$$

where $\underline{D}^0$ contains the terms independent of $\beta$ in the right-hand side of 57, and

$$\lambda \leq \lambda_{\bar{\delta}} := \frac{\bar{\delta}^2}{9(D_2 + D_3)^2} \wedge \lambda_{\max}, \tag{77}$$

and $\lambda k \geq \frac{2}{\dot{c}} \ln \left( \frac{3D_1(1+\mathbb{E}[|\theta_0|^4])}{\bar{\delta}} \right)$. From 72, it follows that

$$k \geq k_{\bar{\delta}} := \frac{D_\star e^{D_\star(1+d/\beta)(1+\beta)}}{\bar{\delta}^2 \dot{c}} \left( 1 + \frac{1}{(1 - e^{-\dot{c}})^2} \right) \ln \left( \frac{D_\star e^{D_\star(1+d/\beta)(1+\beta)}}{\bar{\delta}} \left( 1 + \frac{1}{1 - e^{-\dot{c}}} \right) \right). \tag{78}$$

$\square$

*Proof of Corollary 3.3.* Using triangle inequality, Corollary C.11, and Proposition 3.1 in 59, we get, for any $t \in (kT, (k+1)T]$, and $k \in \mathbb{N}$,

$$W_2(\mathcal{L}(\theta_k^{\text{fSGLD}}), \pi_{\beta,\sigma}^\star) \leq W_2(\mathcal{L}(\bar{\theta}_t^{\text{fSGLD}}), \mathcal{L}(\bar{\Phi}_t^{\lambda,k,\text{fSGLD}})) + W_2(\mathcal{L}(\bar{\Phi}_t^{\lambda,k,\text{fSGLD}}), \mathcal{L}(Z_t^{\lambda,\text{fSGLD}}))$$
$$+ W_2(\mathcal{L}(Z_t^{\lambda,\text{fSGLD}}), \pi_\beta^{\text{fSGLD}}) + W_2(\pi_\beta^{\text{fSGLD}}, \pi_{\beta,\sigma}^\star) \tag{79}$$
$$\leq D_4 e^{-\dot{c}\lambda k/4}(\mathbb{E}[|\theta_0|^4] + 1) + (D_5 + D_6)\lambda^{1/4} + \underline{D},$$

In addition, for any $\widetilde{\delta} > 0$, $\lambda$, $k$ and $\beta$ such that $\lambda \leq \lambda_{\max}$, and

$$D_4 e^{-\dot{c}\lambda k/4}(\mathbb{E}[|\theta_0|^4] + 1) \leq \frac{\widetilde{\delta}}{3}, \qquad (D_5 + D_6)\lambda^{1/4} \leq \frac{\widetilde{\delta}}{3}, \qquad \underline{D} \leq \frac{\widetilde{\delta}}{3},$$

then $W_2(\mathcal{L}(\theta_k^{\text{fSGLD}}), \pi_{\beta,\sigma}^\star) \leq \widetilde{\delta}$. This yields

$$\beta \geq \beta_{\widetilde{\delta}} := \left( 3\underline{D}^0/\widetilde{\delta} \right)^{\frac{1}{\eta}}, \tag{80}$$

where $\underline{D}^0$ is the same as in the proof of Theorem 3.2,

$$\lambda \leq \lambda_{\widetilde{\delta}} := \frac{\bar{\delta}^4}{81(D_5 + D_6)^4} \wedge \lambda_{\max}, \tag{81}$$

and $\lambda k \geq \frac{4}{\dot{c}} \ln \left( \frac{3D_4(1+\mathbb{E}[|\theta_0|^4])}{\bar{\delta}} \right)$. From 74, it follows that

$$k \geq k_{\widetilde{\delta}} := \frac{D_\star e^{D_\star(1+d/\beta)(1+\beta)}}{\bar{\delta}^4 \dot{c}} \left( 1 + \frac{1}{(1 - e^{-\dot{c}/2})^4} \right) \ln \left( \frac{D_\star e^{D_\star(1+d/\beta)(1+\beta)}}{\bar{\delta}} \left( 1 + \frac{1}{1 - e^{-\dot{c}/2}} \right) \right). \tag{82}$$

$\square$

**Remark C.12.** *The convergence rate in Corollary 3.3 can be improved to $O(\lambda^{\frac{1}{2}})$ under substantially stronger assumptions than Assumption 1, 2 and 3. For example, one may assume that the $\pi_\beta^{\text{fSGLD}}$ satisfies a log-Sobolev inequality, as in Huang et al. (2025). However, such assumptions go beyond the scope of our work.*

*Proof of Theorem 3.6.* We begin by decomposing the expected excess risk using the random variable $Z_\infty^{\text{fSGLD}}$, for which $\mathcal{L}(Z_\infty^{\text{fSGLD}}) = \pi_\beta^{\text{fSGLD}}$, and obtain

$$\mathbb{E}[g_\epsilon(\theta_k^{\text{fSGLD}})] - \inf_{\theta \in \mathbb{R}^d} g_\epsilon(\theta)$$
$$= (\mathbb{E}[g_\epsilon(\theta_k^{\text{fSGLD}})] - \mathbb{E}[g_\epsilon(Z_\infty)]) + (\mathbb{E}[g_\epsilon(Z_\infty)] - \inf_{\theta \in \mathbb{R}^d} g_\epsilon(\theta)). \tag{83}$$

We proceed by controlling the two terms on the right-hand side of 83 separately. By using Raginsky et al. (2017, Lemma 3.5), Remark B.1 with $\sigma^2 = \beta^{-\frac{1+\eta}{2}}$ for $\eta \in (0, 1)$, Lemma C.13, and Corollary C.11, the first term on the RHS of 83 can be bounded by

$$\mathbb{E}[g_\epsilon(\theta_k^{\text{fSGLD}})] - \mathbb{E}[g_\epsilon(Z_\infty)] \leq D_1^\# e^{-\dot{c}\lambda k/4} + D_2^\# \lambda^{1/4}, \tag{84}$$

where

$$D_1^\# := D_4(L_1\mathbb{E}[\varphi(X_0)](\mathbb{E}[|\theta_0|^2] + c_1(\lambda_{\max} + a^{-1})) + L_2\mathbb{E}[\bar{\varphi}(X_0)](1 + d\beta^{-(1+\eta)/2}) + \mathbb{E}[\widetilde{G}(\epsilon)])$$
$$\times (\mathbb{E}[|\theta_0|^4] + 1),$$
$$D_2^\# := (D_5 + D_6)$$
$$\times (L_1\mathbb{E}[\varphi(X_0)](\mathbb{E}[|\theta_0|^2] + c_1(\lambda_{\max} + a^{-1})) + L_2\mathbb{E}[\bar{\varphi}(X_0)](1 + d\beta^{-(1+\eta)/2}) + \mathbb{E}[\widetilde{G}(\epsilon)]),$$

(85)

with $\dot{c}$ given in 63, $D_4$, $D_5$, $D_6$ given in 74, and $c_1$ given in 94. The second term on the RHS of 83 can be controlled via Raginsky et al. (2017, Proposition 11), which leads to

$$\mathbb{E}[g_\epsilon(Z_\infty)] - \inf_{\theta \in \mathbb{R}^d} g_\epsilon(\theta) \leq D_\diamond^\#,$$

(86)

where

$$D_\diamond^\# := \frac{d}{2\beta} \log\left(\frac{eL_1\mathbb{E}[\varphi(X_0)]}{a}\left(\frac{b\beta}{d} + 1\right)\right).$$

(87)

Using the estimates from 84 and 86 in 83, we obtain

$$\mathbb{E}[g_\epsilon(\theta_k^{\text{fSGLD}})] - \inf_{\theta \in \mathbb{R}^d} g_\epsilon(\theta) \leq D_1^\# e^{-\dot{c}\lambda k/4} + D_2^\# \lambda^{1/4} + D_\diamond^\#.$$

(88)

Applying 16 on the LHS of 88, along with 17, and choosing $\sigma^4 = \beta^{-(1+\eta)}$, it follows that

$$\mathbb{E}[g_\epsilon(\theta_k^{\text{fSGLD}})] - \inf_{\theta \in \mathbb{R}^d} v(\theta) \leq D_1^\# e^{-\dot{c}\lambda k/4} + D_2^\# \lambda^{1/4} + D_3^\#,$$

(89)

where

$$D_1^\# = O\left(e^{D_\star(1+d/\beta)(1+\beta)}\left(1 + \frac{1}{1 - e^{-\dot{c}/2}}\right)\right),$$
$$D_2^\# = O\left(e^{D_\star(1+d/\beta)(1+\beta)}\left(1 + \frac{1}{1 - e^{-\dot{c}/2}}\right)\right),$$
$$D_3^\# := D_\diamond^\# + \beta^{-(1+\eta)} \inf_{\theta \in \mathbb{R}^d}\left(\sum_{i,j,k,l=1}^d \frac{\partial^4 u}{\partial\theta_i \partial\theta_j \partial\theta_k \partial\theta_l}(\theta) + \sum_{j=6}^\infty \sum_{i_1,i_2,\ldots,i_j=1}^d \frac{\partial^j u}{\partial\theta_{i_1} \partial\theta_{i_2} \ldots \partial\theta_{i_j}}(\theta)\right)$$
$$= O\left((d/\beta)\log(D_\star(\beta^{(1-\eta)/2} + 1))\right),$$

(90)

with $D_\star > 0$ a constant independent of $d$, $\beta$, $k$. In addition, for $\underline{\delta} > 0$, if we choose $\beta$ such that $D_3^\# \leq \underline{\delta}/3$, then choose $\lambda$ such that $\lambda \leq \lambda_{\max}$ and $D_2^\# \lambda^{1/4} \leq \underline{\delta}/3$, and choose $k$ such that $D_1^\sharp e^{-\dot{c}\lambda k/4} \leq \underline{\delta}/3$, we obtain

$$\mathbb{E}[g_\epsilon(\theta_k^{\text{fSGLD}})] - \inf_{\theta \in \mathbb{R}^d} v(\theta) \leq \underline{\delta}.$$

This yields

$$\beta \geq \beta_{\underline{\delta}} := \beta_c \vee \frac{9d}{2\underline{\delta}} \log\left(\frac{eL_1\mathbb{E}[\varphi(X_0)]}{ad}(b+1)(d+1)\right)$$
$$\vee \left[\frac{9}{\underline{\delta}} \inf_{\theta \in \mathbb{R}^d}\left(\sum_{i,j,k,l=1}^d \frac{\partial^4 u}{\partial\theta_i \partial\theta_j \partial\theta_k \partial\theta_l}(\theta) + \sum_{j=6}^\infty \sum_{i_1,i_2,\ldots,i_j=1}^d \frac{\partial^j u}{\partial\theta_{i_1} \partial\theta_{i_2} \ldots \partial\theta_{i_j}}(\theta)\right)\right]^{\frac{1}{1+\eta}},$$

(91)

where $\beta_c$ is the root of the function $f^\sharp(\beta) = \frac{\log(\beta+1)}{\beta} - \frac{2\delta}{9d}$, with $\beta > 0$. Since

$$D_3^\sharp \leq \frac{d}{2\beta} \log\left(\frac{eL_1\mathbb{E}[\varphi(X_0)]}{ad}(b+1)(d+1)(\beta+1)\right)$$
$$+ \beta^{-(1+\eta)} \inf_{\theta \in \mathbb{R}^d}\left(\sum_{i,j,k,l=1}^d \frac{\partial^4 u}{\partial\theta_i \partial\theta_j \partial\theta_k \partial\theta_l}(\theta) + \sum_{j=6}^\infty \sum_{i_1,i_2,\ldots,i_j=1}^d \frac{\partial^j u}{\partial\theta_{i_1} \partial\theta_{i_2} \ldots \partial\theta_{i_j}}(\theta)\right),$$

we can ensure $D_3^\sharp \leq \underline{\delta}/3$ by imposing

$$\frac{d}{2\beta} \log \left( \frac{eL_1 \mathbb{E}[\varphi(X_0)]}{ad} (b+1)(d+1) \right) \leq \frac{\delta}{9}, \qquad \frac{d}{2\beta} \log (\beta + 1) \leq \frac{\delta}{9},$$

$$\beta^{-(1+\eta)} \inf_{\theta \in \mathbb{R}^d} \left( \sum_{i,j,k,l=1}^{d} \frac{\partial^4 u}{\partial \theta_i \partial \theta_j \partial \theta_k \partial \theta_l}(\theta) + \sum_{j=6}^{\infty} \sum_{i_1,i_2,\ldots,i_j=1}^{d} \frac{\partial^j u}{\partial \theta_{i_1} \partial \theta_{i_2} \ldots \partial \theta_{i_j}}(\theta) \right) \leq \frac{\delta}{9}.$$

Moreover, one can verify that

$$\lambda \leq \lambda_{\underline{\delta}} := \frac{\underline{\delta}^4}{81(D_2^\sharp)^4} \wedge \lambda_{\max}, \tag{92}$$

and $\lambda k \geq \frac{4}{\dot{c}} \ln \frac{3D_1^\sharp}{\underline{\delta}}$, where $\dot{c}$ is given explicitly in Lemma C.6. This leads to

$$k \geq k_{\underline{\delta}} := \frac{D_\star e^{D_\star(1+d/\beta)(1+\beta)}}{\underline{\delta}^4 \dot{c}} \left( 1 + \frac{1}{(1-e^{-\dot{c}/2})^4} \right)$$

$$\times \ln \left( \frac{D_\star e^{D_\star(1+d/\beta)(1+\beta)}}{\underline{\delta}} \left( 1 + \frac{1}{1-e^{-\dot{c}/2}} \right) \right). \tag{93}$$

$\square$

## C.3 AUXILIARY RESULTS

We present the auxiliary results required for the convergence analysis in Appendix C.2. Their proofs follow the same lines as Zhang et al. (2023), with Zhang et al. (2023, Assumptions 1–3) replaced by Assumptions 1, together with the properties established in Remark B.1 and Remark B.2. For completeness, we include their statements to make the convergence analysis of fSGLD self-contained.

**Lemma C.13** (Moment bounds of 27). *Let Assumption 1, 2 and 3 hold. For any $0 < \lambda \leq \lambda_{max}$ given in 24, $k \in \mathbb{N}$, $t \in (k, k+1]$,*

$$\mathbb{E}\left[ |\bar{\theta}_t^{fSGLD}|^2 \right] \leq (1 - a\lambda(t-k))(1-a\lambda)^k \, \mathbb{E}[|\theta_0|^2] + c_1(\lambda_{max} + a^{-1}),$$

*where $a$ and $b$ are given in Remark B.2, and*

$$c_1 := c_0 + 2d\beta^{-1}, \quad c_0 := 2b + 8\lambda_{\max} L_2^2 \mathbb{E}\left[ \bar{\varphi}^2(X_0) \right] (1 + \sigma^2 d) + 4\lambda_{\max} \mathbb{E}[\widetilde{G}^2(\epsilon)]. \tag{94}$$

*Moreover, $\sup_{t>0} \mathbb{E}[|\bar{\theta}_t^{fSGLD}|^2] \leq \mathbb{E}[|\theta_0|^2] + c_1(\lambda_{max} + a^{-1}) < \infty$. By a similar argument, one obtains*

$$\mathbb{E}\left[ |\bar{\theta}_t^{fSGLD}|^4 \right] \leq (1 - a\lambda(t-k))(1-a\lambda)^k \, \mathbb{E}[|\bar{\theta}_0^{fSGLD}|^4] + c_3(\lambda_{max} + a^{-1}),$$

*where*

$$M := \max\{(8ba^{-1} + 48a^{-1}\lambda_{\max}(L_2^2 \mathbb{E}\left[ \bar{\varphi}^2(X_0) \right] (1 + \sigma^2 d) + \mathbb{E}[\widetilde{G}^2(\epsilon)]))^{1/2},$$

$$(128a^{-1}\lambda_{\max}^2(L_2^3 \mathbb{E}\left[ \bar{\varphi}^3(X_0) \right] \mathbb{E}[(1+|\epsilon|)^3] + \mathbb{E}[\widetilde{G}^3(\epsilon)]))^{1/3}\},$$

$$c_2 := 4bM^2 + 152(1 + \lambda_{\max})^3 \tag{95}$$

$$\times \left( (1+L_2)^4 \mathbb{E}\left[ (1 + \bar{\varphi}(X_0))^4 \right] \mathbb{E}[(1+|\epsilon|)^4] + \mathbb{E}[(1+\widetilde{G}(\epsilon))^4] \right) (1+M)^2,$$

$$c_3 := (1 + a\lambda_{max})c_2 + 12d^2\beta^{-2}(\lambda_{max} + 9a^{-1}).$$

*Moreover, this implies $\sup_{t>0} \mathbb{E}[|\bar{\theta}_t^{fSGLD}|^4] < \infty$.*

*Proof.* This follows along the same lines as Zhang et al. (2023, Lemma 4.2) under our own Assumptions 1, 2, and 3, and using the estimates in Remark B.1 and B.2. $\square$

Lemma C.13 provides a uniform fourth-moment bound for the process $(\bar{\theta}_t^{fSGLD})_{t\geq 0}$ which in turn yields a uniform bound for $\widetilde{V}_4(\bar{\theta}_t^{fSGLD})$, as given in the next corollary.

**Corollary C.14.** *Let Assumption 1, 2 and 3 hold. For any $0 < \lambda < \lambda_{max}$, $k \in \mathbb{N}$, $t \in (k, k+1]$,*

$$\mathbb{E}[\widetilde{V}_4(\bar{\theta}_t^{fSGLD})] \leq 2(1 - a\lambda)^{\lfloor t \rfloor} \mathbb{E}[\widetilde{V}_4(\bar{\theta}_0^{fSGLD})] + 2c_3(\lambda_{max} + a^{-1}) + 2,$$

*where $c_3$ is given in Lemma C.13.*

*Proof.* This follows from the definition of the Lyapunov function $\widetilde{V}_4$ together with Lemma C.13. $\square$

We establish a drift condition for the flatness Langevin SDE 25, which will be instrumental in deriving moment bounds for the continuous-time proces $\bar{\Phi}_t^{\lambda, k, \text{ fSGLD}}$ in Lemma C.16.

**Lemma C.15.** *(Chau et al., 2021, Lemma 3.5) Let Assumption 1 and 3 hold. Then, for each $p \geq 2$, $\theta \in \mathbb{R}^d$,*

$$\Delta \widetilde{V}_p(\theta)\beta^{-1} - \langle \nabla g_\epsilon(\theta), \nabla \widetilde{V}_p(\theta) \rangle \leq -\bar{c}(p)\widetilde{V}_p(\theta) + \tilde{c}(p),$$

*where $\bar{c}(p) := ap/4$ and $\tilde{c}(p) := (3/4)ap\,\widetilde{v}_p(\bar{M}_p)$ with $\bar{M}_p := (1/3 + 4b/(3a) + 4d/(3a\beta) + 4(p-2)/(3a\beta))^{1/2}$.*

**Lemma C.16.** *Let Assumption 1, 2 and 3 hold. For any $0 < \lambda < \lambda_{max}$, $t \geq kT$, with $k \in \mathbb{N}$, the following inequality holds*

$$\mathbb{E}[\widetilde{V}_2(\bar{\Phi}_t^{\lambda, k, fSGLD})] \leq e^{-\lambda t a/2}\mathbb{E}[\widetilde{V}_2(\theta_0)] + c_1(\lambda_{max} + a^{-1}) + 3\widetilde{v}_2(\bar{M}_2) + 1,$$

*where $c_1$ is given in Lemma C.13. In addition, the following inequality holds*

$$\mathbb{E}[\widetilde{V}_4(\bar{\Phi}_t^{\lambda, k, fSGLD})] \leq 2e^{-a\lambda t}\mathbb{E}[\widetilde{V}_4(\bar{\theta}_0^{fSGLD})] + 3\widetilde{v}_4(\bar{M}_4) + 2c_3(\lambda_{max} + a^{-1}) + 2,$$

*where $\bar{M}_2$ and $\bar{M}_4$ are given in Lemma C.15, and $c_3$ is given in Lemma C.13.*

*Proof.* This follows by applying Lemma C.13, Corollary C.14, and Lemma C.15 in Zhang et al. (2023, Proof of Lemma 4.5). $\square$

**Lemma C.17.** *Let Assumption 1, 2 and 3 hold, and let $\lambda_{max}$ be given in 24. Then, for any $t > 0$,*

$$\mathbb{E}\left[|\bar{\theta}_{\lfloor t \rfloor}^{fSGLD} - \bar{\theta}_t^{fSGLD}|^2\right] \leq \lambda \left[e^{-\lambda a \lfloor t \rfloor}\bar{\psi}_Y \mathbb{E}[\widetilde{V}_2(\theta_0)] + \widetilde{\psi}_Y\right],$$

*where*

$$\bar{\psi}_Y := 2\lambda_{max}L_1^2\mathbb{E}[\varphi^2(X_0)],$$
$$\widetilde{\psi}_Y := 2c_1L_1^2\lambda_{max}\mathbb{E}[\varphi^2(X_0)](\lambda_{max} + a^{-1}) + 4\lambda_{max}L_2^2\mathbb{E}[\bar{\varphi}^2(X_0)] + 4\lambda_{max}\mathbb{E}[\widetilde{G}^2(\epsilon)] + 2d\beta^{-1},$$

(96)

*with $c_1$ given in Lemma C.13.*

*Proof.* This follows by applying Remark B.1 and Lemma C.13 in Zhang et al. (2023, Proof of Lemma A.2). $\square$

**Lemma C.18.** *Let Assumption 1, 2 and 3 hold. For any $t \in (kT, (k+1)T]$, with $k, N \in \mathbb{N}$ and $n = 1, \ldots, N+1$, where $N+1 \leq T$, one obtains*

$$\mathbb{E}[|\nabla g_\epsilon(\bar{\Phi}_t^{\lambda, k, fSGLD}) - \nabla_\theta U(\bar{\Phi}_t^{\lambda, k, fSGLD} + \epsilon_{kT+n}, X_{kT+n})|^2] \leq e^{-a\lambda t/2}\bar{\psi}_Z\mathbb{E}[\widetilde{V}_2(\theta_0)] + \widetilde{\psi}_Z,$$

*where*

$$\bar{\psi}_Z = 8L_2^2\mathbb{E}[(\varphi(X_0) + \varphi(\mathbb{E}[X_0]))^2|X_0 - \mathbb{E}[X_0]|^2],$$
$$\widetilde{\psi}_Z = 8L_2^2E[(\varphi(X_0) + \varphi(\mathbb{E}[X_0]))^2|X_0 - \mathbb{E}[X_0]|^2](3\widetilde{v}_2(\bar{M}_2) + c_1(\lambda_{max} + a^{-1}) + 1 + \sigma^2 d),$$

(97)

*with $\bar{M}_2$ and $c_1$ given in Lemma C.15 and Lemma C.13, respectively.*

*Proof.* We adapt the Zhang et al. (2023, Proof of Lemma A.1). First, we define the filtration $\mathcal{J}_t = \mathcal{F}_\infty^\lambda \vee \mathcal{X}_{\lfloor t \rfloor} \vee \mathcal{H}_{\lfloor \epsilon \rfloor}$. Then, the result follows by an application of Lemma C.19, Remark B.1, and Lemma C.16

$$\mathbb{E}\left[\left|\nabla g_\epsilon(\bar{\Phi}_t^{\lambda,k,\mathrm{fSGLD}}) - \nabla_\theta U(\bar{\Phi}_t^{\lambda,k,\mathrm{fSGLD}} + \epsilon_{kT+n}, X_{kT+n})\right|^2\right]$$

$$= \mathbb{E}\left[\mathbb{E}\left[\left|\nabla g_\epsilon(\bar{\Phi}_t^{\lambda,k,\mathrm{fSGLD}}) - \nabla_\theta U(\bar{\Phi}_t^{\lambda,k,\mathrm{fSGLD}} + \epsilon_{kT+n}, X_{kT+n})\right|^2 \middle| \mathcal{J}_{kT}\right]\right]$$

$$= \mathbb{E}\left[\mathbb{E}\left[\left|\mathbb{E}\left[\nabla_\theta U(\bar{\Phi}_t^{\lambda,k,\mathrm{fSGLD}} + \epsilon_{kT+n}, X_{kT+n})\middle| \mathcal{J}_{kT}\right]\right.\right.\right.$$

$$\left.\left.\left. - \nabla_\theta U(\bar{\Phi}_t^{\lambda,k,\mathrm{fSGLD}} + \epsilon_{kT+n}, X_{kT+n})\right|^2 \middle| \mathcal{J}_{kT}\right]\right]$$

$$\leq 4\mathbb{E}\left[\mathbb{E}\left[\left|\nabla_\theta U(\bar{\Phi}_t^{\lambda,k,\mathrm{fSGLD}} + \epsilon_{kT+n}, X_{kT+n})\right.\right.\right.$$

$$\left.\left.\left. - \nabla_\theta U(\bar{\Phi}_t^{\lambda,k,\mathrm{fSGLD}} + \epsilon_{kT+n}, \mathbb{E}\left[X_{kT+n} \middle| \mathcal{J}_{kT}\right])\right|^2 \middle| \mathcal{J}_{kT}\right]\right]$$

$$\leq 8L_2^2\mathbb{E}\left[(\varphi(X_0) + \varphi(\mathbb{E}[X_0]))^2|X_0 - \mathbb{E}[X_0]|^2\right]\left(\sigma^2 d + \mathbb{E}\left[\left(1 + \left|\bar{\Phi}_t^{\lambda,k,\mathrm{fSGLD}}\right|^2\right)\right]\right)$$

$$\leq 8L_2^2\mathbb{E}\left[(\varphi(X_0) + \varphi(\mathbb{E}[X_0]))^2|X_0 - \mathbb{E}[X_0]|^2\right]$$
$$\times \left(e^{-\lambda ta/2}\mathbb{E}[V_2(\theta_0)] + c_1(\lambda_{\max} + a^{-1}) + 3\widetilde{v}_2(\bar{M}_2) + 1 + \sigma^2 d\right).$$

$$\square$$

In the next lemma, $L^p$ denotes the usual space of $p$-integrable real-valued random variables for $1 \leq p < \infty$.

**Lemma C.19.** *Let $\mathcal{F}, \mathcal{X}, \mathcal{H} \subset \mathcal{M}$ be sigma-algebras. Let $X, Y$ be $\mathbb{R}^d$-valued random vectors in $L^p$ for any $p \geq 1$ such that $Y$ is measurable with respect to $\mathcal{F} \vee \mathcal{X} \vee \mathcal{H}$. Then,*

$$\mathbb{E}^{1/p}\left[|X - \mathbb{E}[X|\mathcal{F} \vee \mathcal{X} \vee \mathcal{H}]|^p| \mathcal{X} \vee \mathcal{H}\right] \leq 2\mathbb{E}^{1/p}\left[|X - Y|^p| \mathcal{X} \vee \mathcal{H}\right].$$

*Proof.* This follows by applying Chau et al. (2019, Lemma 6.1) to $\mathcal{F} \vee \mathcal{N}$, where the sigma-algebra $\mathcal{N} := \mathcal{X} \vee \mathcal{H}$. $\square$

# D EXPERIMENTAL DETAILS

## D.1 DETAILS FOR SECTION 4.2

### D.1.1 SOFTWARE AND HARDWARE ENVIRONMENTS

We conduct all experiments with PYTHON 3.10.9 and PYTORCH 1.13.1, CUDA 11.6.2, NVIDIA Driver 510.10 on Ubuntu 22.04.1 LTS server which equipped with AMD Ryzen Threadripper PRO 5975WX, NVIDIA A100 GPUs.

### D.1.2 IMPLEMENTATION DETAILS

We follow standard data preprocessing and augmentation strategies as adopted in prior work (Li et al., 2017; Wei et al., 2022) on noisy-label benchmarks. For CIFAR-10N and CIFAR-100N, we apply random cropping with padding, random horizontal flipping, and normalization using dataset-specific statistics. For WebVision, we follow the preprocessing protocol of Kodge (2024). We note that Noisy-label benchmarks and ViT fine-tuning are widely used in evaluating the optimizer's generalization ability (Luo et al., 2024; Baek et al., 2024; Tan et al., 2025).

Regarding model architectures, we employ the CIFAR-specific variants of ResNet-34 and ResNet-50 when training on CIFAR-10N and CIFAR-100N, where the first convolution layer is replaced by a $3 \times 3$ kernel with stride 1 (instead of the $7 \times 7$ stride-2 convolution and max pooling used in ImageNet models) to accommodate the smaller $32 \times 32$ resolution. For WebVision, we adopt the standard ResNet implementations as provided for ImageNet-scale data.

For both training-from-scratch and fine-tuning experiments, we use the same hyperparameter search spaces. Table 4 summarizes the ranges considered for each optimizer. We do not employ any early stopping or pruning strategy during the Optuna-based hyperparameter tuning, ensuring that each trial is fully evaluated to its final epoch. We performed the same number of hyper-parameter trials for all methods so that the search-space exploration budget (number of trials) was identical. Because each SAM update requires two gradient evaluations, this design implies that, for the same number of trials and training epochs, SAM consumed roughly twice the wall-clock compute time of the other baselines. Thus our tuning protocol is at least as favorable to SAM as to the proposed fSGLD, ensuring that our reported improvements are not due to weaker tuning of SAM.

For SGLD and fSGLD ($\beta$ fixed), we set a large inverse temperature $\beta = 10^{14}$. This follows the common heuristic of using a near-zero temperature to minimize exploration when employing Langevin Dynamics as a optimizer for a given objective. For fSGLD ($\beta$-$\sigma$ coupled), we leverage our theoretical analysis as a practical tuning strategy. We only search for the optimal perturbation scale $\sigma$ and then deterministically set $\beta$ via our theoretically-derived relationship, $\beta = \sigma^{-4/(1+\eta)}$ with $\eta = 0.01$. This is a practical choice, as a larger $\eta$ would cause $\beta$ to become too small, allowing the Langevin noise term to overwhelm the gradient term and turning the dynamics into a near-random exploration. A small $\eta$ thus ensures stable optimization. This principled approach significantly simplifies the search space.

Table 4: Hyperparameter search spaces for different optimizers.

| Optimizer | Learning rate | Momentum | Weight decay | Other hyperparameters |
|---|---|---|---|---|
| SGD | $10^{[-2,0]}$ | $\{0.1, 0.9\}$ | $5 \times 10^{-4}$ | – |
| AdamW | $10^{[-4,-2]}$ | – | $10^{-2}$ | $[\beta_1, \beta_2] \in \{[0.8, 0.95], [0.99, 0.999]\}$ |
| SGLD | $10^{[-2,0]}$ | – | $5 \times 10^{-4}$ | $\beta = 10^{14}$ |
| SAM | $10^{[-2,0]}$ | $\{0.1, 0.9\}$ | $5 \times 10^{-4}$ | $\rho \in 10^{[-3,-1]}$ |
| fSGLD ($\beta$ fixed) | $10^{[-2,0]}$ | – | $5 \times 10^{-4}$ | $\beta = 10^{14}$, $\sigma \in 10^{[-3,-2]}$ |
| fSGLD ($\beta$-$\sigma$ coupled) | $10^{[-2,0]}$ | – | $5 \times 10^{-4}$ | $\beta = \sigma^{-4/1.01}$, $\sigma \in 10^{[-3,-2]}$ |

## D.2 DETAILS FOR SECTION 4.5

For the Hessian spectrum analysis, we use the best-performing ResNet-34 model trained on CIFAR-10N under each optimizer setting. Given a trained network $f_\theta$ and loss function $L$, we compute Hessian-vector products (HVPs) by applying automatic differentiation to the scalar product $\nabla_\theta L^\top v$ for a random vector $v$. For eigenvalue computation, we adopt the Lanczos algorithm (Lin et al., 2016) as implemented in `scipy.sparse.linalg.eigsh`, which allows us to approximate the top-$k$ eigenvalues without explicitly forming the Hessian. In all reported results, we compute up to the top 50 eigenvalues. As a complementary measure of curvature, we estimate the trace of the Hessian using Hutchinson's stochastic estimator (Avron & Toledo, 2011) with Rademacher random vectors:

$$\mathrm{tr}(H(\theta)) \approx \frac{1}{m} \sum_{i=1}^{m} z_i^\top H(\theta) z_i, \quad z_i \sim \mathrm{Unif}\{\pm 1\}^d,$$

where $m = 1000$ in our experiments and $d$ denotes the number of model parameters.

The analysis is conducted on the CIFAR-10N, where we randomly subsample at most 1,000 examples to reduce computational overhead. Eigenvalue computations are performed with a tolerance of $10^{-4}$ and a maximum of 500 iterations for the Lanczos solver.

## D.3 TRAINING CURVES

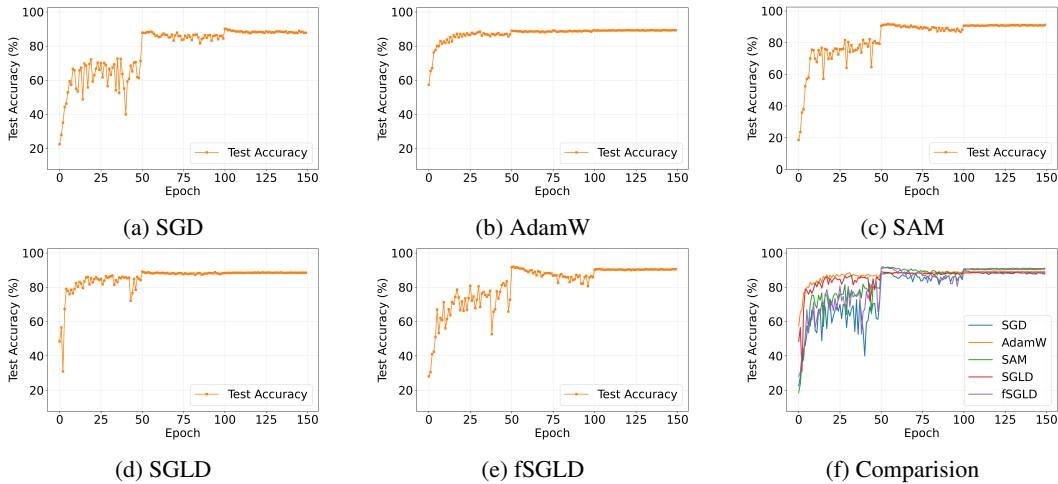

Figure 4: Training accuracy trajectories of ResNet-34 on CIFAR-10N using the best hyperparameter settings for each optimizer.

Figure 5: Test accuracy trajectories of ResNet-34 on CIFAR-10N using the best hyperparameter settings for each optimizer.

Table 5: The best hyperparameter settings for ResNet-34 on CIFAR-10N.

| Optimizer | Learning rate | Momentum | Weight decay | Other hyperparameters |
|---|---|---|---|---|
| SGD | $8.69 \times 10^{-1}$ | 0.71 | $5 \times 10^{-4}$ | – |
| AdamW | $3.32 \times 10^{-4}$ | – | $10^{-2}$ | $[\beta_1, \beta_2] = [0.81, 0.99]$ |
| SGLD | $7.13 \times 10^{-2}$ | – | $5 \times 10^{-4}$ | $\beta = 10^{14}$ |
| SAM | $5.51 \times 10^{-1}$ | 0.62 | $5 \times 10^{-4}$ | $\rho = 2.69 \times 10^{-2}$ |
| fSGLD | $9.58 \times 10^{-1}$ | – | $5 \times 10^{-4}$ | $\beta = \sigma^{-4/1.01}$, $\sigma = 2.46 \times 10^{-3}$ |

Figure 4 and Figure 5 present the training and test accuracy curves, respectively, for ResNet-34 trained on the CIFAR-10N dataset, utilizing the best hyperparameter configuration for each optimizer. The optimal hyperparameters are summarized in Table 5. Similarly, Figure 6 and Figure 7 illustrate the training and test accuracy trajectories for the CIFAR-100N dataset. The corresponding hyperparameter settings used for these experiments are detailed in Table 6.

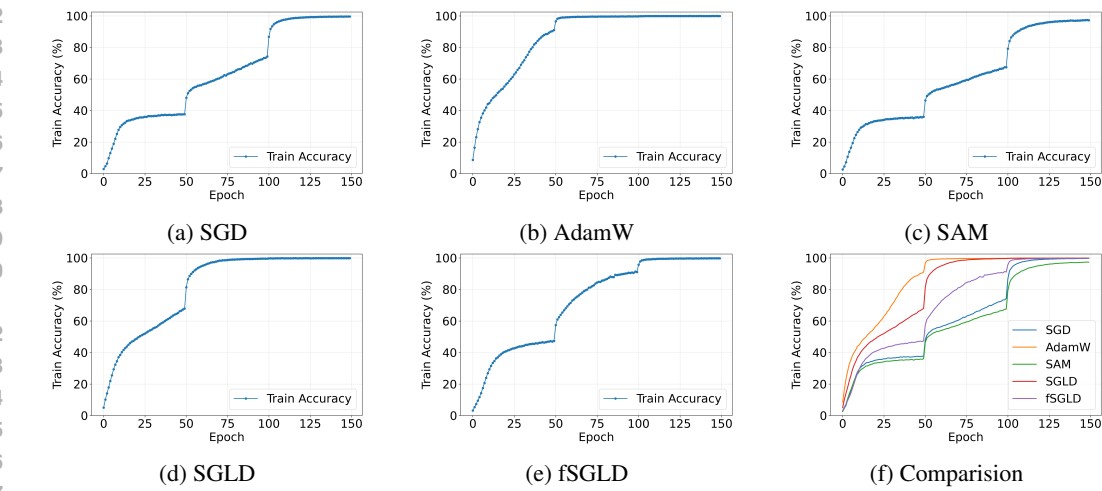

Figure 6: Training accuracy trajectories of ResNet-34 on CIFAR-100N using the best hyperparameter settings for each optimizer.

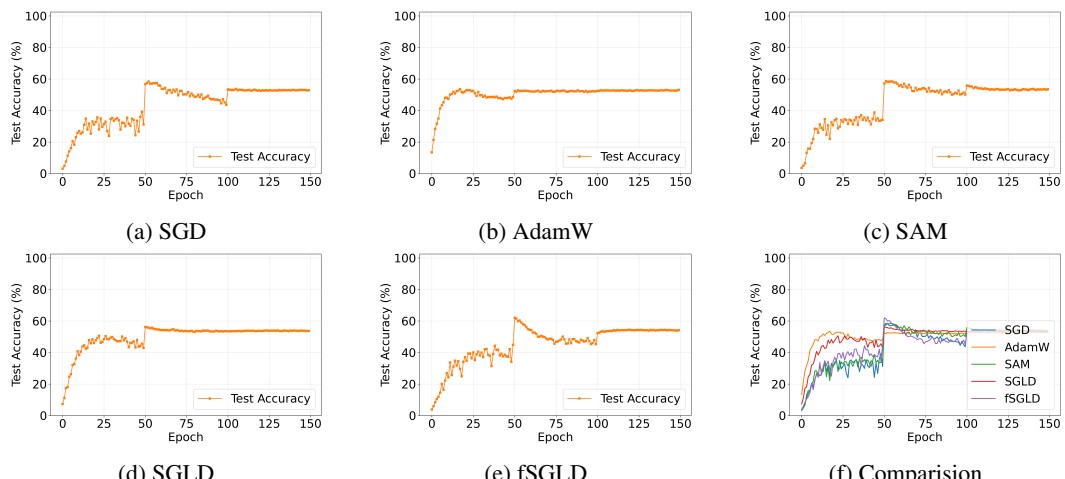

Figure 7: Test accuracy trajectories of ResNet-34 on CIFAR-100N using the best hyperparameter settings for each optimizer.

Table 6: The best hyperparameter settings for ResNet-34 on CIFAR-100N.

| Optimizer | Learning rate | Momentum | Weight decay | Other hyperparameters |
|---|---|---|---|---|
| SGD | $7.76 \times 10^{-1}$ | 0.69 | $5 \times 10^{-4}$ | – |
| AdamW | $3.32 \times 10^{-4}$ | – | $10^{-2}$ | $[\beta_1, \beta_2] = [0.81, 0.99]$ |
| SGLD | $1.59 \times 10^{-1}$ | – | $5 \times 10^{-4}$ | $\beta = 10^{14}$ |
| SAM | $7.77 \times 10^{-1}$ | 0.73 | $5 \times 10^{-4}$ | $\rho = 8.84 \times 10^{-3}$ |
| fSGLD | $9.45 \times 10^{-1}$ | – | $5 \times 10^{-4}$ | $\beta = \sigma^{-4/1.01}$, $\sigma = 2.47 \times 10^{-3}$ |

# E USE OF LARGE LANGUAGE MODELS (LLMS).

In this manuscript, we used LLMs solely for writing assistance, such as grammar checking and minor language polishing. All sentences and substantive content of the paper are entirely our own. LLMs were not used for retrieval, discovery, research ideation, or any other purpose beyond basic language editing.

