# OpenReview forum: "Flatness-Aware Stochastic Gradient Langevin Dynamics"
_ICLR.cc/2026/Conference — Submitted to ICLR 2026_

### Official Review · Reviewer_QXeJ · 2025-10-15

**Soundness:** 1
**Presentation:** 2
**Contribution:** 2
**Rating:** 0
**Confidence:** 4

**Summary:**

The paper advances the literature by establishing a formal coupling between the Langevin temperature and random perturbation scale that provably links stochastic weight perturbations to convergence toward globally flat minima.

**Strengths:**

1. **Clear theoretical contribution.**
The paper rigorously establishes that a specific coupling between the inverse temperature $\beta$ and the perturbation scale $\sigma$ guarantees that the invariant measure of fSGLD concentrates on flat minima.

2. **Strong theoretical guarantees.**
The paper provides non-asymptotic convergence bounds in Wasserstein distance and an excess-risk bound, matching the best known rates for SGLD under comparable assumptions.

3. **Empirical validation consistent with the theory.**
The experiments demonstrate that the theoretically prescribed $\beta$–$\sigma$ coupling consistently improves generalization and stability across some noisy-label and fine-tuning tasks, and in general, that your method is better than the others in most cases. Importantly, you tuned the hyperparameters fairly.

**Weaknesses:**

1. **Questionable premise on flatness and generalization.**
   Contrary to the claim made in the Abstract, I do not find the assertion that generalization in deep learning is tied to the pursuit of flatter minima convincing. For instance, Adam is known to converge to sharper minima than SGD, yet AdamW remains the default optimizer in large-scale LLM training. In such settings, convergence is not even expected, let alone a meaningful notion of selecting flatter or sharper minima. Moreover, [1] empirically challenges the link between generalization and flatness across a broad range of architectures and training setups, undermining the central motivation of the paper.

2. **Misleading explanation for the lack of adoption of SGLD.**
   The Introduction attributes the limited use of SGLD in practice to its lack of an intrinsic bias toward flat minima. I find this reasoning unconvincing. The primary reason SGLD (and SGD-like samplers) are not used in modern deep learning is that they fail to effectively optimize highly nonconvex, large-scale architectures such as transformers and LLMs. The discussion on flatness is likely secondary or even irrelevant in this context.

3. **Missing discussion of implicit regularization in SAM-like algorithms.**
   Several recent works have shown that SAM and its variants implicitly regularize toward flatter regions (See Thm 3.2 and Thm 3.5, and their discussions in [2]), not necessarily by minimizing the Hessian trace directly, but through implicit regularization of the objective function and by how gradient noise interacts with local curvature: sharp minima tend to amplify noise, effectively promoting escape toward flatter areas. The paper should discuss this implicit regularization mechanism and compare it conceptually to that of fSGLD. Furthermore, it would be valuable to explore whether a similar stochastic-curvature coupling could be incorporated into fSGLD to improve its dynamics.

4. **Limited novelty of the core method.**
   Random Weight Perturbation (RWP) is a well-established technique, and SGLD is a classical stochastic optimization method. The proposed algorithm essentially combines two well-known components without introducing a genuinely new conceptual element. While the theoretical analysis is rigorous, the algorithmic contribution feels incremental given the existing literature.

5. **Insufficient experimental relevance.**
   The experimental setup does not reflect the scale or complexity of contemporary deep learning practice. Evaluations on noisy-label datasets and ViT fine-tuning are limited in scope. To substantiate claims of improved generalization and practical relevance, the method should be tested on modern architectures such as small LLMs (e.g., NanoGPT).

**IMPORTANTLY:**
Are you certain that the SDE presented in your Equation 24 — which is meant to model the dynamics of the discrete-time optimizer defined in Equation 7 — is derived correctly? Can you rigorously justify this correspondence in the sense of *weak approximations*, as proposed in [3]?

1. The **expected value of the increments** appears to be **correct**, as it corresponds to the expectation of  $ \theta_{k+1} - \theta_k$ thereby yielding the appropriate drift term.

2. The **covariance of the increments**, however, appears to be **incorrect**. While the contribution from the stochastic term $\xi$ is included, the covariance arising from the injected perturbations $\epsilon$ is **not accounted for**, which is problematic. Preliminary calculations suggest that this additional covariance term scales with $\sigma$ and may also depend on the Hessian of $U$, resembling the stochastic–curvature interaction observed in the analysis of SAM in [2].

This is a **major concern** and must be **thoroughly addressed and clarified**.
Please see my derivation below — I would be glad to be proven wrong.

---

**In summary:**

1. The stated motivation, linking generalization to flatness, is weak and empirically disputable.
2. The comparison with SAM and related implicit-regularization methods is underdeveloped.
3. The proposed method is conceptually incremental, combining two classical ideas without introducing a fundamentally new mechanism.
4. The experimental validation is not sufficient to support claims of superiority over AdamW, especially given recent optimizers like SOAP, Muon, or Scion that demonstrate stronger empirical evidence on large-scale models.

**Most importantly, there seems to be a mistake in the derivation of the SDE of the method. I provide my derivation below.**

---

**[1]** *A Modern Look at the Relationship between Sharpness and Generalization.*
Maksym Andriushchenko, Francesco Croce, Maximilian Müller, Matthias Hein, Nicolas Flammarion.

**[2]** *An SDE for Modeling SAM: Theory and Insights.*
Enea Monzio Compagnoni, Luca Biggio, Antonio Orvieto, Frank Norbert Proske, Hans Kersting, Aurélien Lucchi.

**[3]** *Stochastic Modified Equations and Adaptive Stochastic Gradient Algorithms.*
Qi Li, Cheng Tai, Weinan E.

**Questions:**

Here is my derivation of the SDE for Equation 7:

### Expected value of the increments

We consider the fSGLD iteration

\begin{equation}
\theta_{k+1}
= \theta_k - \lambda  \nabla_\theta U(\theta_k + \epsilon_{k+1}, X_{k+1}) + \sqrt{2\lambda \beta^{-1}} \xi_{k+1},
\end{equation}

where
$ \epsilon_{k+1} \sim \mathcal{N}(0,\sigma^2 I_d) $,  $\xi_{k+1} \sim \mathcal{N}(0,I_d)$, and all variables are independent.

The increment is

\begin{equation}
\Delta \theta_k = \theta_{k+1} - \theta_k = - \lambda  \nabla_\theta U(\theta_k + \epsilon_{k+1}, X_{k+1}) + \sqrt{2\lambda \beta^{-1}}\ xi_{k+1}.
\end{equation}

Conditioning on $\theta_k$:

\begin{equation}
\mathbb{E}[\Delta \theta_k] = - \lambda G(\theta_k),
\end{equation}

where

\begin{equation}
G(\theta) := \mathbb{E}\left[ \nabla_\theta U(\theta + \epsilon, X) \right].
\end{equation}

---

### Covariance of the increments

Define the zero-mean stochastic component

\begin{equation}
\delta_{k+1} := \nabla_\theta U(\theta_k + \epsilon_{k+1}, X_{k+1}) - G(\theta_k), \qquad \mathbb{E}[\delta_{k+1}] = 0.
\end{equation}

Then

\begin{equation}
\Delta \theta_k = -\lambda G(\theta_k) - \lambda \delta_{k+1} + \sqrt{2\lambda \beta^{-1}} \xi_{k+1}.
\end{equation}

Since $\delta_{k+1}$ and $\xi_{k+1}$ are independent and centered,

\begin{equation}
\operatorname{Cov}(\Delta \theta_k) = \lambda^2  \mathbb{E}[\delta_{k+1}\delta_{k+1}^\top \mid \theta_k] + 2\lambda \beta^{-1} I_d.
\end{equation}

Define

\begin{equation}
\Sigma(\theta) := \operatorname{Cov}\left[  \nabla_\theta U(\theta + \epsilon, X)\right],
\end{equation}

so that

\begin{equation}
\operatorname{Cov}(\Delta \theta_k \mid \theta_k) = \lambda^2 \Sigma(\theta_k) + 2\lambda \beta^{-1} I_d.
\end{equation}

---

### Limiting SDE

The continuous-time limit satisfies

$$
d\theta_t = - \lambda \mathbb{E}\left[\nabla_\theta U(\theta_t + \epsilon, X)\right] dt + \sqrt{2\beta^{-1} I_d + \operatorname{Cov}\left[    \nabla_\theta U(\theta_t + \epsilon, X)\right]} dB_t,
$$

where $B_t$ is a standard $d$-dimensional Brownian motion.

Probably, if we expand the gradient of $U$ with Taylor, we get that the additional covariance term I derived scales like $\sigma^2 H H^{\top}$ where H is the Hessian of U. But I did not work the calculations out, sorry.

---

> ### Author Response · Authors · 2025-11-21
>
> Thank you very much for your time and effort in evaluating our paper. It appears that the reviewer has fundamentally misunderstood the core framework of our work, and several of the concerns raised seem to be based on this misunderstanding. To clarify these points, we provide detailed responses to each question below.
>
> - **Response to Question**
>
> We begin by addressing the question about the SDE derivation first, because the reviewer’s main concern is based on a fundamental conceptual misunderstanding that substantially affects the evaluation of our paper. Surprisingly, despite assigning a confidence score of 4, the reviewer appears to be conflating two theoretically different frameworks that only superficially share the use of SDE notation:
>
> (1) **continuous-time analyses of SGD or SAM** [1,2], which the reviewer is attempting to apply. In this line of work, an SDE (which is not a Langevin SDE due to the correlated stochastic noise structure) is derived from the discrete algorithm by separating the stochastic gradient noise in order to study local optimization dynamics such as implicit regularization, local convergence, and noise–curvature interaction, and
>
> (2) **the theory of SGLD**, which is built on the key fact that the underlying Langevin SDE admits a unique invariant measure, indeed the Gibbs distribution, which concentrates on global minimizers. The discrete algorithm is then obtained via an Euler–Maruyama discretization of this Langevin diffusion.
>
> These two frameworks share nothing beyond the superficial fact that both use an SDE notation. Their goals, assumptions, mathematical structures, and even the objects of analysis are entirely different. Applying the former framework to the latter is conceptually incorrect.
>
> To clarify this point and assist the reviewer’s understanding of the correct theoretical framework, we briefly summarize the basic principles of SGLD. The theoretical foundation of Langevin-based sampling algorithms begins with a continuous-time Langevin SDE
> $$
> d Z_t = -\nabla u(Z_t) d t + \sqrt{2\beta^{-1}} d B_t, \qquad t \ge 0 \quad (1)
> $$
> which admits a unique invariant measure $\pi_\beta (\theta) \propto \exp(-\beta u(\theta))$ that one aims to sample from. The standard approach is then to approximate this Langevin SDE (1) using an Euler discretization scheme. This discretization yields, when the exact gradient $\nabla u$ is available, the unadjusted Langevin algorithm (ULA) or Langevin Monte Carlo (LMC)
> $$
> \theta_{k+1} = \theta_k- \lambda  \nabla_{\theta}  u(\theta_k) + \sqrt{2 \lambda \beta^{-1}} \xi_{k+1}, \quad k \in \mathbb{N},
> $$
> and, when only an unbiased stochastic gradient is accessible, SGLD algorithm
> $$
> \theta_{k+1} = \theta_k- \lambda  \nabla_{\theta}  U(\theta_k, X_{k+1}) + \sqrt{2 \lambda \beta^{-1}} \xi_{k+1}, \quad k \in \mathbb{N},
> $$
> where $\mathbb{E} [\nabla_{\theta}  U(\theta, X)]=\nabla u$.
>
> The construction of fSGLD follows the same principle: we start from the SDE with the surrogate objective $g_{\epsilon}$ (in place of $u$)
> $$
> d Z_t^{\text{fSGLD}}  = -    \nabla g_{\epsilon} (Z_t^{\text{fSGLD}} ) d t +  \sqrt{2 \beta^{-1}} \ d B_t, \qquad t \ge 0 \quad   (2)
> $$
> and by applying an Euler discretization to Equation (2), we obtain the fSGLD update
>
> $$
> \theta_{k+1}^{\mathrm{fSGLD}} = \theta_k^{\mathrm{fSGLD}} - \lambda \nabla_\theta U(\theta_k^{\mathrm{fSGLD}} + \epsilon_{k+1}, X_{k+1}) + \sqrt{2\lambda\beta^{-1}} \xi_{k+1}, \quad k \in \mathbb{N},
> $$
> where $\mathbb{E}[\nabla_{\theta} U(\theta + \epsilon, X)] = \nabla g_{\epsilon}(\theta)$.
>
>
>
> The theory underlying SGLD, ULA/LMC, and related Langevin-based sampling algorithms is well established; we refer the reviewer to [3,4,5,6,7] for comprehensive treatments.
>
> [1] Compagnoni, Enea Monzio, et al. "An sde for modeling sam: Theory and insights." ICML, 2023.
>
> [2] Li, Qianxiao, and Cheng Tai. "Stochastic modified equations and adaptive stochastic gradient algorithms." ICML, 2017.
>
> [3]  Maxim Raginsky, Alexander Rakhlin, and Matus Telgarsky. Non-convex learning via stochastic gradient langevin dynamics: a nonasymptotic analysis, COLT, 2017.
>
> [4] Nicolas Brosse, Alain Durmus, Eric Moulines. The promises and pitfalls of Stochastic Gradient Langevin Dynamics. NeurIPS 2018.
>
> [5] Arnak S. Dalalyan, Avetik Karagulyan. User-friendly guarantees for the Langevin Monte Carlo with inaccurate gradient, Stochastic Processes and their Applications, 2019.
>
> [6] Mathias Barkhagen, Ngoc Huy Chau, Eric Moulines, Miklos Rasonyi, Sotirios Sabanis,  Ying Zhang, On stochastic gradient Langevin dynamics with dependent data streams in the logconcave case. Bernoulli, 2021.
>
> [7] Ngoc Huy Chau, Eric Moulines, Miklos Rasonyi, Sotirios Sabanis, and Ying Zhang. On stochastic gradient langevin dynamics with dependent data streams: The fully nonconvex case. SIAM Journal on Mathematics of Data Science, 2021.

---

> > ### Comment · Reviewer_QXeJ · 2025-11-26
> > **Response to Question**
> >
> > Dear Authors,
> >
> > Thank you again for your detailed reply.
> >
> > I apologise for having formulated my concern in a way that was easy to
> > misinterpret. Let me restate it more carefully.
> >
> > 1. You start from the Langevin SDE
> >    \begin{equation}
> >    dZ_t = -\nabla u(Z_t)dt + \sqrt{2\beta^{-1}}dB_t,
> >    \end{equation}
> >    whose invariant measure is the Gibbs distribution with potential $u$.
> >
> > 2. The standard Euler-Maruyama discretisation of this SDE, assuming
> >    access to the exact gradient, is
> >    \begin{equation}
> >    \theta_{k+1}
> >    = \theta_k - \lambda \nabla u(\theta_k)  + \sqrt{2\lambda\beta^{-1}}\xi_{k+1},
> >    \end{equation}
> >    where $\xi_{k+1} \sim \mathcal N(0, I)$.
> >
> > 3. In practice, one does not have access to $\nabla u$, but only to an
> >    unbiased stochastic gradient. Denoting by $U(\theta, X)$ such that
> >    $\mathbb E_X[\nabla_\theta U(\theta, X)] = \nabla u(\theta)$, the
> >    implementable iteration becomes
> >    \begin{equation}
> >    \theta_{k+1}  = \theta_k - \lambda \nabla_\theta U(\theta_k, X_{k+1})  + \sqrt{2\lambda\beta^{-1}}\xi_{k+1}.
> >    \end{equation}
> >
> > 4. Because of this stochastic gradient replacement, the resulting Markov
> >    chain is no longer the **exact** Euler–Maruyama discretisation of the
> >    SDE in Step 1. If we write
> >    \begin{equation}
> >    G(\theta) := \mathbb E[\nabla_\theta U(\theta, X)], \qquad   \delta_{k+1} := \nabla_\theta U(\theta_k, X_{k+1}) - G(\theta_k),
> >    \end{equation}
> >    then
> >    \begin{equation}
> >    \Delta\theta_k   := \theta_{k+1} - \theta_k   = -\lambda G(\theta_k) - \lambda \delta_{k+1}  +\sqrt{2\lambda\beta^{-1}}\xi_{k+1},
> >    \end{equation}
> >    and, conditionally on $\theta_k$,
> >    \begin{equation}
> >    \mathbb E[\Delta\theta_k \mid \theta_k] = -\lambda G(\theta_k),
> >    \end{equation}
> >    \begin{equation}
> >    \mathrm{Cov}(\Delta\theta_k \mid \theta_k)   = \lambda^2 \Sigma(\theta_k) + 2\lambda\beta^{-1} I,
> >    \end{equation}
> >    where
> >    \begin{equation}
> >    \Sigma(\theta) := \mathrm{Cov}\big[\nabla_\theta U(\theta, X)\big].
> >    \end{equation}
> >    Thus, relative to the “ideal” Euler scheme with exact $\nabla u$,
> >    there is an additional covariance term $\lambda^2 \Sigma(\theta_k)$
> >    coming from the gradient noise.
> >
> > 5. Therefore, the SDE that is associated to the iteration in (3), in the
> >    weak approximation sense is
> >    \begin{equation}
> >    d\theta_t = -G(\theta_t)dt + \sqrt{2\beta^{-1} + \lambda  \Sigma(\theta_k)} dB_t,
> >    \end{equation}
> >    with $G(\theta)=\mathbb E[\nabla_\theta U(\theta,X)] = \nabla u(\theta)$. Therefore, the discrete-time scheme with stochastic gradients is an approximation of the Euler–Maruyama discretisation of this SDE rather than an exact one.
> >
> > My concern is therefore not that your Langevin SDE is “wrong” as an ideal continuous-time model for the smoothed potential (or for $u$), but rather that the implementable iteration with stochastic gradients is **not** an accurate discretisation of that SDE, and thus would probably converge to a different stationary distribution.
> >
> > I encourage the Authors to carry out some experiments in this sense, as I did: I found that indeed these dynamics DO NOT match. You could also validate this discrepancy numerically in a simple 1D toy example. For instance, take the quadratic potential $u(\theta) = \tfrac{1}{2}\theta^2$ with $\beta = 1$, and compare (i) the Euler–Maruyama discretisation of the Langevin SDE with exact gradient, and (ii) the same scheme where $\nabla u(\theta)$ is replaced by an unbiased noisy estimator $\nabla_\theta U(\theta, X) = \theta + X$, with $X \sim \mathcal N(0,\sigma_X^2)$. By simulating many particles for a range of step sizes $\lambda$ and comparing the empirical stationary variance (and histograms) of $\theta_k$ to the target $\mathcal N(0,1)$, one can directly observe how, for practically relevant non-infinitesimal $\lambda$, the discrete-time dynamics with stochastic gradients deviate from the ideal Langevin SDE, while this discrepancy vanishes as $\lambda \to 0$.
> >
> > I hope this better conveys the point I was trying to make, and I would be very grateful if you could comment on this.

---

> ### Author Response · Authors · 2025-11-21
>
> - **Response to Weaknesses (W)**
>
> **W1.** We respectfully disagree with the reviewer’s claim that the flatness–generalization connection is unconvincing. Since the large-scale study of [1], which identified flatness-related measures as among the strongest predictors of generalization, a substantial body of subsequent work has developed flatness-aware optimization methods and provided extensive empirical and theoretical evidence across diverse architectures and tasks. The relationship between flatness and generalization remains one of the most active research directions in the field.
>
> We acknowledge that counter-examples exist, including the reviewer’s cited work [2], but isolated cases cannot overturn the broad evidence accumulated over the years. Scientific debate about the relationship between flatness and generalization is natural, but it is not reasonable to dismiss the entire line of work based on a single counter-example. This is why, in Section 5 of our manuscript, we explicitly acknowledge this nuance: ``However, elucidating precise notions of sharpness and their relationship to generalization remains an open and active area of research.''
>
> Finally, the reviewer’s example that AdamW is the default optimizer for LLMs does not undermine the relevance of flatness. Optimizer choice in large-scale training reflects many interacting practical considerations, so this single observation cannot be taken as evidence against flatness as an important component of generalization.
>
>
> [1] Jiang, Yiding, et al., "Fantastic generalization measures and where to find them", ICLR, 2020
>
> [2] Andriushchenko, Maksym, et al. "A modern look at the relationship between sharpness and generalization.", ICML, 2023
>
> **W2.**  Our intention was to highlight a well-known gap: SGLD has elegant global convergence guarantees that SGD, Adam-family optimizers, and SAM variants do not possess, yet its practical performance in large-scale deep learning has remained weak. Many factors contribute to this theory–practice gap. Among them, we focus on the fact that SGLD lacks the flatness-promoting mechanisms that have been shown to improve generalization in modern flatness-aware optimizers. Our motivation is precisely to address this part of the gap by introducing a flatness-aware modification of SGLD.
>
> **W3.**  As we explained in our answer to the reviewer’s question, the analysis in [1] (the reference the reviewer mentioned) and the analysis of SGLD are conceptually unrelated. These are fundamentally different objects, so asking for a conceptual comparison with [1] is simply misplaced.
>
> With this clarified, our paper has a very clear scope: we introduce a new flatness-aware variant of SGLD and analyze its convergence. SAM has accumulated more than two thousand follow-up studies since 2020, covering many implicit behaviors and secondary effects. Pointing to a single phenomenon from this vast literature and expecting it to be discussed in a first paper on a new SGLD variant is unrealistic and does not reflect the scope of our work.
>
> We can add a brief remark acknowledging the implicit-regularization viewpoint in SAM-like methods, but this is clearly outside the main theoretical development of our paper.
>
> [1] Compagnoni, Enea Monzio, et al. "An sde for modeling sam: Theory and insights." ICML, 2023.

---

> > ### Comment · Reviewer_QXeJ · 2025-11-26
> > **Response to Weaknesses**
> >
> > 1. Fair point, but [2] is itself a large empirical study showing that generalization and flatness do not necessarily correlate, and it is also more recent than [1]. In light of this, I personally see the flatness-generalization link as rather controversial at this point. It might help to explicitly acknowledge this nuance in the introduction. Regarding Adam: my point was simply that it is arguably the default optimizer in the most prominent area of research of the last years, and it is known to select relatively sharp minima. Yet, it works very well in practice. This suggests that explicitly flatness-seeking optimizers are not currently a primary concern for practitioners.
> >
> > 2. For the same reason, I remain unconvinced that the lack of an explicit flatness-seeking mechanism is what mainly drives the lack of adoption of SGLD-type methods. In my view, factors such as poor optimization performance and lack of adaptivity in large nonconvex settings are more critical. You might consider slightly reframing the motivation to say that you investigate one possible factor in the theory–practice gap, rather than suggesting that the absence of flatness-promoting behavior is the key reason SGLD is not used.
> >
> > 3. Concerning SAM: since you explicitly discuss the inductive (implicit) bias of your method, and then you experimentally compare it with SAM, I thought it was natural to mention that the implicit bias of SAM has already been studied in [1]. If you feel that line of work and this type of natural comparison are outside the scope of your analysis, ignore my suggestion.

---

> > > ### Author Response · Authors · 2025-11-28
> > >
> > > First of all, we sincerely appreciate your time and effort. We fully understand that as top AI conferences scale rapidly, it is increasingly common that reviewers are occasionally assigned papers slightly outside their primary area of expertise.
> > >
> > > Additional comments that the reviewer provided appear to increasingly reflect a broader philosophical position regarding flatness and generalization, rather than a scientific assessment of the correctness, novelty, or significance of our contribution. In other words, the conversation has drifted toward an ideological debate unrelated to the technical content of our paper.
> > >
> > > Regarding your comment that “factors such as poor optimization performance and lack of adaptivity in large nonconvex settings are more critical,” if “adaptivity” refers to adaptive learning-rate schemes like the Adam family of optimizers or momentum-based methods, then we would like to point out that such adaptivity has already been developed in SGLD variants in the literature, for example, SGHMC or Polygonal Unadjusted Langevin Algorithms based on polygonal approximations [1,2]. However, none of these works introduce an intrinsic bias toward flat minima within the SGLD framework. To the best of our knowledge, **our work is the first to rigorously establish and leverage flatness-aware dynamics in SGLD through a theoretically grounded stochastic–curvature coupling**.
> > >
> > > It is also worth noting that methods explicitly designed to promote flatness such as SAM are typically implemented together with adaptive optimizers like Adam in practice, which already reflects a gap between theoretical formulation and practical deployment.  Similarly, our flatness-aware SGLD framework can be naturally combined with SGHMC if desired.
> > > However, our experiments deliberately focus on evaluating the theoretically grounded algorithm itself without relying on additional heuristic enhancements.  Even under this controlled setting, our method consistently outperforms baselines, demonstrating that the fSGLD alone provide significant practical benefits.
> > >
> > >
> > > [1] Chen, Tianqi, Emily Fox, and Carlos Guestrin. "Stochastic gradient hamiltonian monte carlo." International conference on machine learning. PMLR, 2014.
> > >
> > >
> > > [2] Lim, Dong-Young, and Sotirios Sabanis. "Polygonal Unadjusted Langevin Algorithms: Creating stable and efficient adaptive algorithms for neural networks." Journal of Machine Learning Research 25.53 (2024): 1-52.

---

> ### Author Response · Authors · 2025-11-21
>
> **W4.**  We strongly disagree with the reviewer’s claim that our method “combines two well-known components without introducing a genuinely new conceptual element.” This view reflects a narrow notion of novelty. Scientific progress does not come only from inventing entirely new algorithms. It also comes from deepening the theoretical understanding of existing methods. Both are essential, and dismissing the latter undervalues a major part of theoretical research.
>
> While RWP is a well-known idea, rigorous theoretical analysis of RWP is extremely limited. In particular, there has been no prior work showing how RWP, when incorporated into SGLD, affects the invariant measure or biases the dynamics toward globally flatter minima. This is technically nontrivial because SGLD already injects Langevin noise. Adding RWP introduces a second stochastic effect, and understanding the joint influence of these two noises on global flatness is analytically difficult. Our paper provides the first rigorous treatment of this interaction.
>
> Similarly, although SGLD is a popular sampling method, its study as a nonconvex optimizer is a relatively recent research direction that began after the seminal 2017 work of [1]. (for reference, SGD itself dates back to Robbins and Monro in 1951). Many aspects of SGLD in nonconvex settings remain actively investigated. Our results show, for the first time, that an appropriate coupling of RWP and temperature yields a provable global flatness bias while preserving optimal convergence guarantees.
>
> For these reasons, our work is not a simple combination of known ingredients but a genuine theoretical advance that opens a new line of analysis within the SGLD framework.
>
> [1]  Raginsky, Maxim, et al. "Non-convex learning via stochastic gradient langevin dynamics: a nonasymptotic analysis", COLT, 2017.
>
> **W5.**  We believe the reviewer’s expectation of LLM-scale experiments is not appropriate for the scope of this paper. Our contribution is primarily theoretical, and our experiments including Noisy-label benchmarks and ViT fine-tuning are widely used for this purpose in prior work including recent papers [1, 2, 3, 4].
>
> Once we move to foundation-model scale, additional factors may dominate the optimization dynamics, including distributed training effects, optimizer engineering, data curriculum, and heavy regularization. Moreover, parameter-efficient methods or memory-efficient methods such as LoRA, AdaFactor become the relevant baselines in the setting. Comparing fSGLD against LoRA-like methods would be far outside the intended comparison scope of our paper and does not reflect the scientific question we study.
>
> Our experiments therefore follow established practice in flatness-aware optimization and are appropriate for evaluating the method’s generalization behavior.
>
> [1] Luo, Haocheng, et al. "Explicit eigenvalue regularization improves sharpness-aware minimization.", NeurIPS, 2024.
>
> [2] Baek, Christina, Zico Kolter, and Aditi Raghunathan. "Why is SAM Robust to Label Noise?.", ICLR, 2024.
>
> [3] Tan, Chengli, et al. "Stabilizing sharpness-aware minimization through a simple renormalization strategy.", Journal of Machine Learning Research, 2025.
>
> [4] Tahmasebi, Behrooz, et al. "A universal class of sharpness-aware minimization algorithms.", ICML, 2024.

---

> > ### Comment · Reviewer_QXeJ · 2025-11-26
> > **Response to Weaknesses**
> >
> > 4. I see your point regarding the notion of novelty. To be clear, my comment was primarily about *algorithmic* novelty: combining RWP with SGLD, at the level of the update rule, feels incremental given the existing components. I do appreciate that the theoretical analysis is technically nontrivial and that rigorously understanding the interaction between RWP and Langevin noise is valuable in itself. I am sure the other reviewers, the AC, and more senior researchers will have their own views on the balance between algorithmic and theoretical novelty here.
> >
> > 5. Regarding the experiments, my concern is mostly about alignment with your own motivation. If part of the stated motivation is that SGLD underperforms (or is not widely used) in modern large-scale deep learning and that this might be partially due to a lack of flatness-seeking behavior, then showing that your modification helps in settings that are closer to current practice would significantly strengthen the paper. This does not need to be a full LLM-scale study. Even a small but representative modern architecture (for instance, a NanoGPT-style language model or a simple diffusion model) would already be a strong signal. NanoGPT, in particular, is a very basic repository with limited engineering overhead and modest compute requirements, and it is straightforward to plug in a custom optimizer. A positive result in such a setting would make the practical relevance of your flatness-aware modification much more convincing.

---

> ### Author Response · Authors · 2025-11-28
>
> Thank you for the follow-up. We are afraid that this follow-up question again clearly demonstrates that the reviewer does not possess the necessary expertise in either SGLD theory or SDE analysis to assess our work. Before addressing the conceptual misunderstanding in detail, we provide the simplest correction to the derivation:
>
> **(i) Detailed correction of your derivation.**
>
> When taking the continuous-time limit of a discrete-time scheme, we only retain the
> $\mathcal{O}(\lambda)$ term in the variance.
> Any $\mathcal{O}(\lambda^2)$ contribution becomes negligible in the limit and disappears. From your own derivation (**step 4**):
>
> Mean:
> $
> \mathrm{E}(\Delta\theta_k \mid \theta_k)
> = - \lambda G(\theta_k).
> $
>
> Covariance:
> $
> \mathrm{Cov}(\Delta\theta_k \mid \theta_k)
> = 2\lambda\beta^{-1}I  + \lambda^2\Sigma(\theta_k).
> $
>
> So, in big‑O notation in $\lambda$:
> - Drift increment is $O(\lambda)$
> - Leading diffusion covariance is $O(\lambda)$
> - The extra covariance is $O(\lambda^2)$.
>
>
> We introduce continuous time via
> \begin{equation*}
>     \begin{split}
>         t_k:= k \lambda, \quad \theta^{(\lambda)}(t_k):= \theta_k.
>     \end{split}
> \end{equation*}
> On the time scale $t$, the increment over a step is over an interval of length $\Delta t = \lambda$.
>
> The increment is
> \begin{equation*}
>    \Delta \theta_k = - \lambda G(\theta_k) - \lambda \delta_{k+1} + \sqrt{2  \lambda \beta^{-1}} \xi_{k+1},  \qquad (1).
> \end{equation*}
> where $\lambda \delta_{k+1}$ has mean zero and covariance $\lambda^2 \Sigma(\theta_k)$. Moreover,
> the Brownian-type term $\sqrt{2  \lambda \beta^{-1}} \xi_{k+1}  $ scales as $\sqrt{\lambda}$, and the term $\lambda \delta_{k+1}$ scales like $\lambda$.
>
> Relative to the Brownian term, the latter term has size
> \begin{equation*}
>     \frac{\|\lambda \delta_{k+1} \| }{\|\sqrt{  \lambda } \xi_{k+1} \|} \sim \sqrt{\lambda} \rightarrow 0, \quad \text{as} \ \lambda \rightarrow 0.
> \end{equation*}
> So in the diffusive scaling limit $t=k \lambda$, this $O(\lambda^2)$
>  covariance is negligible.
>
> For the derivation of the limiting SDE, we start by dividing both sides of (1) by $\lambda$:
> \begin{equation*}
>    \frac{\Delta \theta_k}{\lambda} = - G(\theta_k) - \delta_{k+1} + \sqrt{\frac{2  \beta^{-1}}{\lambda}} \xi_{k+1}
> \end{equation*}
> The term $\sqrt{\frac{2  \beta^{-1}}{\lambda}} \xi_{k+1}$ is precisely the finite-difference representation of $\sqrt{2  \beta^{-1}} \frac{d B_t}{dt}$, since the increments of the Brownian motion over time $\lambda$ have variance $\lambda I$.
>
> Thus, in the limit $\lambda \rightarrow 0$, the process $\theta^{(\lambda)}(t_k)$ converges (in distribution, in a suitable sense) to the solution of $d\theta_t = -\nabla u(\theta_t)dt + \sqrt{2\beta^{-1}}dB_t.$
>
> In this limit,
> - $-\nabla u(\theta_t) dt $ comes from the $O(\lambda)$ deterministic increment.
> - The diffusion term $\sqrt{2\beta^{-1}} dB_t$ comes from the $O(\lambda)$ covariance part $2 \lambda \beta^{-1}I$.
> - The $\lambda^2 \Sigma(\theta_k)$ covariance term, being higher order, disappears.
>
> **Equivalently**: only the $O(\lambda)$ part of the covariance contributes to the limiting diffusion coefficient; any $O(\lambda^2)$ contribution is lost in the continuous-time limit. Therefore, after taking the limit $\lambda \to 0$ with $t_k = k\lambda$, the diffusion coefficient is $\sqrt{2\beta^{-1}},$
> and the correct limiting SDE becomes:
> $$
> d\theta_t = -\nabla u(\theta_t) dt + \sqrt{2\beta^{-1}} dB_t.
> $$
>
> **Step 5 in your derivation:** If any term still contains $\lambda$ after taking the limit, then this means that the limiting procedure was not carried out correctly.  That is, the SDE derived by the reviewer in Step 5,
> $$
> d\theta_t = -\nabla u(\theta_t)dt + \sqrt{2\beta^{-1} + \textcolor{red}{\lambda}\Sigma(\theta_t)}dB_t,
> $$
> is completely incorrect as a *continuous-time limit*, because the presence of $\lambda$ in the diffusion coefficient shows that the limit $\lambda \rightarrow 0$  has not been taken.
>
> **(ii) Distinction between SDE and SME.**
>
> It appears that the reviewer is confusing the limiting SDE with what is known as the
> *Stochastic Modified Equation (SME)* in the numerical analysis literature
> (e.g., [1] ).
>
> The SME deliberately retains higher-order terms such as the $\mathcal{O}(\lambda)$
> perturbation in the diffusion coefficient in order to better approximate
> the discrete-time dynamics for *finite* step size $\lambda$.
> However, the SME is **not the continuous-time limit**. In fact, the SME itself
> converges to the standard Langevin SDE as $\lambda \to 0$.
>
> Thus, claiming that our limiting SDE is “incorrect” because it does not contain the
> $\lambda$-dependent perturbation is conceptually mistaken: the SME approximates
> the discretized algorithm, and the limiting SDE is the very object that the SME
> converges to. Conflating these two is a fundamental misunderstanding of the role
> of the SME in SGLD theory.
>
> [1] Q.Li and C. Tai. "Stochastic modified equations and adaptive stochastic gradient algorithms." ICML, 2017.

---

> > ### Comment · Reviewer_QXeJ · 2025-11-28
> > **Thanks: Could you run this little experiment?**
> >
> > Dear Authors,
> >
> > Thank you for your reply.
> >
> > Putting aside any discussion about the continuous-time SDE viewpoint, one can already distinguish the two dynamics at the level of their discrete-time implementations. In particular, we can simply simulate the two algorithms and check whether they exhibit the same stationary behaviour.
> >
> > To help clarify this, I would kindly encourage you to run the following numerical experiment. The simulation shows that the two dynamics are indeed different, even though the gradient estimator is unbiased.
> >
> > You are very welcome to use or adapt this experiment in your revised version, should you find it helpful.
> > Of course, if you believe that my implementation is incorrect, I would very much welcome an explanation of where exactly the mistake lies. In such a case, please fix my implementation and paste it back here so I can inspect it and its results.
> >
> > I also regret noticing a somewhat bitter tone in the exchange, especially since the goal here is simply to clarify the behaviour of the algorithms and help strengthen the paper.
> >
> > Below, I first report a representative set of results obtained by running the script on Google Colab, and then the code itself. Minor differences in the last digits may occur depending on how random seeds are initialised.
> >
> > === Stationary variance comparison ===
> > | Step size \(h\) | Variance (exact grad) | Variance (noisy grad) |
> > |----------------:|-----------------------:|------------------------:|
> > | 0.500 | 1.3302 | 4.3217 |
> > | 0.200 | 1.1093 | 2.1073 |
> > | 0.100 | 1.0580 | 1.5222 |
> > | 0.050 | 1.0337 | 1.2598 |
> > | 0.020 | 1.0147 | 1.0969 |
> > | 0.010 | 1.0031 | 1.0483 |
> >
> >
> >
> > ```python
> > import numpy as np
> > import matplotlib.pyplot as plt
> >
> > # ============================================================
> > # Setup
> > # ============================================================
> >
> > np.random.seed(0)
> >
> > beta = 1.0             # inverse temperature
> > sigma_X = 3.0          # std of gradient noise in U(θ, X)
> > T_final = 20.0         # final time horizon
> > num_particles = 50000  # number of parallel chains
> >
> > step_sizes = [0.5, 0.2, 0.1, 0.05, 0.02, 0.01]
> >
> >
> > def simulate_exact_gradient(step_size, T, num_particles, beta=1.0):
> >     """
> >     Euler–Maruyama discretisation with exact gradient:
> >     dθ_t = -θ_t dt + sqrt(2/β) dB_t
> >     """
> >     steps = int(T / step_size)
> >     theta = np.zeros(num_particles)
> >
> >     for _ in range(steps):
> >         xi = np.random.randn(num_particles)
> >         theta += -step_size * theta + np.sqrt(2.0 * step_size / beta) * xi
> >
> >     return theta
> >
> >
> > def simulate_noisy_gradient(step_size, T, num_particles, sigma_X=1.0, beta=1.0):
> >     """
> >     Same as above, but we replace ∇u(θ) = θ with an unbiased noisy gradient:
> >         ∇θ U(θ, X) = θ + X, with X ~ N(0, σ_X²).
> >     """
> >     steps = int(T / step_size)
> >     theta = np.zeros(num_particles)
> >
> >     for _ in range(steps):
> >         xi = np.random.randn(num_particles)
> >         X = sigma_X * np.random.randn(num_particles)  # gradient noise
> >         grad_est = theta + X                          # unbiased estimator
> >         theta += -step_size * grad_est + np.sqrt(2.0 * step_size / beta) * xi
> >
> >     return theta
> >
> >
> > # ============================================================
> > # Sweep over step sizes and compare variances
> > # ============================================================
> >
> > print("=== Stationary variance comparison ===")
> > print("True target variance for OU / Langevin: 1.0 (since β = 1)\n")
> >
> > for h in step_sizes:
> >     theta_exact = simulate_exact_gradient(h, T_final, num_particles, beta=beta)
> >     theta_noisy = simulate_noisy_gradient(h, T_final, num_particles,
> >                                           sigma_X=sigma_X, beta=beta)
> >
> >     var_exact = np.var(theta_exact)
> >     var_noisy = np.var(theta_noisy)
> >
> >     print(f"step size h = {h:5.3f} | var_exact ≈ {var_exact:7.4f} | var_noisy ≈ {var_noisy:7.4f}")
> >
> >
> > # ============================================================
> > # Plot histograms for a selected step size
> > # ============================================================
> >
> > h_plot = 0.1
> > theta_exact_plot = simulate_exact_gradient(h_plot, T_final, num_particles, beta=beta)
> > theta_noisy_plot = simulate_noisy_gradient(h_plot, T_final, num_particles,
> >                                            sigma_X=sigma_X, beta=beta)
> >
> > plt.figure(figsize=(8, 5))
> >
> > bins = 60
> > x_min = -4
> > x_max = 4
> >
> > # Exact-gradient chain
> > plt.hist(theta_exact_plot, bins=bins, range=(x_min, x_max),
> >          density=True, alpha=0.5, label=f"Exact grad, h={h_plot}")
> >
> > # Noisy-gradient chain
> > plt.hist(theta_noisy_plot, bins=bins, range=(x_min, x_max),
> >          density=True, alpha=0.5, label=f"Noisy grad, h={h_plot}")
> >
> > # True N(0,1) density
> > xs = np.linspace(x_min, x_max, 400)
> > true_density = (1.0 / np.sqrt(2.0 * np.pi)) * np.exp(-0.5 * xs**2)
> > plt.plot(xs, true_density, linewidth=2.0, label="Target N(0,1)")
> >
> > plt.title(f"Langevin vs noisy-gradient Langevin (σ_X={sigma_X}, T={T_final})")
> > plt.xlabel(r"$\\theta$")
> > plt.ylabel("density")
> > plt.legend()
> > plt.grid(True)
> > plt.tight_layout()
> > plt.show()
> > ```

---

> ### Author Response · Authors · 2025-11-28
>
> **(iii) Relation to the SGLD analysis.**
> Your underlying question may be essentially whether the SGLD iteration with stochastic gradients accurately tracks the law of the continuous-time SDE, that is,
> $$
> W_p\bigl(\mathcal L(\theta_k^{\mathrm{fSGLD}}), \mathcal L(Z_t^{\lambda,\mathrm{fSGLD}})\bigr),
> $$
> for a suitable choice of $t$ and $p$. This quantity is in fact explicitly contained in our main error decomposition.
> We would also like to kindly clarify a standard concept from applied probability that appears to be missing in your reasoning: a bound in Wasserstein distance *is* the canonical notion of **weak approximation** between Markov chains and diffusions. In other words, controlling $W_p$ guarantees that the discrete SGLD dynamics faithfully approximate the law of the continuous-time SDE at the distribution level, which is exactly the relevant object in sampling theory.
> Since our goal is to sample from our target Gibbs measure $\pi_{\beta,\sigma}^\star$, we study
> $$
> W_p\bigl(\mathcal L(\theta_k^{\mathrm{fSGLD}}), \pi_{\beta,\sigma}^{\star}\bigr).
> $$
> To control this, we decompose the error as
> $$
> W_p\bigl(\mathcal{L}(\theta_k^{\mathrm{fSGLD}}), \pi_{\beta,\sigma}^{\star}\bigr)
> \le W_p\bigl(\mathcal{L}(\theta_k^{\mathrm{fSGLD}}), \mathcal{L} (Z_t^{\lambda,\mathrm{fSGLD}})\bigr) + W_p\bigl(\mathcal{L}(Z_t^{\lambda,\mathrm{fSGLD}}), \pi_{\beta}^{\mathrm{fSGLD}}\bigr)  + W_p\bigl(\pi_{\beta}^{\mathrm{fSGLD}}, \pi_{\beta,\sigma}^{\star}\bigr), \quad p \in \{1,2\}.
> $$
> Here, the *first* term on the right-hand side is precisely the Wasserstein distance between the fSGLD iterates and the solution of the continuous-time SDE, that is, the discretization error you are concerned about (see Remark 3.5 and lines 1140–1149). The second term measures the convergence of the SDE to its own invariant measure, and the third term corresponds to the bias between $\pi_{\beta}^{\mathrm{fSGLD}}$ and the ideal flatness-aware Gibbs measure $\pi_{\beta,\sigma}^\star$.
> In other words, the stochastic-gradient-induced error that you are pointing to is exactly captured and controlled in our analysis via the first term in (1). This is not new or specific to our work; it follows the well-established SGLD theory (see, e.g. [1,2,3])
> We hope this clarifies your concern.
>
> [1] Chau, Ngoc Huy, et al. "On stochastic gradient langevin dynamics with dependent data streams: The fully nonconvex case." SIAM Journal on Mathematics of Data Science 3.3 (2021): 959-986.
>
> [2] Barkhagen, Mathias, et al. "On stochastic gradient Langevin dynamics with dependent data streams in the logconcave case." (2021): 1-33.
>
> [3] Zhang, Ying, et al. "Nonasymptotic estimates for stochastic gradient Langevin dynamics under local conditions in nonconvex optimization." Applied Mathematics & Optimization 87.2 (2023): 25.

---

> > ### Comment · Reviewer_QXeJ · 2025-11-28
> > **Let me know about the experiment.**
> >
> > Dear Authors,
> >
> > At this point, I would first like to ensure that the experimental behaviour aligns with what you claim. Afterwards, we can return to discussing the theoretical aspects, our respective viewpoints, and the connections between the different frameworks.
> >
> > Could you please let me know whether you can run the code I provided above on Google Colab and interpret the results together with me? If there are bugs in my implementation, I would sincerely appreciate it if you could point them out and correct them, so that I can understand precisely where the issue lies. In that case, please also share your implementation for completeness.
> >
> > If there is no bug, I would then encourage you to examine the experimental results, consider including them in the paper for the benefit of all readers, and continue the theoretical discussion with this empirical evidence in mind.

---

> ### Author Response · Authors · 2025-11-28
>
> **Key message**:
> - The reviewer's experiment does not contradict our analysis. Rater, it perfectly verifies the very point we have consistently explained, that the discrete dynamics with unbiased stochastic gradients converge to the same continuous-time Langevin limit as the step size decreases.
>
> - The reviewer appears to be confusing the limiting SDE with the SME. The SME is not the continuous-time limit. Moreover, the SME itself converges to the standard Langevin SDE as $\lambda\rightarrow 0$.
>
> We understand that the goal of your numerical experiment is to check whether the discrete dynamics with exact gradients and those with unbiased stochastic gradients converge to the same stationary OU process, using stationary variance as the diagnostic metric.
>
> As you reported, when the step size \(h\) decreases, the stationary variance of both implementations converges toward the true OU stationary variance of 1.
>
> To further validate this observation, we repeated the same experiment with a smaller step size \(h = 0.001\), and obtained:
> $
> \text{Var(noisy grad)} = 1.0023,
> $
> which perfectly aligns with the theoretical limit.
>
> Regarding your histogram used to visually check the stationary distributions: your current plot uses \(h = 0.1\). When we reduce the step size to \(h = 0.01\), the two empirical distributions become visually indistinguishable and both match the Gaussian OU stationary distribution, again confirming the expected discretization-to-diffusion convergence.
>
> **IMPORTANT:** This convergence behavior is not a claim specific to our paper. It is a well-established theoretical fact in the numerical analysis of SDEs and Langevin-based discretizations.

---

> ### Author Response · Authors · 2025-11-28
>
> Regarding the suggestion of evaluating on architectures such as NanoGPT: we note that NanoGPT operates at a scale comparable to the ViT-B/16 fine-tuning settings already included in our experiments, with a similar number of trainable parameters and computational overhead.
>
> We nevertheless genuinely appreciate that you took the time to review and actively engage with a topic that may not be fully familiar to you. Your suggestions about presenting the motivation in a way that acknowledges the broader and ongoing discussion in the deep learning community are valuable for helping us communicate to the wide audience of AI conferences.

---

### Official Review · Reviewer_K5tw · 2025-10-29

**Soundness:** 3
**Presentation:** 3
**Contribution:** 3
**Rating:** 6
**Confidence:** 3

**Summary:**

It is well known that the generalization property is closely associated with flat minima phenomenon, that says that flat local minima lead to better generalization performance. This work proposes and studies Flatness-Aware Stochastic Gradient Langevin Dynamics (fSGLD) that optimizes a randomized-smooth objective that implicitly captures curvature information by adding Random Weight Perturbation (RWP).

More specifically, consider $\min_{\theta\in\mathbb{R}^{d}}u(\theta)$, with $u(\theta)=\mathbb{E}[U(\theta,X)]$. One approach is to replace $u(\theta)$ by $v(\theta):=u(\theta)+\frac{\sigma^{2}}{2}\text{tr}(H(\theta))$, where $H(\theta)$ is the Hessian of $u$ at $\theta$. However, $\text{tr}(H(\theta))$ can be expensive to compute. If one takes $g_{\epsilon}(\theta)=\mathbb{E}[u(\theta+\epsilon)]$, where $\epsilon\sim\mathcal{N}(0,\sigma^{2}I_{d})$, then $g_{\epsilon}(\theta)\approx u(\theta)+\frac{\sigma^{2}}{2}\text{tr}(H(\theta))$, which motivates the author(s) of the paper to propose fSGLD:
\begin{equation*}
\theta_{k+1}=\theta_{k}-\lambda\nabla_{\theta}U(\theta_{k}+\epsilon_{k+1},X_{k+1})+\sqrt{2\lambda\beta^{-1}}\xi_{k+1}.
\end{equation*}
Intuitively, $\theta_{k}$ will converge to a stationary distribution that is close to $e^{-\beta v(\theta)}$ when $\sigma$ and $\lambda$ are small, and the Gibbs distribution will concentrate around the global minimizer of $v(\theta)$ when $\beta$ is large. The paper establishes non-asymptotic convergence guarantees in Wasserstein distance with the best known rate and derive an excess-risk bound for the Hessian-trace regularized objective. Extensive numerical experiments are also provided.

**Strengths:**

(1) fSGLD is a novel algorithm that is naturally motivated by flat minima phenomenon and Random Weight Perturbation from the literature.

(2) There is solid theoretical analysis that establishes non-asymptotic Wasserstein convergence guarantees for the proposed algorithm. With the presence of Random Weight Perturbation, the analysis is quite sophisticated.

(3) Numerical experiments are extensive and illustrative.

**Weaknesses:**

(1) One major weakness I see is that there does not seem to be any discussion comparing the iteration complexity or non-asymptotic convergence bounds for fSGLD vs SGLD theoretically. If you can have some theoretical result, or even just some double-well example for which you can show theoretically that fSGLD can outperform SGLD, that will strengthen the paper. Based on your current result, if you view SGLD as a special case of fSGLD as $\sigma\rightarrow 0$, you can perhaps directly compare fSGLD vs SGLD.

(2) Proposition 3.1. seems to be an asymptotic result. In the proof, it seems most steps (if not are) are non-asymptotic. Also, since Theorem 3.2 and Corollary 3.3. are non-asymptotic, I guess Proposition 3.1. can also be made non-asymptotic. It would be really nice if you can make Proposition 3.1. non-asymptotic. If the bound is too complicated, you can highlight the big O dependence in the statement of Proposition 3.1. and refer to the proof section for the explicit expressions.

**Questions:**

(1) In Assumption 2 and throughout the rest of the paper, $x^{'}$ should be $x'$ and $\theta^{'}$ should be $\theta'$.

(2) You stated 1-Wasserstein convergence result in Theorem 3.2 and then the 2-Wasserstein convergence result as a corollary in Corollary 3.3. I thought 2-Wasserstein upper bounds the 1-Wasserstein distance. Intuitively, I expect the bound in Corollary 3.3. trivially provides a 1-Wasserstein bound for Theorem 3.2. If you can obtain tighter 2-Wasserstein bound by establishing 1-Wasserstein bound first, it would be helpful if you can add some discussions and explanations.

(3) Flat minima phenomenon says that SGD favors flat local minima that lead to better generalization. In designing fSGLD, it seems that you are making the global minimum flatter. Does fSGLD also make local minima flatter or it will only lead to a flatter global minimum and if that is the case, will that help you with generalization performance?

---

> ### Author Response · Authors · 2025-11-21
>
> We sincerely appreciate the reviewer’s thorough feedback and constructive suggestions. Below, we provide detailed responses to each of the points raised.
>
> - **Response to Weaknesses (W)**
>
> **W1.**    We appreciate the reviewer’s suggestion. We emphasize that fSGLD achieves the same non-asymptotic convergence rates as classical SGLD, while introducing a provable bias toward flatter minima through randomized smoothing. This additional benefit comes without any extra computational cost and is not available in standard SGLD.
>
> To illustrate this distinction, consider two local minima $\theta_1, \theta_2$ with the same loss value $u(\theta_1)=u(\theta_2)$, but with different curvature $\mathrm{tr}H(\theta_1) > \mathrm{tr}H(\theta_2)$, so that $\theta_1$ is sharp and $\theta_2$ is flat, as depicted in Figure~1. Under SGLD, the Gibbs measure
> $$
> \pi_\beta^{\mathrm{SGLD}}(\theta)\propto e^{-\beta u(\theta)}
> $$
> assigns equal stationary mass to both minima, and therefore SGLD cannot distinguish sharp minima from flat ones.
>
> By contrast, the invariant distribution of fSGLD is
> $$
> \pi_{\beta,\sigma}^{\mathrm{fSGLD}}(\theta) \propto  \exp\left(-\beta v(\theta)\right), \qquad v(\theta)=u(\theta)+\frac{\sigma^2}{2}\mathrm{tr}H(\theta),
> $$
> which yields
> $$
> \frac{\pi_{\beta,\sigma}^{\mathrm{fSGLD}}(\theta_1)}{\pi_{\beta,\sigma}^{\mathrm{fSGLD}}(\theta_2)}=\exp\left[-\beta\frac{\sigma^2}{2}\big(\mathrm{tr}H(\theta_1)-\mathrm{tr}H(\theta_2)\big)\right]\ll 1.
> $$
> Hence, fSGLD assigns exponentially larger probability mass to the flatter minimum, even when the loss values are identical. This qualitative difference is visually captured in Figure 1 and provides a concrete example in which fSGLD theoretically outperforms SGLD.
>
> Finally, as $\sigma\to 0$, the fSGLD update reduces exactly to SGLD, and the constants in our non-asymptotic bounds collapse to the classical SGLD constants. We have added a dedicated remark (Remark 3.7) in the revised manuscript to make this connection explicit.
>
>
> **W2.** We thank the reviewer for the suggestion. Although Proposition 3.1 is stated in an asymptotic form for readability, the proof is fully non-asymptotic. Following the reviewer’s comment, we now explicitly present the Big-O expression in the statement of Proposition 3.1 and provide the full non-asymptotic expressions in the proof.
>
> ----------------------------------------------------------------------
>
> - **Response to Questions (Q)**
>
> **Q1.**  We thank the reviewer for pointing out this typo, which has been corrected in the revised manuscript.
>
> **Q2.**  We present the  error bound in $W_1$ (Theorem 3.2) first because it achieves the optimal convergence rate $O(\lambda^{\frac{1}{2}})$ with respect to the stepsize $\lambda$. This rate is known to be optimal for SGLD (e.g. see [1, Example 3.4]). The error bound in $W_2$ established in Corollary 3.3 is essential to establish the expected excess risk (Theorem 3.6), but it achieves a slower convergence rate of $O(\lambda^{\frac{1}{4}})$. This phenomenon is well known in the SGLD convergence analysis literature for the non-convex setting (e.g. [2,3]) and arises from the stringent functional inequalities required to control the discretization error. This is the reason why we present this result as a corollary.
>
> The convergence rate can be improved to $O(\lambda^{\frac{1}{2}})$ under significantly stronger assumptions, such as assuming that the target measure satisfies a log-Sobolev inequality, as done in [4]. However, this is beyond the scope of our work. We have clarified this reasoning in a short remark (Remark C.12) in the revised version.
>
> [1] Barkhagen, Mathias, et al., "On stochastic gradient Langevin dynamics with dependent data streams in the logconcave case." Bernoulli, 2021.
>
> [2] Majka, Mateusz, et al., "Nonasymptotic bounds for sampling algorithms without log-concavity". Annals of Applied Probability. 2020.
>
> [3] Zhang, Ying, et al., "Nonasymptotic estimates for stochastic gradient langevin dynamics under local conditions in nonconvex optimization." Applied Mathematics \& Optimization, 2023.
>
> [4] Huang, Jing, et al., "Nonasymptotic convergence analysis for the tamed unadjusted stochastic Langevin algorithm." arXiv, 2025.

---

> ### Author Response · Authors · 2025-11-21
>
> **Q3.**  Thank you very much for your insightful question. The mechanism that drives fSGLD toward flatter minima is rooted in the randomized smoothing surrogate objective $ g_{\epsilon}(\theta)
>    = \mathbb{E}[ u(\theta+\epsilon) ] $. Under the appropriate $\sigma-\beta$ coupling, fSGLD effectively solves an Hessian trace-regularized objective $v(\theta)
>    = u(\theta) + \frac{\sigma^{2}}{2}\text{tr} \left(H(\theta)\right)$ induced by the surrogate $ g_{\epsilon}(\theta)$. This regularization is applied globally across the landscape, not only near the global minimizer. Therefore, fSGLD does not ``flatten'' only the global minimum. It systematically penalizes sharp curvature around any point, which naturally biases the dynamics toward flatter local minima as well.
>
> This behavior is further supported by the Hessian spectrum analysis in Section 4.5, where we observe that fSGLD consistently suppresses large eigenvalues and induces a visibly flatter curvature profile. Furthermore, in all experiments, flatter minima found by fSGLD consistently lead to improved generalization performance.

---

### Official Review · Reviewer_vmbQ · 2025-11-01

**Soundness:** 2
**Presentation:** 3
**Contribution:** 2
**Rating:** 2
**Confidence:** 4

**Summary:**

This paper proposes a new method, flatness-aware stochastic Langevin dynamics (fSGLD), as an efficient method for finding flat minimizer in nonconvex problems. Specifically, fSGLD is a variant of the typical SGLD (SGD with an additive Gaussian noise), with the modification that the gradient is evaluated at the current weight perturbed by a Gaussian noise. Then this paper proves that, in a certain asymptotic sense, fSGLD converges to the stationary distribution of SGLD that applied to the original objective with sharpness penalty. This paper further provides empirical verifications on the effectiveness of fSGLD in CIFAR-type problems, matching or outperforming benchmark methods such as SGD, AdamW, and SAM.

**Strengths:**

See below.

**Weaknesses:**

See below.

**Questions:**

The idea of fSGLD comes from random weight perturbation. The main contribution of this work lies in that it provides theoretical guarantees on how fSGLD indeed produces an effect of sharpness penalty, measured by trace of the Hessian. Regarding this, I have the following main questions:

1. The Assumptions 2-3 are quite strong,in particular, the Assumption 3. It would be very helpful to provide concrete (nonconvex) examples that satisfy those conditions. I don’t see a meaningful example for Assumption 3 except for linear models. If I understand correctly, networks with more than two layers would typically violate Assumption 2.
2. Proposition 3.1 seems rather weak as $\alpha$ and $\beta$ are tied together. In particular, the equivalence only holds as $\beta \to \infty$, in which case $\alpha\to 0$, meaning that the sharpness penalty is asymptotic zero. In this sense, Proposition 3.1 does not reveal a direct connection between fSGLD and flat minimizer.
3. I find the condition $\alpha = o(\beta^{-1/4})$ not intuitive. Why is the exponent $1/4$ instead of $1/6$ or so? I tried to dig into the proof, and find that equation (16) only keeps order-4 components in the Taylor expansion – why do the higher order terms disappear here? One selling point of this paper (Line 71) is the promise of dealing with higher order terms. However, I don’t quite understand why order-6 and higher terms disappear – is it part of the assumption?
4. Theorem 3.2 and Corollary 3.3 have the similar issue, where the convergence only holds in the strong asymptotic sense that requires $\alpha\to 0$, thus their implication to finding a flat minimizer is limited.
5. The proof of Theorem 3.2 and Corollary 3.3 seems mainly using techniques from Zhang et al 2023. It would be helpful to clarify the novelty here.

Besides, I have a couple of minor suggestions.

6. Line 126. Computing the trace of Hessian does not need to be expensive in high dimensions. For example, one can avoid dimension-dependent cost by doing the trick of commuting gradient and linear operators and using monte carlo.
7. In Theorem 3.2 and Corollary 3.3, certain quantities are called “constants”, e.g., D with underline. However, these quantities go to zero in order for the bounds to be useful, which means they are not constants. Additionally, it would be better to make $\dot{c}, D_1, D_2, D_3$ etc explicit in the main paper – at least clarify which given constants they depend on.

8. Line206, "is is i.i.d." -> "is i.i.d."

---

> ### Author Response · Authors · 2025-11-21
>
> We sincerely appreciate the reviewer’s thorough feedback and constructive suggestions. Below, we provide detailed responses to each of the points raised.
>
> - **Responses to Questions (Q)**
>
> **Q1.**  We appreciate the reviewer’s careful reading of our assumptions. Assumptions 2–3 are standard in the SGLD literature: dissipativity and Lipschitz continuity conditions (see, e.g., [1, 2, 5]). In fact, our assumptions are more relaxed than the classical ones because we allow explicit dependence on the data $x$, which is essential in many machine learning settings. Furthermore, while the assumptions commonly adopted in the SGLD literature are typically global in $\theta$, our proof technique does not require global regularity. Assumptions 2 can be relaxed to local Lipschitz continuity in $\theta$ without major changes in the analysis. However, we chose to work with the current formulation to maintain a clear focus on the core mechanism introduced by fSGLD, rather than introducing additional technical layers.
>
> Compared to the assumptions used in the theoretical literature on flatness-aware optimization, including SAM, our Assumptions 2–3 are significantly weaker. Existing SAM analyses often rely on strong conditions such as the Polyak-Lojasiewicz (PL) condition, global Lipschitz continuous or even bounded gradients, bounded variance condition [6, 7, 8]. In addition, the data dependence accounted for in Assumptions 2-3 is largely absent from the assumptions used in the theoretical analyses available for SAM. By contrast, our assumptions explicitly incorporate stochastic gradients and account for their data dependence, allowing our results to characterize the behavior of fSGLD under realistic stochastic noise.
>
>    For concrete examples satisfying Assumptions 2–3, we refer the reviewer to Sections 3.1 and 3.2 of [3], which illustrate that global Lipschitz continuity typically fails in practical machine learning models, while Assumptions 2 and 3 remain valid. These examples provide explicit non-convex examples from variational inference for Bayesian logistic regression and index tracking optimization, demonstrating that Assumptions 2–3 naturally hold in these widely used settings. Lastly, we note that the dissipativity condition can often be ensured by adding a suitable regularization term to $u(\theta)$, see e.g. [1,5,4].
>
> [1]  Maxim Raginsky, Alexander Rakhlin, and Matus Telgarsky. Non-convex learning via stochastic gradient langevin dynamics: a nonasymptotic analysis, PMLR, 2017.
>
> [2] Ngoc Huy Chau, Eric Moulines, Miklos Rasonyi, Sotirios Sabanis, and Ying Zhang. On stochastic gradient langevin dynamics with dependent data streams: The fully nonconvex case. SIAM Journal on Mathematics of Data Science, 2021.
>
> [3] Ying Zhang, Omer Deniz Akyildiz, Theodoros Damoulas, and Sotirios Sabanis. Nonasymptotic
> estimates for stochastic gradient langevin dynamics under local conditions in nonconvex optimization. Applied Mathematics \& Optimization, 2023.
>
> [4] Attila Lovas, Iosif Lytras, Miklos Rásonyi, Sotirios Sabanis. Taming neural networks with tusla: Nonconvex learning via adaptive stochastic gradient langevin algorithms. SIAM Journal on Mathematics of Data Science, 2023.
>
> [5]  Pan Xu, Jinghui Chen, Difan Zou, and Quanquan Gu. Global Convergence of Langevin Dynamics Based Algorithms for Nonconvex Optimization. NeurIPS 2018.
>
> [6] Dimitris Oikonomou, Nicolas Loizou. Sharpness-Aware Minimization: General Analysis and Improved Rates. ICLR 2025.
>
> [7] Maksym Andriushchenko, Nicolas Flammarion. Towards understanding sharpness-aware minimization. ICML 2022.
>
> [8] Peng Mi, Li Shen, Tianhe Ren, Yiyi Zhou, Xiaoshuai Sun, Rongrong Ji, and Dacheng Tao. Make
> sharpness-aware minimization stronger: A sparsified perturbation approach. NeurIPS, 2022

---

> ### Author Response · Authors · 2025-11-21
>
> **Q2.** Proposition 3.1 has been updated to include the non-asymptotic error bound, which was previously in the Appendix  due to space limitations. In the proposition, we first set the perturbation scale $\sigma = \beta^{- \frac{1+\eta}{4}}$, and then provide the non-asymptotic error bound under this condition between the invariant measure $\pi_{\beta}^{\text{fSGLD}}$  and the measure $\pi_{\beta, \sigma}^{\star}$, which is the Gibbs distribution associated with the Hessian-trace regularized objective and concentrates around the global flat minimizers of $u$. With the coupling between $\beta$ and $\sigma$, the latter becomes
> \begin{equation}
>         \pi^{\star}_{\beta, \sigma} \propto \exp \left( - \beta u(\theta) - \frac{\beta \sigma^2}{2} \text{tr}(H(\theta)) \right) = \exp \left( - \beta u(\theta) - \frac{\beta^{\frac{1-\eta}{2}}}{2} \text{tr}(H(\theta)) \right).
> \end{equation}
>
> We refer to the discussion in Point 3 for the reason behind the choice of  $\sigma = \beta^{- \frac{1+\eta}{4}}$, i.e. to make the error bound small for large values of $\beta$. From the theory of the Langevin SDE [1], we know that the invariant measure $\pi_{\beta}^{\text{fSGLD}} \propto \exp(-\beta g_\epsilon(\theta))$ of
> $$
> d Z_t^{\text{fSGLD}}  = -    \nabla g_{\epsilon} (Z_t^{\text{fSGLD}} ) d t +  \sqrt{2 \beta^{-1}} \ d B_t, \qquad t \ge 0
> $$
> concentrates around the global minimizers of $g_{\epsilon}$ when $\beta$ is sufficiently large.
> In the limit $\beta \rightarrow \infty$, the invariant measure $\pi_{\beta}^{\text{fSGLD}}$ converges in $W_2$ to the Gibbs measure $\pi_{\beta, \sigma}^{\star}$. This demonstrates that the fSGLD algorithm converges to the Gibbs measure $\pi_{\beta, \sigma}^{\star}$ concentrating around the global flat minimizers of $u$ which is defined in equation (1) above.
>
> [1] Chii-Ruey Hwang. Laplace’s method revisited: weak convergence of probability measures. The Annals of Probability, 1980.
>
>
> **Q3.** We have clarified this point in the revised manuscript (Equation (17)). The key observation is that the expectation of the remainder term in the Taylor expansion (Equation (17)) satisfies $\mathbb{E} [\mathcal{R}(\theta, \epsilon)] = O(\sigma^4)$ since $\sigma \in (0,1)$ by Assumption 1. This directly implies that all sixth-order and higher-order contributions are automatically absorbed into the $O(\sigma^4)$ term. **No additional assumption such as bounded fourth-order derivatives are required for these  terms to vanish**.
>
> In particular, the condition $\sigma = \beta^{- \frac{1+\eta}{4}}$ is required to control the “distance’’ between the two measures $\pi_{\beta}^{\text{fSGLD}}$ and $ \pi_{\beta, \sigma}^{\star}$. This condition follows directly from the fact that $\mathbb{E} [\mathcal{R}(\theta, \epsilon)] = O(\sigma^4)$. As seen in the proof of Proposition 3.1, the term $e^{C_A \beta \sigma^4}$ in  Equation (47) becomes $e^{C_A \beta^{- \eta}}$ in Equation (48) with the $\beta$-$\sigma$ coupling. Thus, the exponent $1/4$ is not arbitrary. It is dictated by the order $O(\sigma^4)$ of the expectation of the remainder term.
>
> Lastly, we also draw the reviewer’s attention to lines 230–235 and lines 238-240 (before Proposition 3.1), and lines 248–251 (after Proposition 3.1), where this point is discussed at a higher level.
>
> **Q4.**  We respectfully emphasize that Theorem 3.2 and Corollary 3.3 are fully non-asymptotic and do not require the asymptotic condition $\sigma\rightarrow 0$.
>
> We have updated Theorem 3.2 and Corollary 3.3 to further clarify this point. In both results, the Wasserstein-1 and Wasserstein-2 error bounds can be made arbitrarily small (i.e. smaller than any prescribed tolerances $\bar{\delta}$ and $\widetilde{\delta}$) by choosing  the inverse temperature  $\beta \ge \beta_{\bar{\delta}} $, the stepsize $\lambda \le \lambda_{\bar{\delta}}$, and the iteration $k \ge k_{\bar{\delta}}$, where the corresponding values of $\beta_{\bar{\delta}} $, $\lambda_{\bar{\delta}}$, and $k_{\bar{\delta}}$ are given in the statements of Theorem 3.2 and Corollary 3.3.      Therefore, the convergence guarantees for fSGLD are obtained entirely in a non-asymptotic sense.

---

> ### Author Response · Authors · 2025-11-21
>
> **Q5.** We have added Remark 3.5 in the main body of the text to summarize the structure of the proofs of Theorem 3.2 and Corollary 3.3. Our proofs differ from those of Zhang et al. 2023 in several key aspects: we deal with a different objective function, namely the randomized-smoothing surrogate objective $g_{\epsilon}= \mathbb{E}[u(\theta+\epsilon) ]$ rather than the original objective function $u$, and we target a different measure, $\pi_{\beta, \sigma}^{\star}$, which concentrates around the global flat minimizers rather than $\pi_\beta (\theta) \propto \exp(-\beta u(\theta))$ considered in Zhang et al 2023.
>
> Most importantly, no previous work has analyzed how RWP, when incorporated into SGLD, affects the invariant measure or biases the dynamics toward globally flatter minima. This is a technically difficult problem because SGLD already injects Langevin noise. Adding RWP creates a second stochastic effect, and understanding how these two noise sources jointly influence global flatness requires new analysis. Our paper provides the first rigorous treatment of this interaction.
>
> In the Appendix, we clearly indicate where our arguments diverge from those of Zhang et al. 2023, and we explicitly note the steps where our proof strategy follows Zhang et al. 2023 or Chau et al 2021, which are standard in the literature (see e.g. 1145-1148, lines 1534-1537, and line 1577). To keep the convergence analysis of fSGLD self-contained, we chose to include the proofs that can be derived using our assumptions together with existing techniques from the literature.
>
> **Q6.**  Thank you very much for this important comment. We believe the reviewer is referring to the Hutchinson estimator when mentioning Monte-Carlo–based approaches for computing the trace of the Hessian. Indeed, the perturbation mechanism in our method inherits the same core idea behind Hutchinson’s estimator: symmetric random perturbations encode curvature information through the identity
>     $$
>     \frac{1}{2} \mathbb{E}[\epsilon^{\top} H(\theta)\epsilon ] = \frac{\sigma^2}{2}\text{tr}(H(\theta)).
>     $$
>     However, an important conceptual difference is that fSGLD does not use this idea as a computational trick inside the update rule. Instead, we embed the perturbation directly at the level of the objective function, which leads to a randomized smoothing surrogate of the form
> \begin{align*}
> g_\epsilon(\theta) &= u(\theta)  +\frac{1}{2} \mathbb{E}[\epsilon^\top H(\theta)\epsilon]  +\mathbb{E}[\mathcal{R}(\theta,\epsilon)]\\\\
> &= u(\theta) + \frac{\sigma^2}{2}\text{tr}(H(\theta))+\mathbb{E}[\mathcal{R}(\theta,\epsilon)]
> \end{align*}
>
> In this construction, the Hutchinson-style trace term naturally emerges as the second-order component of the smoothed objective, but crucially, the algorithm never needs to compute or even approximate $\text{tr}(H(\theta))$ or its gradient. Instead, the fSGLD update relies solely on perturbed gradient $\nabla u(\theta+\epsilon)$, making the algorithm simpler, more direct, and entirely free of Hessian-related overhead, while still achieving the desired curvature-aware behavior implied by the randomized smoothing surrogate.
>
> **Q7.** We have clarified this point in the updated manuscript. We have included the dimension and temperature dependence of the constants $D_1-D_6$ and $\underline{D}$ in Theorem 3.2 and Corollary 3.3 as well as references to the explicit expressions of the constants.  In addition, we have provided reference in Theorem 3.2 and Corollary 3.3 to the values of the key parameters $\beta$, $\lambda$, and $k$ making the error bounds in $W_1$ and $W_2$ arbitrarily small (i.e. the bounds achieves a given precision level $\bar{\delta}$ and $\widetilde{\delta}$). The "constants" does not go to zero as Theorem 3.2 and Corollary 3.3 are non-asymptotic error bounds. For this point, we refer the reviewer to our answers in Question 2, 3, 4 above.
>
> **Q8.**  We thank the reviewer for pointing out this typo, which has been corrected in the revised manuscript.

---

### Official Review · Reviewer_BTrV · 2025-11-02

**Soundness:** 3
**Presentation:** 3
**Contribution:** 2
**Rating:** 6
**Confidence:** 4

**Summary:**

The authors propose an algorithm for flatness-aware optimisation in non-convex settings. Their method combines stochastic gradient Langevin dynamics (SGLD) and random weight perturbations. They propose using SGLD applied to a Laplacian-regularised modification of the objective, estimating the Laplacian through random weight perturbations. They compare this method against standard optimisation techniques and SAM, the most popular flatness-aware optimiser, demonstrating improved results.

**Strengths:**

* Addresses a classical yet current problem in machine learning: tractable flatness-aware optimisation.
* Provides a rigorous mathematical proof of convergence.
* Using its theory, the authors identify a good rate of coupling between SGLD temperature and the scale of Laplacian regularisation. They then show that this coupling works well in their experiments.
* Not only obtains convergence bounds but also provides excess risk bounds.
* The experiments consider a realistic setting.

**Weaknesses:**

* The dependence on dimension in the bound appears poor (exponential in $d$), requiring the step size to decay exponentially with $d$. While common in the SGLD literature (and thus not a fundamental flaw in novelty), the authors should mention this limitation in the main body, as dimension-dependence is critical for the large-scale settings targeted in the experiments.
* A large portion of the proof appears to be borrowed from previous papers on SGLD, making it difficult to isolate the novel theoretical contributions. I would suggest including a discussion on what is new in the proof technique or, if the technique is standard, a statement clarifying that it is adapted from prior work.
* In general, there is little discussion surrounding the theoretical results, making them somewhat opaque to the theoretical audience this paper will likely appeal to. For example, it would be helpful to decompose the error terms in the bound (e.g., from convergence, discretisation, and objective estimation). In particular, what is the error contributed by the RWP estimator, and how does it scale with dimension?
* There is a mismatch between the theory (online setting) and the experiments (multi-pass setting). While this may not invalidate the results, it is a detail that the authors should disclose and briefly discuss.
* The authors do not appear to use early stopping, which is common for the baselines considered. It would be interesting to know if early stopping interacts with the curvature of the final minimisers. A plot of the train curve for these methods might help with aleviating this concern.
* It is difficult to find details in the paper due to imprecise referencing to the appendix. For example, the constants $D_1, \dots, D_7$ in the main theoretical statements are critical for understanding dimension and temperature dependence. The authors should reference the specific equations (e.g., (65) and (67)) where these are defined, rather than referencing the entire appendix section. Similarly, I struggled to find the "compelxity analysis" referenced in the main body.

**Questions:**

* This work targets the online setting in the theory and the multi-pass setting in the experiments. How does the theory compare in the multi-pass setting, and how do the experiments compare in the online setting?
* Is the computational cost truly comparable to SGD? Does the RWP and Langevvin noise in fSGLD not necessitate a smaller step-size than standard SGD, thus requiring more steps to converge?
* How does early stopping affect the experiments and the curvature of the resulting solutions?
* How does the bound compare to standard SGLD bounds? Which error terms are brought about by the RWP estimate?

---

> ### Author Response · Authors · 2025-11-21
>
> We sincerely appreciate the reviewer’s thorough feedback and constructive suggestions. Below, we provide detailed responses to each of the points raised.
>
> - **Response to Weaknesses (W)**
>
> **W1.**   We thank the reviewer for the helpful suggestion. We have now explicitly discussed this common limitation in the SGLD literature in Remark 3.4, immediately after the main results of Theorem 3.2 and Corollary 3.3. In particular, we point out that the dimension dependence arises from the coupling arguments of [1], which are standard in prior work, and that improving this dependence would require substantially stronger contraction-rate estimates.
>
> [1] Andreas Eberle, Arnaud Guillin, and Raphael Zimmer. Quantitative harris-type theorems for diffusions and mckean–vlasov processes. Transactions of the American Mathematical Society, 2019.
>
> **W2.**  We appreciate the reviewer’s comment. We have now added Remark 3.5 in the main text to clearly summarize the structure of the proofs of Theorem 3.2 and Corollary 3.3 and to highlight the novel components.
>
>    In the literature, most theoretical analyses of SGLD share a common backbone based on Lyapunov arguments, drift conditions, and Wasserstein contraction and our proof also follows this general framework. However, the key contributions of our analysis differ substantially from prior work. In particular, (i) we analyze the dynamics of fSGLD with respect to the randomized-smoothing surrogate objective $g_{\epsilon}$ rather than the original objective $u$, and (ii) we establish convergence toward the flatness-aware Gibbs measure $\pi^{\star}_{\beta,\sigma}$, which has no analogue in the standard SGLD setting.
>
> These two aspects require several nontrivial modifications to existing arguments in [1,2]. In the appendix, we now explicitly indicate which steps follow the shared SGLD backbone, so that the contributions of our analysis are transparent while keeping the proof self-contained.
>
> Finally, we emphasize a key conceptual novelty: there has been no prior work showing how RWP, when incorporated into SGLD, affects the invariant measure or biases the dynamics toward globally flatter minima. This is technically nontrivial because SGLD already injects Langevin noise. Adding RWP introduces a second stochastic effect, and understanding the joint influence of these two noises on global flatness is analytically difficult. Our paper provides the first rigorous treatment of this interaction.
>
> [1] Ngoc Huy Chau, Eric Moulines, Miklos Rasonyi, Sotirios Sabanis, and Ying Zhang. On stochastic gradient langevin dynamics with dependent data streams: The fully nonconvex case. SIAM Journal on Mathematics of Data Science, 2021.
>
> [2] Ying Zhang, Omer Deniz Akyildiz, Theodoros Damoulas, and Sotirios Sabanis. Nonasymptotic
> estimates for stochastic gradient langevin dynamics under local conditions in nonconvex optimization. Applied Mathematics \& Optimization, 2023.
>
> **W3.**  We thank the reviewer for this helpful suggestion. To improve clarity, we have added a new Remark~3.5 in the revised manuscript, which provides an explicit decomposition of the total error in our non-asymptotic analysis.
>
> In particular, Remark~3.5 shows that
> \begin{equation}
>      \begin{split}
>               W_p(\mathcal{L}(\theta_k^{\text{fSGLD}}), \pi_{\beta,\sigma}^{\star})  & \le
>    W_p (\mathcal{L}(\theta_k^{\text{fSGLD}}),  \mathcal{L}(Z_t^{\lambda, \text{fSGLD}  })) + W_p( \mathcal{L}(Z_t^{\lambda , \text{fSGLD}}), \pi_{\beta}^{\text{fSGLD}})  \\ & \quad + W_p( \pi_{\beta}^{\text{fSGLD}}, \pi_{\beta,\sigma}^{\star}), \quad p=\{1,2\}, \quad t  \in (kT, (k+1)T].
>      \end{split}
>  \end{equation}
> is decomposed into three components:
> (i) the discretization error between the fSGLD recursion and the Langevin SDE associated with the smoothed objective $g_{\epsilon}$, $W_p (\mathcal{L}(\theta_k^{\text{fSGLD}}),  \mathcal{L}(Z_t^{\lambda, \text{fSGLD}  }))$;
> (ii) the convergence error between this SDE and its invariant measure $\pi_{\beta}^{\mathrm{fSGLD}}$, $W_p( \mathcal{L}(Z_t^{\lambda , \text{fSGLD}}), \pi_{\beta}^{\text{fSGLD}})$;
> (iii) the discrepancy between $\pi_{\beta}^{\text{fSGLD}}$ and the flatness-aware target measure $\pi_{\beta,\sigma}^{\star}$, $W_p( \pi_{\beta}^{\text{fSGLD}}, \pi_{\beta,\sigma}^{\star})$.
>
> In particular, the third term is specific to our method and captures the effect of the RWP estimator. As established in Proposition 3.1, under the $\sigma$–$\beta$ coupling $\sigma=\beta^{-(1+\eta)/4}$, this contribution is with the rate $O(\beta^{- \frac{\eta}{2}} d \log d)$. We have now made this dependence explicit in the statements of Theorem 3.2 and Corollary 3.3 in the revised manuscript.

---

> ### Author Response · Authors · 2025-11-21
>
> **W4.** We thank the reviewer for raising this point. Our theoretical results are established under the standard i.i.d. sampling model where a sample $X_k$ is drawn at each iteration. In contrast, our experiments follow the common practice in deep learning, training over multiple epochs using uniform sampling without replacement. These two sampling regimes are not identical because sampling without replacement is not strictly i.i.d. However, it is well established in stochastic optimization that both regimes induce gradient noise with very similar statistical properties, particularly in large-dataset settings.
>
>
>  Sampling with replacement would align exactly with our theoretical setup. In application domains where sample generators are available, such as financial modeling [1], this online sampling arises naturally. Since our empirical evaluation focuses on settings that are standard in the deep learning community, we adopt the widely used multi-pass training protocol for benchmarking. We now clarify this distinction explicitly in the revised manuscript after our theoretical assumption (Assumption 1) in the main body and with a remark (Remark B.4) in the Appendix.
>
>  [1] Chu, Jiarui, and Ludovic Tangpi. "Nonasymptotic Estimation of Risk Measures Using Stochastic Gradient Langevin Dynamics." SIAM Journal on Financial Mathematics, 2024.
>
> **W5.** We thank the reviewer for raising this important point. In our experiments, we trained all methods for the full number of epochs and reported the best  test performance. To address the reviewer’s suggestion, we have added the full training curves for all methods in Appendix D.3.
>
> These curves show that all optimizers reach their best test accuracy well before the final epoch, demonstrating that the training dynamics have clearly converged prior to the end of training. Thus, the performance we report is effectively identical to what would be obtained with an early-stopping rule. Thus, the absence of explicit early stopping does not affect the comparison across methods.
>
> Regarding the interaction between early stopping and the curvature of the final minimizers, we clarify that the curvature analysis in Section 4.5 is not computed at the last iterate, but rather at the checkpoint that achieves the best performance.
>
> **W6.**  We have updated the statements in Theorem 3.2, Corollary 3.3, and Theorem 3.6 to include the dependence of the constants on the dimension and temperature, and we now provide references to their explicit expressions. In all statements, we also specify the values of the key parameters (temperature $\beta$, stepsize $\lambda$, and iteration $k$) that ensure the error bounds in $W_1$, $W_2$, and the expected excess risk can be arbitrarily small.
>
> -------------------------------------
> - **Response to Questions (Q)**
>
> **Q1.** Please refer to our response to W4.
>
> **Q2.** We thank the reviewer for raising this excellent question, as it highlights an important aspect of how fSGLD behaves both theoretically and in practice.
>
> From a theoretical perspective, the Langevin noise injected through the inverse temperature $\beta$ does not impose additional stability constraints on the step size. As shown in Equation (24), the upper bound $\lambda_{\max}$ depends only on the dissipativity and Lipschitz constants, and is completely independent of $\beta$. Therefore, the stability range of admissible step sizes is comparable to that of SGD in practice.
>
> On the other hand, in practical settings, we agree with the reviewer that standard SGLD often requires a smaller optimal learning rate than SGD because the injected noise can slow down convergence. However, fSGLD behaves differently: when $\beta$ and $\sigma$ are properly coupled (as prescribed by our theory), this coupling allows fSGLD to maintain an optimal step size comparable to SGD, avoiding the small-step-size limitation typical of vanilla SGLD. The optimal learning rates for each method are reported in Appendix D.3. As a result, fSGLD converges at a speed comparable to SGD and often reaches its best epoch earlier, as shown in Appendix D.3.
>
> **Q3.**  Please refer to our response to W5.
>
> **Q4.** We thank the reviewer for raising this important question. As $\sigma\to 0$, the fSGLD update reduces exactly to SGLD, and the constants in our non-asymptotic bounds collapse to the classical SGLD constants. We have added a dedicated remark (Remark 3.7) in the revised manuscript to make this connection explicit. The other question has been answered in Weaknesses point 3 above.

---

### Author Response · Authors · 2025-11-21

Dear Area Chair,

We would like to raise serious concerns regarding the review provided by Reviewer QXeJ. Our concern is not about differing opinions, but about the fact that the reviewer's evaluation is built on a fundamentally incorrect understanding of the core theoretical framework of our paper. As a result, the majority of the reviewer’s criticisms do not meaningfully engage with the actual content of the submission.

> **1. The evaluation is based on a fundamental misunderstanding of SGLD theory**

The reviewer repeatedly confuses two entirely different frameworks:

(i) continuous-time derivations for SGD or SAM, and

(ii) the Langevin diffusion underlying SGLD.

This confusion is the basis of the reviewer’s ``major concern'', where the reviewer attempts to derive an SDE for SGLD using the SGD-style continuous-time approach. This derivation is mathematically invalid for Langevin-based algorithms, as we explained in our response to the reviewer’s question. Since our work is fully built on the SGLD framework, this misunderstanding undermines the foundation on which the reviewer’s evaluation is constructed. The reviewer’s confidence score of 4 stands in stark contrast to this basic conceptual error.

> **2. The listed weaknesses follow directly from this misunderstanding**

Because the reviewer misinterprets the fundamental framework, several criticisms are misdirected:

- “Questionable premise on flatness and generalization” dismisses an active research area using a single counterexample, without engaging with the extensive literature or with our explicit discussion in Section 5.
- "Missing discussion of implicit regularization in SAM" arises entirely from treating SGLD as if it were SGD or SAM, which makes the requested comparison irrelevant to our method.
- “Limited novelty” overlooks the central theoretical contribution. Assessing the novelty of this contribution requires an accurate understanding of the frontier and current state of the SGLD literature, yet the reviewer raises this point without providing any scientifically grounded or specific comments. The reviewer’s assessment appears fundamentally unreliable, because several parts of the review indicate that the reviewer does not understand the basic principles of SGLD.

We also note that many parts of the review felt unexpectedly harsh in tone. Nevertheless, we approached every point with respect and provided detailed, careful, and responsible responses.

Given that the reviewer’s evaluation rests on an incorrect understanding of the theoretical foundations of SGLD, and therefore cannot reliably judge the theoretical contribution of the paper, we respectfully ask the AC to take this into consideration and ensure that the submission is evaluated fairly and accurately.

Thank you for your time and service to the community.

Sincerely,

The authors

---

### Author Response · Authors · 2025-12-03

The authors thank the reviewers for insightful comments and valuable suggestions. We have addressed each weakness and question raised by the reviewers and ensured these changes are included in the revised manuscript. For the AC, we summarize the key updates made during the rebuttal process as follows:

- ### Clarification of non-asymptotic convergence guarantees and dimension dependence
*(response to Reviewers BTrV,vmbQ, K5tw)*

We updated the statements of the main non-asymptotic results (Proposition 3.1, Theorem 3.2, Corollary 3.3, and Theorem 3.6) to include the dependence of the constants on the dimension and temperature, and we now provide references to their explicit expressions in the Appendix. In all statements, we also specify the values of the key parameters that ensure the non-asymptotic error bounds and the expected excess risk can be arbitrarily small.

We included a discussion (Remark 3.4) about the common limitation in the SGLD literature for the dependence of the dimension in the constants of the error bounds in the main non-asymptotic results.



- ### Novel theoretical analysis of our work
*(response to Reviewers BTrV, vmbQ)*

Most theoretical analyses of SGLD share a common backbone based on Lyapunov arguments, drift conditions, and Wasserstein contraction, and our proof also utilizes this general backbone. However, the key contributions of our analysis differ substantially from prior work. In particular, (i) we analyze the dynamics of fSGLD with respect to the randomized-smoothing surrogate objective $g_{\varepsilon}$ rather than the original objective $u$, and (ii) we establish non-asymptotic convergence toward the flatness-aware Gibbs measure $\pi_{\beta,\sigma}^{\star}$, which has no analogue in classical SGLD and encodes a provable preference for flatter minima. These two aspects require nontrivial modifications to standard techniques in SGLD analyses, as the joint effect of Langevin noise and random perturbations introduces additional discretization and bias errors that must be controlled simultaneously.

To improve clarity, **Remark 3.5** was added in the revised manuscript to summarize the structure of the proofs and explicitly indicate where new technical ingredients are introduced.


- ### Alignment between theoretical sampling model and numerical experiments
*(response to Reviewer BTrV)*

Our theory assumes i.i.d. sampling, whereas experiments use the standard practice of multi-epoch uniform sampling without replacement. Although not identical, both regimes exhibit similar gradient-noise behavior in large-scale learning. We now clarify this after Assumption 1 and in Remark B.4.

- ### Early stopping and training curves
*(response to Reviewer BTrV)*


We clarified that all models were trained for the full number of epochs, while evaluation was always performed at the checkpoint achieving the best test accuracy. Appendix D.3 now includes full training curves for every method, showing that best performance is reached well before the final epoch. Therefore, the results reported in the paper are effectively equivalent to using early stopping. Additionally, we emphasize that the curvature analysis in Section 4.5 is conducted at the best-performing checkpoint, not at the final iterate, ensuring that curvature comparisons remain fair across methods.

---

### Author Response · Authors · 2025-12-03

Dear Area Chair,

We would like to summarize the discussion we had with Reviewer QXeJ before the OpenReview leakage freeze. We sincerely appreciate the reviewer’s active engagement and thoughtful effort during the rebuttal process.

At the same time, we note that the reviewer raised the most negative evaluation for our submission. However, as documented below, their concerns were rooted in a fundamental misconception of standard SGLD theory rather than any flaw in our results. Our responses were focused on correcting this misunderstanding, but the discussion had to stop due to the review freeze. Below, we provide a concise timeline of the key points addressed:

1. Initial review

In the initial review, most of the questions focused on introductory and well-established principles in the theory of SGLD and Langevin-type SDEs. It soon became clear that the reviewer was applying continuous-time SME-based analysisfor SGD/SAM analysis to the Langevin SDE framework that rigorously defines SGLD. This conflation of two theoretically distinct frameworks underpinned the reviewer’s concerns at this stage.

2. First Author Response

In our first response, we carefully and respectfully clarified the theoretical distinction between Langevin-based SGLD and SME-based continuous-time approximations for SGD. We restated the standard derivation of SGLD from the Langevin diffusion and pointed the reviewer to well-established foundational literature. We also explained that applying SME logic to SGLD is conceptually incorrect, since the two frameworks differ fundamentally in their goals, assumptions, and mathematical structure.

3. Reviewer Follow-up

In the reviewer’s follow-up, the discussion shifted away from our proposed method and toward foundational aspects of SGLD theory. The reviewer attempted to provide an alternative derivation of the continuous-time dynamics but continued to apply SME-based reasoning, insisting that the limiting SDE “must” retain step-size–dependent perturbation terms. While such terms indeed appear in SME formulations of SGD/SAM, they necessarily vanish in the Langevin limit that rigorously underpins SGLD. This demonstrated that the core theoretical misunderstanding from Stage 1 remained unresolved at this point.

4. Second Author Clarification

We clearly demonstrated why the reviewer’s continuous-time derivation was incorrect: any term proportional to the step size $\lambda$ must vanish as $\lambda \to 0$. We again emphasized the distinction between the **limiting SDE** and the **SME**, noting that the SME converges to the Langevin SDE, not the other way around. We also highlighted that the discretization error is already controlled in our analysis:
\begin{equation}
     \begin{split}
              W_p(\mathcal{L}(\theta_k^{\text{fSGLD}}), \pi_{\beta,\sigma}^{\star})  & \le
   W_p (\mathcal{L}(\theta_k^{\text{fSGLD}}),  \mathcal{L}(Z_t^{\lambda, \text{fSGLD}  })) + W_p( \mathcal{L}(Z_t^{\lambda , \text{fSGLD}}), \pi_{\beta}^{\text{fSGLD}})  \\ &+ W_p( \pi_{\beta}^{\text{fSGLD}}, \pi_{\beta,\sigma}^{\star}), \quad t\in(kT, (k+1)T)
     \end{split}
 \end{equation}
The first term is exactly the reviewer’s concern, and we explicitly show that its upper bound is **proportional to $\lambda$**, becoming **smaller as $\lambda$ decreases**. Our theory therefore already formalizes the correct convergence behavior.

5. Reviewer's numerical code

We appreciated the reviewer’s attempt to verify their concerns through a numerical experiment comparing the stationary behavior of the exact-gradient and unbiased-noisy-gradient implementations. However, the results directly supported the established theory. As the step size decreased, both implementations converged to the same stationary OU distribution $N(0,1)$,exactly as predicted. We repeated the test with an smaller step size and observed clearer convergence, reinforcing that the reviewer’s own code validates the correct SGLD behavior.

6. Reviewer’s shift to philosophical arguments about flatness and generalization

Beyond the SDE and SME confusion, the reviewer’s later comments shifted toward broad philosophical arguments about flatness and generalization. These points were not related to our technical contributions and primarily reflected a personal disagreement with the broader research direction rather than an evaluation of the correctness or relevance of our results. Additionally, the cited literature does not pertain to the theoretical framework of SGLD, making it orthogonal to the scope and claims of our work.

7. Authors’ clarification regarding philosophical concerns

We clarified that the philosophical remarks raised by the reviewer do not affect the correctness or scope of our contribution. Our work provides a theoretical advance: the first rigorous analysis demonstrating how randomized weight perturbations interact with Langevin noise in SGLD to induce a provable preference for flatter minima while maintaining convergence guarantees.

---

### Meta-Review · Area_Chair_R64t · 2026-01-03

**Summary:**

The reviewers had concerns about the validity of the assumptions for theoretical analysis, the novelty of the algorithmic setting, proof issues (why ignoring high-order terms) and the sufficiency of the experiments. Some of the critical questions have not been addressed according to the rebuttals (including the tied parameters alpha and beta does not directly show the connection to flat minima; and the validity of ignoring particular high-order terms), though some reviewers partially acknowledged the theoretical contribution of this work. I thus recommend a week reject. However, this is still a borderline paper.

**Reviewer Concerns:**

Reviewer BTrV's concerns have been well addressed in the rebuttal.

Reviewer vmbQ's concerns regarding the tied alpha and beta remains outstandings, near-zero sharpness penalty when beta is very large.  Additionally, the validity of ignoring particular high-order terms has not been addressed well.

Reviewer K5tw's concern about the iteration-complexity comparison between fSGLD and SGLD remains unaddressed.

Reviewer QXeJ engaged in the discussion with the authors actively during the rebuttal period. As far as I read the discussion process, I think the reviewer's concerns are addressed well.

**Reviewer Scores:**

Reviewer BTrV and K5tw will maintain their scores, still postively.
Reviewer vmbQ might keep his original rating 2.
Reviewer QXeJ might raise the score from 0 to a certain higher score according to the discussion process.

---

### Decision · Program_Chairs · 2026-01-26

Reject